# Cirrus cloud thinning using a more physically-based ice microphysics scheme in the ECHAM-HAM GCM

Colin Tully , David Neubauer , Nadja Omanovic , and Ulrike Lohmann

Institute for Climate and Atmospheric Science, ETH Zurich, Zurich, Switzerland

**Correspondence:** Colin Tully (colin.tully@env.ethz.ch) or Ulrike Lohmann (ulrike.lohmann@env.ethz.ch)

**Abstract.** Cirrus cloud thinning (CCT) is a relatively new radiation management proposal to counteract anthropogenic climate warming by targeting Earth's terrestrial radiation balance. The efficacy of this method was presented in several general circulation model (GCM) studies that showed widely varied radiative responses, originating in part from the differences in the representation of cirrus ice microphysics between the different GCMs. The recent implementation of a new, more physically-based ice microphysics scheme (Predicted Particle Properties, P3) that abandons ice hydrometeor size class separation into the ECHAM-HAM GCM, coupled to a new approach for calculating cloud fractions that increases the relative humidity (RH) thresholds for cirrus cloud formation, motivated a reassessment of CCT efficacy. In this study, we first compared CCT sensitivity between the new cloud fraction approach and the original ECHAM-HAM cloud fraction approach. Consistent with previous approaches using ECHAM-HAM, with the P3 scheme and the higher RH thresholds for cirrus cloud formation, we do not find a significant cooling response in any of our simulations. The most notable response from our extreme case is the reduction of the maximum global-mean net top-of-atmosphere (TOA) radiative anomalies from overseeding by about $50\,\%$, from $9.9\,\mathrm{Wm^{-2}}$ with the original cloud fraction approach, down to $4.9\,\mathrm{Wm^{-2}}$ using the new cloud fraction RH thresholds that allow partial gridbox coverage of cirrus clouds above ice saturation, unlike the original approach. Even with this reduction with the updated cloud fraction approach, the TOA anomalies from overseeding far exceed those reported in previous studies. We attribute the large positive TOA anomalies to seeding particles overtaking both homogeneous nucleation and heterogeneous nucleation on mineral dust particles within cirrus clouds to produce more numerous and smaller ice crystals. This effect is amplified by longer ice residence times in clouds due to the slower removal of ice via sedimentation in the P3 scheme. In an effort to avoid this overtaking effect of seeding particles, we increased the default critical ice saturation ratio ($S_{i,seed}$) for ice nucleation on seeding particles from the default value of 1.05 to 1.35 in a second sensitivity test. With the higher $S_{i,seed}$ we drastically reduce overseeding, which suggests that $S_{i,seed}$ is a key factor to consider for future CCT studies. However, the global-mean TOA anomalies contain high uncertainty. In response, we examined the TOA anomalies regionally and found that specific regions only show a small potential for targeted CCT, which is partially enhanced by using the larger $S_{i,seed}$. Finally, in a seasonal analysis of TOA responses to CCT, we find that our results do not confirm the previous finding that high-latitude wintertime seeding is a feasible strategy to enhance CCT efficacy, as seeding in our model enhances the already positive cirrus longwave cloud radiative effect for most of our simulations. Our results also show feedbacks on lower-lying mixed-phase and liquid clouds through the reduction of ice crystal sedimentation that reduces cloud droplet depletion and results in stronger cloud albedo effects. However, this is outweighed by stronger longwave trapping from cirrus clouds with more numerous and

smaller ice crystals. Therefore, we conclude that CCT is unlikely to act as a feasible climate intervention strategy on a global scale.

## 1 Introduction

Limiting 21st century global average warming to within $2\,°C$, following the 2015 Paris Climate Agreement, through greenhouse gas emissions reduction alone remains a highly ambitious goal. Amid growing concern of this infeasibility, several climate intervention (CI, also referred to as climate engineering or geoengineering) methods were proposed as potential mitigation strategies in order to limit future warming (Vaughan and Lenton, 2011). CI strategies encompass carbon sequestration, which
targets one of the main drivers of anthropogenic climate change, namely increased atmospheric $CO_2$ concentrations, and radiation management (RM), which indirectly counteracts warming by altering Earth's radiation balance. These RM schemes can be further divided between solar, shortwave (SW) and terrestrial, longwave (LW) radiation strategies. The focus of this study is on one particular LW radiation strategy, cirrus cloud thinning (CCT), also referred to as cirrus seeding, that aims to increase the amount of outgoing LW radiation to space by altering the formation pathways of cirrus clouds using artificial ice
nucleating particles (INPs).

Cirrus clouds are found in the upper troposphere at temperatures below $238\,K$ (cirrus regime) and as such consist entirely of ice crystals. Unlike their lower-altitude mixed-phase or liquid counterparts, cirrus clouds possess a relatively weak SW albedo effect while significantly modulating outgoing LW radiation. They absorb LW radiation emitted at warmer temperatures from Earth's surface and the lower-lying atmosphere, and re-emit it at their lower temperatures, resulting in a top-of-atmosphere
(TOA) "trapping" effect that warms the atmosphere below (Hong et al., 2016; Gasparini et al., 2020). However, the magnitude of this cirrus cloud radiative effect (CRE) is strongly influenced by the microphysical properties of the clouds (e.g. the ice crystal number concentration (ICNC) and ice crystal sizes), which in turn are determined by the ice formation pathways (Stephens et al., 1990; DeMott et al., 2003, 2010; Krämer et al., 2016; Heymsfield et al., 2017).

Ice formation in cirrus occurs via two modes: homogeneous and heterogeneous nucleation. The former occurs as the spon-
taneous freezing of aqueous solution droplets at a relative humidity with respect to ice between $150\,\%$ and $170\,\%$ (Koop et al., 2000; Kärcher and Lohmann, 2002; Heymsfield et al., 2017) in the absence of a surface for ice nucleation. Due to the stochastic nature of a homogeneous nucleation event, numerous ice particles can form (Krämer et al., 2016; Heymsfield et al., 2017; Gasparini et al., 2018) that are limited in size due to their competition for the available water vapor (Ickes et al., 2015). The resulting cirrus ICNC, however, is sensitive to the appropriate conditions, namely the updraft speed that determines the magni-
tude of ice supersaturation (Kärcher and Lohmann, 2002; Lohmann and Kärcher, 2002; Kärcher et al., 2006; Kuebbeler et al., 2014; Jensen et al., 2016b).

Heterogeneous ice nucleation occurs on the surface of a solid aerosol particle called an INP. The availability of the INP surface lowers the energy barrier for ice germ formation, allowing ice nucleation at lower ice supersaturations and higher temperatures than homogeneous freezing. However, understanding how heterogeneous nucleation impacts cirrus cloud properties
is complicated by the fact that several mechanisms exist for ice formation via an INP (Heymsfield et al., 2017). Plus, only a

small fraction of aerosols acts as INPs, which are even more sparsely populated in the upper troposphere, with limited measurements in the cirrus regime (DeMott et al., 2003, 2010; Cziczo et al., 2013). Significant research continues on the ability of various materials (e.g. mineral dust (Möhler et al., 2008; Lohmann et al., 2008; Murray et al., 2012; Ullrich et al., 2017), and aircraft soot (Mahrt et al., 2018, 2020; Lohmann et al., 2020)) to act as INPs (Kanji et al., 2017).

The differences in the ice formation pathways via the two nucleation modes can result in cirrus clouds with different properties (Krämer et al., 2016; Heymsfield et al., 2017; Gasparini et al., 2018). While homogeneous nucleation tends to form numerous small ice crystals, the number of ice particles formed by heterogeneous nucleation is dependent on the availability of INPs, especially in the case of slow updrafts (Kärcher and Lohmann, 2003; Spichtinger and Cziczo, 2010). In the case of stronger updrafts or in an environment with a low INP concentration, heterogeneous nucleation may not be sufficient to deplete

the excess water vapor so that homogeneous nucleation occurs in addition (DeMott et al., 2010; Jensen et al., 2016b). Krämer et al. (2016) and Gasparini et al. (2018) reported noticeable differences in the ice water content (IWC) of cirrus formed directly from the gas phase ("in-situ") via the two nucleation modes, with heterogeneously-formed cirrus associated with having lower IWC and smaller ICNC than homogeneously-formed cirrus. Differences are also evident in ice particle sizes, which are indirectly related to the ICNC, with fewer, larger particles in heterogeneously-formed cirrus than numerous small particles in

homogeneously-formed cirrus (Heymsfield et al., 2017). DeMott et al. (2010) found that the smaller ice particles formed by homogeneous nucleation form cirrus clouds at higher altitudes (i.e. colder temperatures), contributing to a stronger warming effect. The fewer and larger ice particles formed on INPs result in lower and warmer cirrus that have a weaker warming effect. The differences in radiative effects between the ice nucleation modes was also assessed by Lohmann et al. (2008) with the ECHAM general circulation model (GCM). In a series of sensitivity tests they found that switching cirrus ice nucleation from

homogeneous only to purely heterogeneous nucleation reduced the net cloud radiative forcing by roughly $2\,\mathrm{Wm^{-2}}$. A similar response was found when a simplified simulation of competition between the two nucleation modes in the cirrus regime was included. The responses can be explained through changes in ice crystal fall speeds, which are closely related to nucleation rates that determine the initial size of the ice crystals (Mitchell et al., 2008). Following these findings, Mitchell and Finnegan (2009) were the first to propose using efficient artificial INPs (i.e. "seeding particles") to alter cirrus ice environments away

from small ice particles formed via homogeneous nucleation to predominantly larger ice particles formed via heterogeneous nucleation that sediment quicker and reduce cirrus cloud lifetimes, following a process coined as the negative Twomey effect (Kärcher and Lohmann, 2003). In the preliminary analysis by Mitchell and Finnegan (2009), they proposed CCT could have a cooling potential of about $-2.8\,\mathrm{Wm^{-2}}$ that could noticeably counteract warming from a doubling of $CO_2$.

    Natural nucleation competition in cirrus was excluded in the first dedicated modeling study of CCT by Storelvmo et al.

(2013), who assumed all cirrus formed via homogeneous nucleation in the CAM5 general circulation model (GCM). Globally uniform seeding produced a maximum negative net $\Delta$CRE around $-2.0\,\mathrm{Wm^{-2}}$, corresponding to optically thinner cirrus with an average ice crystal effective radius increase of $4\,\mu\mathrm{m}$ and decrease of ICNC by more than $250\,\mathrm{L^{-1}}$. Of note from their study was evidence of an optimal seeding particle concentration around $18\,\mathrm{L^{-1}}$, below which the seeding particles were ineffective due to insufficient water vapor consumption. However, a seeding concentration above the optimal concentration led

to "overseeding", whereby the numerous seeding INPs formed smaller ice particles that elongated cirrus lifetimes and exerted a warming effect (Storelvmo et al., 2013).

The assumption that cirrus form primarily by homogeneous nucleation was challenged when Cziczo et al. (2013) observed heterogeneous nucleation as the dominant source of cirrus ice over North and Central America. To account for the uncertainty surrounding the dominant ice nucleation mode in cirrus, Storelvmo and Herger (2014) conducted several seeding simulations with different configurations of ice nucleation competition, including different concentrations of background dust as active INPs. They found a reduced CRE response up to $-2\,\mathrm{Wm^{-2}}$ in their simulations where seeding particles were added to homogeneous-heterogeneous nucleation competition and homogeneous-only configurations, with an optimal seeding particle concentration of $18\,\mathrm{L^{-1}}$ as in Storelvmo et al. (2013). Additionally, they found that seeding at this optimal concentration in their model led to optically thinner clouds that contained a weaker overall SW CRE (i.e. reduced albedo), allowing more SW to reach the surface. However, this effect was outweighed by the reduction in cirrus LW CRE (i.e. reduced LW "trapping"). To some extent, this finding is in line with the latest compilation of in-situ observations of unseeded cirrus by Krämer et al. (2020), who found that optically thicker, liquid-origin cirrus (cloud optical depth, $\tau > 1$) tend to have a strong cooling effect due to a higher albedo, whereas optically thinner, in-situ origin cirrus ($\tau < 1$) have a large warming effect in response to a weaker albedo and a larger LW-trapping potential (i.e. cooler temperatures) that peaks with $\tau$ between 0.4 and 0.5. Krämer et al. (2020) further divide in-situ origin cirrus between fast and slow updrafts, with the latter having a stronger warming potential than the former. As CCT targets the slower updraft cirrus, due to weaker dynamic forcing (Gasparini et al., 2017; Krämer et al., 2016; Krämer et al., 2020), thinning these cirrus weakens their warming potential. Therefore, reducing the optical thickness of these latter cirrus through seeding, like in Storelvmo and Herger (2014), not only reduces their already weak SW CRE, but reduces their LW CRE more effectively. At higher seeding particle concentrations and for their heterogeneous-only simulation, Storelvmo and Herger (2014) found warming of more than $1.0\,\mathrm{Wm^{-2}}$ as a result of overseeding. They also showed that non-uniform seeding of only $40\,\%$ or $15\,\%$ of the globe, to avoid ineffective regions like the tropics, has a cooling potential similar to their uniform cases due to a lack of cirrus SW radiative effect at higher latitudes in winter and a reduced natural background aerosol loading. Seeding a smaller area around $15\,\%$ of the globe in winter resulted in a similar $\Delta$CRE response of $-2.1\,\mathrm{Wm^{-2}}$, through mostly LW cloud forcing reduction while avoiding large compensating SW forcing increases (Storelvmo et al., 2014). Similarly, Gruber et al. (2019) simulated CCT using the higher-resolution ICON-ART model in a small region in the Arctic centered over Greenland. They also found large negative TOA LW anomalies from seeding, but only in their simulations where background mineral dust concentrations were limited. The CCT cooling potential decreased in their simulations with increasing background mineral dust concentrations.

Penner et al. (2015) re-evaluated the results by Storelvmo et al. (2013), Storelvmo and Herger (2014), and Storelvmo et al. (2014) using an updated version of CAM5 that not only included the cirrus ice nucleation competition between homogeneous and heterogeneous nucleation, but also accounted for the consumption of water vapor by pre-existing ice transported into the cirrus regime. Additional updates were made to the dynamical environment to allow higher updraft velocities for the cirrus ice nucleation scheme, and to the aerosol environment to include secondary organic aerosols (SOAs) as potential INPs. Only their seeding simulation with no pre-existing ice, no SOAs acting as INPs, and a limited updraft velocity showed any significant

net negative TOA forcing up to $-0.74\,\mathrm{Wm}^{-2}$ in a similar optimal seeding particle concentration range as found by Storelvmo et al. (2013). All other simulations that included higher concentrations of INPs and higher updraft velocities resulted in positive net forcings. Gasparini and Lohmann (2016) extended these results using the ECHAM-HAM GCM with a cirrus ice nucleation scheme that also considered the competition between homogeneous and heterogeneous nucleation, and water vapor consumption on pre-existing ice (Kärcher et al., 2006; Kuebbeler et al., 2014). Like Storelvmo et al. (2013), Storelvmo and Herger

(2014), and Penner et al. (2015), Gasparini and Lohmann (2016) also reported an optimal seeding particle concentration, but its magnitude of $1\,\mathrm{L}^{-1}$ was an order of magnitude lower than previous studies. The maximum net TOA negative forcing in their full nucleation competition setup with the optimal seeding particle concentration was $-0.25\,\mathrm{Wm}^{-2}$, which was also smaller than in previous studies. Seeding with more than $1\,\mathrm{L}^{-1}$ resulted in warming from overseeding, which could be limited by the presence of pre-existing ice particles. However, in all of their simulations the net TOA responses contained high uncertainty.

Overall, the more positive forcing responses presented by Gasparini and Lohmann (2016) were attributed to a decrease in the average size of ice crystals post-seeding, and an increase in cirrus coverage in previously clear-sky areas, a potential side effect of seeding presented by Mitchell and Finnegan (2009). The efficiency of the seeding particles to consume water vapor was cited as the cause of the observed IC response, and as they highlight, points to the dominance of heterogeneous nucleation to background cirrus formation in ECHAM-HAM. A source attribution analysis revealed that most cirrus formed

via heterogeneous nucleation at a typical altitude of $200\,\mathrm{hPa}$, even in high latitude regions (Gasparini and Lohmann, 2016), contrasting previous studies by Storelvmo et al. (2013), Storelvmo and Herger (2014), and Penner et al. (2015). This difference between the nucleation mode dominance in different model setups is further evaluated in Gasparini et al. (2020), where even without seeding the global mean cirrus CRE is $2.0\,\mathrm{Wm}^{-2}$ greater in CAM5 than in ECHAM. With more heterogeneous nucleation present in cirrus in ECHAM-HAM, it is less sensitive to seeding and has a much lower optimal seeding particle

concentration than CAM5 (Gasparini et al., 2020). Overseeding can therefore occur more readily as water vapor consumption affects more particles.

Unintended side effects are likely with any climate intervention strategy. For example, a widely studied solar radiation management strategy, stratospheric aerosol injection, aims to increase planetary albedo by mimicking natural sulphur aerosol perturbations from volcanoes (Robock, 2000; Crutzen, 2006). However, numerous studies found that injecting such particles

into the stratosphere may deplete ozone and reduce the efficacy of renewable energy production (Crutzen, 2006; Robock et al., 2008; Murphy, 2009; Vaughan and Lenton, 2011). Alternatives to sulphur particles, like calcite, were investigated and found to lead to increased stratospheric ozone (Dykema et al., 2016; Keith et al., 2016). Stratospheric aerosol injection may also impact cirrus clouds (Kuebbeler et al., 2012; Cziczo et al., 2019). In summary, assessing the potential side effects of any climate intervention strategy is crucial in order to understand future implementation.

To date, assessing the climate impact of CCT is limited to global or regional modeling studies that require a comprehensive understanding of the complex ice processes occurring in cirrus. With different approaches employed in each model, the climate impact of CCT, including any unintended side effects, remains uncertain, which highlights the need for a consistent, physically-based approach to simulating the complex microphysical processes governing ice formation and growth in cirrus clouds (Gasparini et al., 2020). In this study, we investigate the climate impact of CCT using a new ice microphysics scheme

in the ECHAM-HAM GCM that includes a prognostic treatment of ice sedimentation by introducing a single ice category, and an updated approach for calculating ice cloud fractions that allows for fractional cirrus gridbox coverage (Section 2). We perform CCT simulations using a cirrus ice nucleation scheme that accounts for the competition between homogeneous and heterogeneous nucleation, and depositional growth onto pre-existing ice particles (Section 2). Additional ice source number and mass mixing ratio tracers are implemented to directly investigate the impacts of seeding on the competition between the different ice nucleation modes. Results are presented in Section 3, followed by a discussion of our findings in Section 4. We present our conclusions in Section 5.

## 2  Methods

### 2.1  Model Description

We conduct our seeding experiments using the ECHAM6.3-HAM2.3 aerosol-climate GCM (Stier et al., 2005; Zhang et al., 2012; Stevens et al., 2013; Neubauer et al., 2019; Tegen et al., 2019). We use the horizontal resolution T63 ($1.875^{\circ}$ x $1.875^{\circ}$), with 47 vertical levels (L47) up to $0.01\,\mathrm{hPa}$, which corresponds to a vertical resolution of around 1 km in the upper troposphere at cirrus altitudes. The model timestep is 7.5 minutes.

The two-moment ice microphysics scheme by Lohmann et al. (2007), used in the default version of ECHAM6.3-HAM2.3, was succeeded by the Predicted bulk Particle Properties (P3) scheme by Morrison and Milbrandt (2015) that was ported to ECHAM-HAM by Dietlicher et al. (2018, 2019). It replaces an earlier method of artificially separating ice particles into different size classes (Levkov et al., 1992), rendering the use of the tuning parameter for the rate of snow formation unnecessary (Dietlicher et al., 2019). Instead, ice is represented with a single prognostic category based on mass-to-size relationships. With the single ice category no longer differentiating between in-cloud and precipitating ice, vertical advection and precipitation processes were also updated to include a substepping approach for prognostically solving ice sedimentation. This allows for sedimenting ice to be subjected to cloud processes as it falls, and for numerical stability within the cloud scheme (Dietlicher et al., 2018). For more specific information on P3 and its implementation within ECHAM6, please refer to Dietlicher et al. (2018, 2019).

A separate scheme by Kärcher et al. (2006) that was adapted for ECHAM-HAM by Kuebbeler et al. (2014) handles in-situ ice nucleation within cirrus clouds. It simulates the competition for water vapor between heterogeneous and homogeneous nucleation, and between depositional growth onto pre-existing ice particles from an existent cirrus clouds or those that are transported into the cirrus regime from deep convective detrainment, or from stratiform mixed-phase clouds. The scheme uses a sub-stepping approach to simulate the temporal evolution of ice saturation ratio ($S_i$) in an air parcel rising adiabatically during the formation-stage of a cirrus cloud. Ice formation occurs only when $S_i$ reaches the critical values for heterogeneous or homogeneous nucleation (see below). The evolution of $S_i$ is determined by the balance between the adiabatic cooling rate of rising air and the diffusional growth of ice particles that consume the available water vapor. As the cooling rate, and therefore the magnitude of $S_i$, is directly related to the strength of vertical velocity, a fictitious downdraft that counteracts the vertical velocity is introduced at the start of each timestep of the cirrus sub-model to quantify the effect of water vapor consumption

onto pre-existing ice particles, which includes new ice formation in the previous cirrus sub-model timestep (Kuebbeler et al., 2014). This "effective vertical velocity" (updraft + fictitious downdraft), therefore, determines the magnitude of $S_i$, and is

calculated at the end of a single sub-timestep of the cirrus scheme. It is used in the subsequent sub-timestep to update $S_i$.

Vertical velocity is represented by a grid-mean value plus a turbulent component based on the turbulent kinetic energy (TKE), (Brinkop and Roeckner, 1995; Kuebbeler et al., 2014). Orographic effects on vertical velocity as well as small-scale gravity waves (Kärcher et al., 2006; Joos et al., 2008, 2010; Jensen et al., 2016a) in the upper troposphere are not included in this study. We provide a short analysis that verifies our model without orographic effects in Appendix A. In summary, by using the new

P3 ice microphysics with the in-situ cirrus ice nucleation scheme (Muench and Lohmann, 2020), including orographic effects acts to drastically increase cirrus ICNC while reducing spatial heterogeneity, in worse agreement with observations. Muench and Lohmann (2020) updated the water vapor consumption by ice, following the diffusional growth equation (Lohmann et al., 2016). The temporal change of the saturation ratio follows such that if the updraft is stronger than the water vapor consumption by pre-exisiting ice and heterogeneous INPs, then it may reach a suitable magnitude for homogeneous nucleation to occur. The

opposite is true in weaker updraft regimes or in high INP concentration environments (Kärcher et al., 2006). The sub-stepping approach in the cirrus scheme is computed dynamically based on a $1\,\%$ rate of change of the ice saturation ratio between each sub-timestep.

To simulate the competition between homogeneous and heterogeneous nucleation, several freezing modes are introduced into the cirrus scheme (Table 1), including pre-existing ice. In general, the cirrus nucleation scheme follows an "energy-barrier"

approach, with pre-existing ice and the most efficient INP, dust (in the default setup), consuming water vapor at a lower $S_i$. An ice formation event in each mode can occur as either a threshold freezing process or as a continuous freezing process (Muench and Lohmann, 2020). The former is based on the original cirrus scheme by Kärcher et al. (2006), whereby ice forms by a particular mode when its critical ice saturation ratio ($S_{i,crit}$) is reached. In our setup, homogeneous nucleation of liquid-sulphate aerosols with a temperature-dependent $S_{i,crit}$ between 1.4 and roughly 1.75 (Koop et al., 2000), and immersion freezing of

soluble material coated dust with a $S_{i,crit}$ of 1.3, act as threshold freezing modes. As a threshold process, all aerosol particles associated with the mode form ice that proceeds to deplete available water vapor and reduce $S_i$. For dust immersion freezing, only $5\,\%$ of the total dust aerosol concentration from the aerosol module, HAM, act as INPs within the mode, following Gasparini and Lohmann (2016). Muench and Lohmann (2020) introduced the latter, continuous freezing process to account for the saturation-dependent activated fraction (AF) of INPs available for heterogeneous nucleation. We include deposition

on insoluble accumulation and coarse size mode (Stier et al., 2005; Zhang et al., 2012; Tegen et al., 2019) dust particles as continuous freezing modes. The AF is calculated using a temperature-dependent $S_{i,crit}$ threshold of 1.2 for $T > 220\,K$, and 1.1 for $T \leq 220\,K$ based on laboratory measurements by Möhler et al. (2006). At every timestep in the cirrus scheme, the AF of these modes is calculated, and if ice forms it is added to the ice concentration.

Following Gasparini and Lohmann (2016), we introduce seeding particles as a separate threshold freezing mode into the

cirrus scheme for temperatures below $238\,K$, increasing the competition for available water vapor. All seeding particles can nucleate ice with a $S_{i,crit}$ (hereafter seeding particle critical saturation ratio ($S_{i,seed}$)) of 1.05 (Storelvmo and Herger, 2014), and later with $S_{i,seed} = 1.35$ (Section 2.2). The seeding particles have a modal radius of $0.5\,\mu m$ like in Gasparini and Lohmann

**Table 1.** Summary of the different aerosol species available for in-situ ice nucleation within the cirrus scheme, including information on the average radius of the particles, the critical ice saturation ratio above which these particles will nucleate ice, the freezing mechanism by which nucleation will occur, and the freezing method within the context of the cirrus scheme following Muench and Lohmann (2020).

| Particle type | Mean radius (μm) | Critical $S_i$ | Freezing mechanism | Freezing method |
|---|---|---|---|---|
| Insoluble dust | 0.05 to 0.5 | Temperature-dependent, but > 1.1 | Deposition nucleation | Continuous |
| | > 0.5 | Temperature-dependent, but > 1.2 | | |
| Soluble dust | > 0.05 | 1.3 | Immersion freezing | Threshold |
| Aqueous sulfate | All size modes: < 0.005 to > 0.5 | 1.4 | Homogeneous nucleation | Threshold |

(2016). We perform uniform global seeding with no spatial or temporal variability in seeding particle concentration for comparability with previous GCM studies, except for an altitude restriction below $100\,\mathrm{hPa}$ to minimize seeding of the stratosphere.
This seeding restriction to altitude levels below $100\,\mathrm{hPa}$ (i.e. higher pressure levels) is in line with proposed real-world delivery mechanisms for seeding particles with commercial aircraft (Mitchell and Finnegan, 2009).

Cloud cover is based on the diagnostic approach by Sundqvist et al. (1989), hereafter referred to as S89, that assumes fractional cloud formation exists due to relative humidity (RH) variability within the gridbox. The formulation was developed for liquid (warm) clouds, using a critical RH ($\mathrm{RH_{crit}}$) above which fractional cloud cover in a gridbox can occur. Full grid-box
coverage occurs when grid-mean RH reaches $100\,\%$ with respect to liquid water. This formulation works well for warm clouds, but as Kuebbeler et al. (2014) and Dietlicher et al. (2018, 2019) note, it breaks down for mixed-phase clouds (T < 273 K) that may or may not include ice, presenting a difficult choice between RH with respect to liquid ($\mathrm{RH_l}$) or ice ($\mathrm{RH_i}$) to determine cloud fraction. The S89 approach for pure ice clouds (T < 235 K) is analogous to warm clouds in earlier versions of our model, where instead of liquid water saturation, full gridbox coverage occurs at ice saturation. As Kuebbeler et al. (2014) explain,
when accounting for the ice supersaturation required for homogeneous or heterogeneous nucleation, this leads to full gridbox coverage of freshly nucleated cirrus clouds, an inconsistency between cloud fraction and the microphysics scheme (Kärcher and Burkhardt, 2008). This also may explain the high cirrus CRE in ECHAM6 found by Gasparini and Lohmann (2016). Dietlicher et al. (2019) updated the cloud fraction formulation for pure ice clouds to differ from liquid clouds by updating the RH conditions in which an ice cloud can partially cover a gridbox. In this new scheme (hereafter, D19) that we use in this study,
ice saturation ($S_i$ = 1.0) is set as the lower boundary condition for partial ice cloud fractions. The upper boundary condition for full gridbox coverage for ice clouds is set following the theory for homogeneous nucleation of solution droplets by Koop et al. (2000). The difference between the two schemes is illustrated in Figure 1. As a contextual example, if ice were to form at 233 K in an environment with $S_i$ = 1.2, then D19 would calculate an ice cloud fraction <1.0, whereas S89 would adjust the ice supersaturation down to ice saturation and would produce a cloud fraction of 1.0.
Additional ice number and mass mixing ratio tracers were added to the model, following Dietlicher et al. (2019), to trace the origin of in-situ cirrus ice directly. We include two tracers for ice from homogeneous and heterogeneous nucleation, with

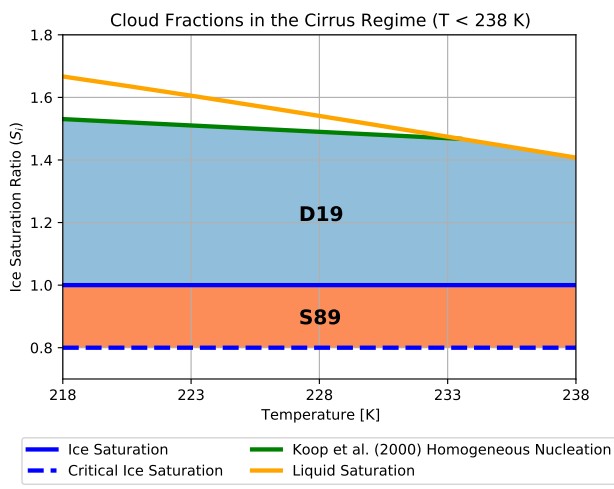

**Figure 1.** Cloud fraction schematic adapted from Dietlicher et al. (2019) showing the difference between the D19 and S89 approaches for calculating ice cloud fractions. The shaded areas show the temperature versus ice saturation ratio conditions where clouds can form, with the orange area for S89 and the blue area for D19. The blue line is the ice saturation line ($S_i = 1.0$), the blue dotted line is the critical ice saturation line for cloud formation in the S89 approach, the orange line is for liquid saturation with respect to ice saturation, and the green line is the homogeneous nucleation limit according to Koop et al. (2000).

additional tracers for heterogeneously-formed ice on dust and seeding particles, the sum of which equates to the total in-situ heterogeneously-nucleated ice tracer. The implementation of these tracers highlighted an error when accounting for the number of aerosols that previously nucleated ice. The aerosol concentration of each freezing mode of the cirrus scheme was scaled by

the total amount of pre-existing ice. This approach overestimated the concentrations of in-cloud aerosols and underestimated the interstitial aerosol concentration. We updated the scaling of each aerosol concentration mode to account for the fraction of each mode out of the total pre-existing ice concentration. These updates warranted a re-tuning of the model to primarily target the balance between global annual mean TOA SW and LW fluxes (Lohmann and Ferrachat, 2010; Mauritsen et al., 2012; Neubauer et al., 2019). A summary of the model configuration we utilize in this study compared to the "HET_CIR"

simulation by Dietlicher et al. (2019) is presented in Table 2. Ice self-collection ($\gamma_{islf}$) was increased from its original value in the base version of the model (Dietlicher et al., 2019) to 5.5 to account for too small TOA SW and LW fluxes. This adjustment strengthened both TOA fluxes, but the LW flux remained too weak. Therefore, to compensate, the auto-conversion rate from cloud liquid water to rain within convective cores was increased to 1.75. In addition to radiative flux imbalance, we found that the model produced a global mean liquid water path (LWP) value that was beyond the upper value of the observations reported

by Neubauer et al. (2019). To address this issue we halved the convective cloud mass-flux above the level of non-buoyancy ($\gamma_{ctop}$) to 0.1. As reducing this flux leads to more frequent and thicker boundary layer clouds (Mauritsen et al., 2012), we compensated this by increasing the autoconversion rate within stratiform liquid clouds ($\gamma_r$) to 8.3 to maintain radiative balance. All other tuning parameters were kept the same as the "HET_CIR" configuration in Dietlicher et al. (2019). We also note a too

**Table 2.** Model configuration comparison between the "HET_CIR" simulation by Dietlicher et al. (2019) and our "Full_D19" reference simulation presented in this study. The tuning parameters include: ice self-collection ($\gamma_{slf}$), the autoconversion rate from cloud liquid water to rain within convective cores ($\gamma_{cpr}$), the convective cloud mass-flux above the level on non-buoyancy ($\gamma_{ctop}$), and the autoconversion rate within stratiform liquid clouds ($\gamma_r$).

| Parameter | HET_CIR | Full_D19 |
|-----------|---------|----------|
| $\gamma_{islf}$ | 3.0 | 5.5 |
| $\gamma_{cpr}$ | $1.5 \times 10^{-4}$ | $1.75 \times 10^{-4}$ |
| $\gamma_{ctop}$ | 0.2 | 0.1 |
| $\gamma_r$ | 4.4 | 8.3 |

negative net CRE after tuning. Dietlicher et al. (2019) state this points to a possible structural problem within the model, which
is related to the coarse vertical resolution that results in the under-prediction of low-level clouds (Pelucchi et al., 2021).

## 2.2 Experimental Setup

We performed cirrus seeding simulations using P3 with the cirrus scheme coupled to the new ice-cloud fraction approach (D19) described above. We examined seeding with full nucleation competition between heterogeneous, homogeneous, and pre-existing ice. Additionally, we tested the original S89 ice-cloud fraction approach (Stevens et al., 2013; Neubauer et al.,
2014, 2019) within the framework of the P3 scheme; we did not re-tune the model for simulations using S89 in order to examine the sensitivity of cirrus seeding to the ice cloud fraction scheme. Previous CCT studies include additional simulations in which they allow only homogeneous nucleation to occur in cirrus. Here, we chose to pursue full nucleation competition as a more realistic approach to examine the impact of seeding particles, mimicking a real-world implementation. For both model configurations (see Table 3) we implemented seeding particles as an additional heterogeneous freezing mode in the
cirrus ice-nucleation scheme continuously at every timestep, following on from previous approaches (i.e. without accounting for those that already formed ice). Only gridboxes that are supersaturated with respect to ice (i.e. $S_i > 1.0$) are seeded. We test four seeding INP concentrations of 0.1, 1, 10, and 100 INP $L^{-1}$ to represent the spread of concentrations tested in previous studies (Storelvmo and Herger, 2014; Penner et al., 2015; Gasparini and Lohmann, 2016). Each simulation was conducted for five years between 2008 and 2012, inclusive, with three months of spin-up from 1st October 2007. Monthly mean sea surface
temperatures and sea ice coverage are prescribed, and emissions are from the year 2010 following CMIP6 methodology (van Marle et al., 2017; Hoesly et al., 2018; McDuffie et al., 2020).

The $S_{i,seed}$ of 1.05 follows Storelvmo and Herger (2014) and Gasparini and Lohmann (2016), and is based on suggestions of a hypothetical, highly-efficient seeding particle material. However, it is unclear whether this $S_{i,seed}$ can be applied to a realistic seeding particle material. Mitchell and Finnegan (2009) suggested bismuth tri-iodide, but the specific ice nucleating properties
of this material are unknown. Therefore, to test the sensitivity of ice nucleation competition to $S_{i,seed}$, we conducted additional

**Table 3.** Experimental setup for cirrus seeding for the two ice-cloud fraction schemes. Both configurations include seeding particle concentrations of 0.1, 1, 10, and 100 $L^{-1}$. In addition, seeding is conducted for a seeding particle critical ice saturation ratio ($S_i$) of 1.05 and 1.35. The "Full" in the reference simualtions refers to full ice nucleation competition between pre-existing ice, heterogeneous nucleation on minereal dust particles, and homogeneous nucleation of liquid sulfate aerosols in the in-situ cirrus scheme (Kärcher et al., 2006; Kuebbeler et al., 2014).

| Ice-cloud fraction scheme | Description | Reference simulation | $S_{i,seed} = 1.05$ | $S_{i,seed} = 1.35$ |
|---|---|---|---|---|
| D19 | new cloud fraction by Dietlicher et al. (2018, 2019) | Full_D19 | Seed0.1 | Seed0.1_1.35 |
| | | | Seed1 | Seed1_1.35 |
| S89 | original cloud fraction by Sundqvist et al. (1989) | Full_S89 | Seed10 | Seed10_1.35 |
| | | | Seed100 | Seed100_1.35 |

seeding simulations with all seeding particle concentrations described above, with a $S_{i,seed}$ of 1.35 (Table 3). We chose this relatively high $S_{i,seed}$ value to ensure that seeding can occur in ice supersaturated environments below the lower homogeneous nucleation $S_{i,crit}$ threshold roughly $\geq 1.40$ and, in order to be less competitive with background heterogeneous nucleation processes, above the maximum $S_{i,crit}$ for dust of 1.3.

### 2.3 Uncertainty

We take particular care to quantify significance in our results, following the "false discovery rate (FDR)" method by Wilks (2016). The updated approach for conducting independent t-tests accounts for high spatial correlation of neighboring gridpoints, i.e. the null hypothesis cannot be as widely rejected when calculating significance. We calculate a $5\,\%$ significance based on the inter-annual variability over the five years of simulation (Section 2.2). The inter-annual variability is also used to calculate the $95\,\%$ confidence interval around the five-year mean.

### 3 Results

### 3.1 Model Validation

We start by evaluating the model with the new P3 ice microphysics scheme and the new D19 ice-cloud fraction approach for the unseeded reference case, by comparing ICNC data to the latest compilation of in-situ aircraft measurements by Krämer et al. (2016, 2020) in Figure 2. Model results represent the five-year mean temperature versus ICNC between 2008 and 2012. The observational data comprise multiple in-situ aircraft field campaigns between 1999 and 2017, totalling around 90 hours of flight data (Krämer et al., 2020), with different meteorological situations captured in the tropics, mid-latitudes, and the Arctic;

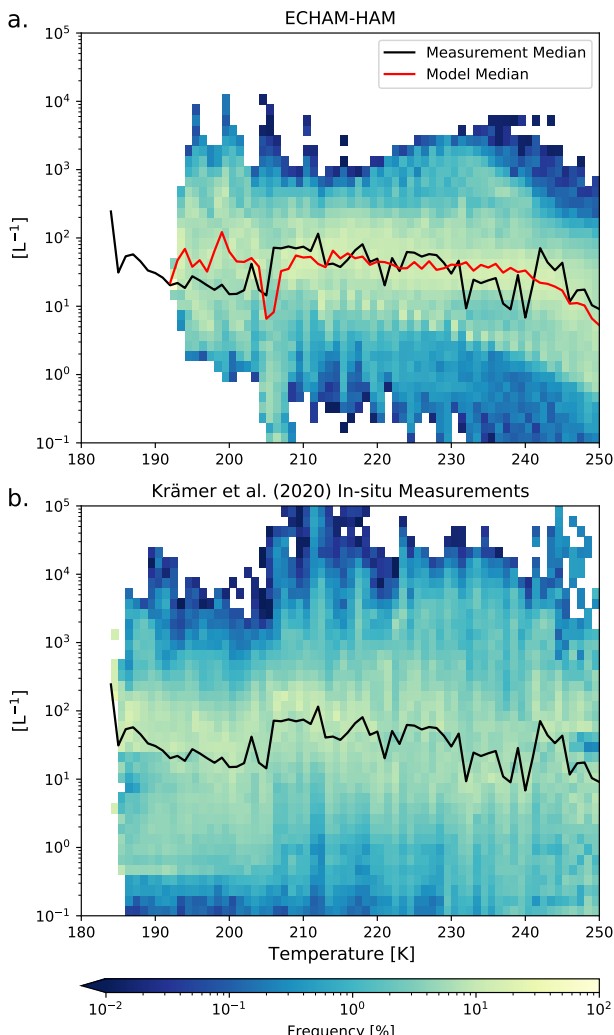

**Figure 2.** ICNC (in $L^{-1}$) frequency diagrams for ice crystals with a diameter of at least $3\,\mu m$ as a function of temperature between $180\,K$ and $250\,K$ binned like in Krämer et al. (2020) for every $1\,K$. The five-year global mean data from the model is plotted in (a) and the compilation of in-situ flight data from Krämer et al. (2020) is plotted in (b). The red line in the upper plot represents the binned median ICNC value of the model data, and the black line in both plots is the same value for the observational data.

southern high latitudes are not included. Although this is a much more significant compilation of observational data than was previously available, there remains a caveat that these data are not representative of the entire atmosphere (Krämer et al., 2020).

The median ICNC per temperature bin between $180\,K$ and $250\,K$ is also shown for both data sets, with the observational median also presented with the model data for comparison. Model-median ICNC values agree rather well with the observational median at temperatures between roughly $205\,K$ and $230\,K$. Between $230\,K$ and $240\,K$ the model-median diverges above the observational median where it does not capture the more frequent occurrence of lower ICNC values. Both the model and

observations capture the tailing-off of ICNC at temperatures warmer than $240\,\mathrm{K}$, with the model being slightly lower than the observations. The small disagreements in these two temperature ranges may be linked to the default parameterization for heterogeneous nucleation on mineral dust particles in mixed-phase clouds in ECHAM. The results by Villanueva et al. (2021) offer an explanation in this regard. In their study, they conducted several sensitivity tests with ECHAM-HAM using the default rate-based immersion freezing scheme by Lohmann and Diehl (2006) and a newer AF approach based on dust particle surface area and active site density. They found better agreement with satellite-based observations using the AF approach in combination with higher dust particle freezing efficiency as compared to the default rate-based approach, and noted an under-prediction of mixed-phase ice with the latter that led to a higher abundance of cloud droplets being transported into the cirrus regime where they could undergo homogeneous nucleation. Our model median ICNC values between $230\,\mathrm{K}$ and $250\,\mathrm{K}$ indicate a similar behavior. The higher ICNC values between $230\,\mathrm{K}$ and $240\,\mathrm{K}$, as compared to the observations, are likely of liquid-origin. Whereas, the lower ICNC values above $240\,\mathrm{K}$ are likely due to the under-prediction of mixed-phase ice using the default rate-based scheme for dust immersion freezing. The Villanueva et al. (2021) study suggests using a different approach for mixed-phase cloud glaciation for better comparability to observations and to address this issue of an over-abundance of liquid-origin cirrus ice. Krämer et al. (2020) suggest that these liquid-origin cirrus in the mid-latitudes originate from warm conveyor belts or mesoscale convective systems. Therefore, their formation is tied to a stronger dynamical forcing that allows for abundant homogeneous nucleation from numerous cloud droplets being transported into the cirrus regime. As CCT targets in-situ formed cirrus in regions with less dynamical forcing (Gasparini et al., 2017), we deem this over-prediction of ICNC values insignificant relative to our study.

The model diverges from the observed median at temperatures below about $205\,\mathrm{K}$. According to Krämer et al. (2020) ICNC values at such cold temperatures likely originate from tropical deep convection. Between $195\,\mathrm{K}$ and $205\,\mathrm{K}$ the model median ICNC is higher than the in-situ measurements. This may be linked to a lack of cloud-top measurements at these cold temperatures, or the fact that high ICNC values in this temperature range are short-lived and therefore difficult to capture by aircraft (Gryspeerdt et al., 2018; Krämer et al., 2020). The model also does not capture the ICNC occurrence at temperatures below roughy $195\,\mathrm{K}$. A simple analysis on the number of data points belonging to this temperature regime shows that in the observations, there is a large drop-off in the number of recorded points (not shown). Therefore, these measurements make up a small portion of the total observational dataset. Furthermore, CCT in a real-world context would target in-situ formed cirrus away from systems with strong dynamical forcing (Gasparini et al., 2017), like in the tropics. The model also does not capture the wide variability of ICNC values as seen in the in-situ measurements, like the higher frequency of low ICNC values between roughly $205\,\mathrm{K}$ and $250\,\mathrm{K}$. This is due to the fact that we compare five-year annual mean model data to instantaneous values recorded during various aircraft campaigns. However, for the purposes of our CCT analysis we find that the model median ICNC as a function of temperature agrees well with the Krämer et al. (2020) measurements for in-situ formed cirrus.

## 3.2 D19 versus S89 seeding

The net global-mean radiative balance between TOA SW and TOA LW fluxes, including the net CRE is presented in Figure 3a and c, respectively, for $S_{i,seed} = 1.05$. The results are tabulated along with the constituent SW and LW CRE fluxes in Table 4,

**Table 4.** Five-year annual global mean net top-of-atmosphere total radiative balance (TOA) and net CRE as well as SW CRE and LW CRE in $\mathrm{Wm}^{-2}$ for D19 and S89 ice-cloud cloud fraction approaches for seeding with $S_{i,seed} = 1.05$. Each quantity includes the $95\%$ confidence interval equating to two standard deviations of the mean values of the five-year data.

| Seeding Concentration [$\mathrm{L}^{-1}$] | | 0.1 | 1 | 10 | 100 |
|---|---|---|---|---|---|
| D19 | net TOA | $0.06 \pm 0.40$ | $-0.17 \pm 0.37$ | $0.61 \pm 0.35$ | $4.88 \pm 0.43$ |
| | net CRE | $0.08 \pm 0.39$ | $0.02 \pm 0.38$ | $0.89 \pm 0.37$ | $4.13 \pm 0.39$ |
| | SW CRE | $0.11 \pm 0.36$ | $-0.06 \pm 0.36$ | $-0.66 \pm 0.30$ | $-3.30 \pm 0.35$ |
| | LW CRE | $-0.03 \pm 0.16$ | $0.07 \pm 0.16$ | $1.54 \pm 0.15$ | $7.42 \pm 0.18$ |
| S89 | net TOA | $0.14 \pm 0.38$ | $0.45 \pm 0.39$ | $2.66 \pm 0.31$ | $9.88 \pm 0.32$ |
| | net CRE | $0.17 \pm 0.34$ | $0.56 \pm 0.40$ | $3.20 \pm 0.36$ | $8.85 \pm 0.35$ |
| | SW CRE | $0.09 \pm 0.34$ | $-0.13 \pm 0.35$ | $-1.61 \pm 0.40$ | $-5.94 \pm 0.33$ |
| | LW CRE | $0.08 \pm 0.18$ | $0.69 \pm 0.10$ | $4.80 \pm 0.12$ | $14.79 \pm 0.20$ |

with the $95\%$ confidence interval; CRE fluxes are discussed below. We find no net negative mean TOA anomalies for any of our simulations except Full_D19 Seed1 (Table 4). Some cooling may be evident within the range of uncertainty surrounding the mean anomalies for the Seed0.1 (D19 and S89) and Seed1 (D19 only) simulations. However, as the uncertainty is high relative to the mean, a clear response at these low seeding particle concentrations is unclear from a TOA perspective. For larger seeding concentrations ($\geq 10\,\mathrm{L}^{-1}$), the radiative anomalies indicate a certain warming response likely from overseeding. Furthermore, the differences between the two cloud cover approaches become abundantly clear. The largest warming occurs for Seed100, with $4.9\,\mathrm{Wm}^{-2}$ (D19) and $9.9\,\mathrm{Wm}^{-2}$ (S89). These responses are an order of magnitude larger than the maximum TOA anomaly found by Gasparini and Lohmann (2016) of $0.5\,\mathrm{Wm}^{-2}$ at the same seeding particle concentration and for a similar configuration of the cirrus scheme. Instead, our results more closely resemble their simulations where seeding was applied to cirrus that could form only by homogeneous nucleation, but are more than two times what they found at a seeding concentration of $100\,\mathrm{L}^{-1}$. This difference in results further highlights the importance of a consistent approach to simulate cirrus ice microphysics (Gasparini et al., 2020), and will be discussed further in Section 4. In addition, the maximum responses shown here are well above the latest available IPCC estimate of the effective radiative forcing from a doubling of atmospheric $CO_2$ from the pre-industrial period of $3.9\,\mathrm{Wm}^{-2}$ (Forster et al., 2021), highlighting the potential dangerous side-effects of cirrus seeding.

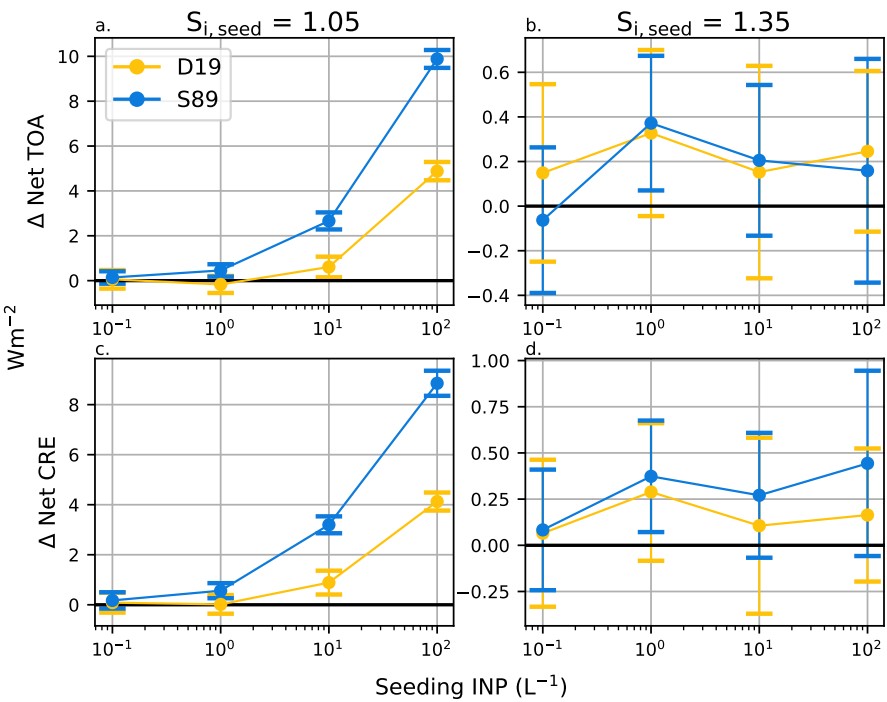

**Figure 3.** Five-year mean global mean net top-of-atmosphere (TOA) radiative balance anomalies in $\mathrm{Wm}^{-2}$ between total SW and longwave fluxes, and cloud radiative fluxes comprising the CRE. Anomalies are defined as the differences between each seeding simulation and the reference simulation without seeding. The left column (a,c) shows the radiative anomalies for simulations with $S_{i,seed} = 1.05$, and the right column (b,d) is the same for $S_{i,seed} = 1.35$. The errors bars represent the $95\%$ confidence ($2\sigma$). Note the differences in scales for the $S_{i,seed} = 1.05$ plots and the $S_{i,seed} = 1.35$ plots.

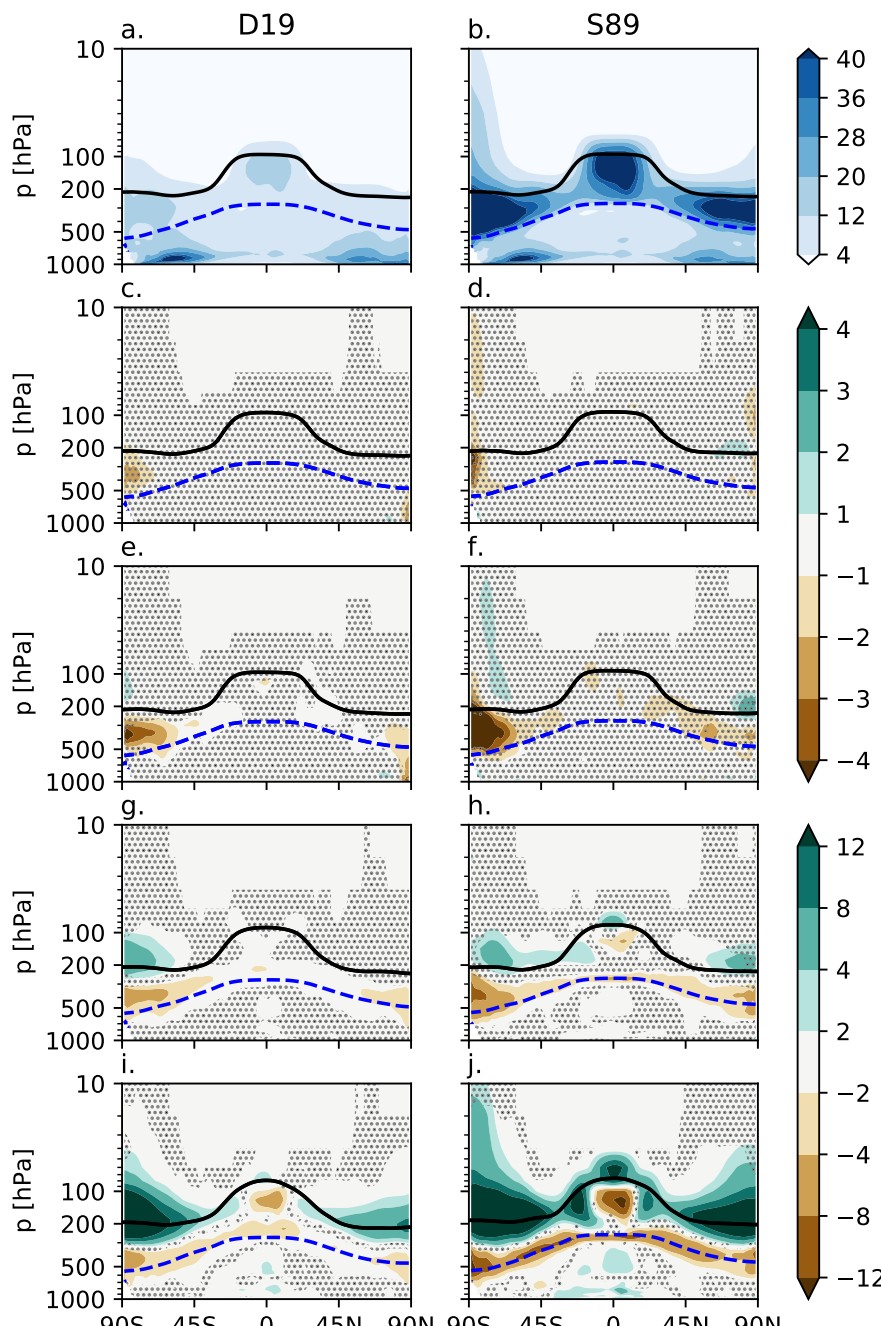

**Figure 4.** Five-year zonal mean cloud fractions [%] on pressure levels [hPa] for D19 and S89 ice-cloud fraction approaches for the unseeded reference cases (a-b). The cloud fraction anomalies respective to the unseeded reference cases are plotted in the subsequent rows for $S_{i,seed}$ = 1.05: Seed0.1 (c-d), Seed1 (e-f), Seed10 (g-h), and Seed100 (i-j). The black line is the five-year mean zonal mean WMO-defined tropopause height on pressure levels, and the blue dashed line is the 238 K isotherm. The stippling in the difference plots shows insignificant data points on the 95 % confidence level according to the independent t-test controlled by the "false discovery rate" method.

Cloud effects are the largest contributor to the TOA radiative anomalies (Figure 3c and Table 4). In the Seed100 case, the net CRE anomalies make up roughly $85\,\%$ and $90\,\%$ of the total TOA radiative anomalies for both D19 and S89. Like the TOA anomalies, there is slight evidence of cooling at lower seeding particle concentrations ($< 10\,\mathrm{L}^{-1}$) within the range of uncertainty. At higher concentrations, it is clear that clouds exert a positive forcing on the atmosphere, which is fuelled by positive LW CRE anomalies (Table 4). These large anomalies are only partially counteracted by increasingly negative SW CRE anomalies at higher seeding particle concentrations, indicating perhaps a shift in cirrus formation pathway towards optically thicker liquid origin cirrus (Krämer et al., 2020) or a feedback on lower-lying liquid and mixed-phase clouds.

To examine the cloud impacts further, in particular the overseeding at high seeding particle concentrations, we show the zonal mean cloud fraction anomalies between each seeding simulation and their respective reference simulation for both cloud fraction schemes in Figure 4. Firstly, the difference between D19 and S89 stands out from the respective reference simulations (top panels). With the larger $S_i$ bounds for ice cloud fractions in D19, there is a clear cloud fraction reduction within the cirrus regime, above the blue dotted line (238 K isotherm) in Figure 4, which leads to less warming in the reference simulation compared to S89. The new cloud fractions in D19 were found to agree better with the observed satellite product from CALIPSO than the original S89 approach (Dietlicher et al., 2019). Secondly, a significant pattern in the zonal cloud fraction does not emerge until Seed1, with small regions of cirrus cloud fraction reductions larger than about $4\,\%$. S89 Seed1 shows a small region of positive cloud fraction anomaly in the stratosphere over the northern high latitudes, however the signal is not clear as all anomalies are insignificant. A clearer pattern emerges for Seed10 and Seed100, where what appears as a shift in cloud height starts developing within the cirrus regime at these seeding concentrations and reaches a maximum for Seed100. Seeding decreases cloud fraction by up to $8\,\%$ and $12\,\%$ in D19 and S89 respectively in the mid-troposphere between $300\,\mathrm{hPa}$ and $800\,\mathrm{hPa}$ at higher latitudes, and between $300\,\mathrm{hPa}$ and $100\,\mathrm{hPa}$ in the tropics. Note that the tropopause is located at roughly $200\,\mathrm{hPa}$ in polar regions and at $100\,\mathrm{hPa}$ in the tropics, as shown by the black line in Figure 4. There are noticeable cloud fraction increases around the tropopause by more than $12\,\%$ over the southern high latitudes for D19 and over all latitudes for S89. The difference between the two cloud fraction approaches in this case is also clear, with S89 showing much more extensive regions of cirrus cloud fraction increases in the stratosphere than D19. The difference between the cloud fraction approaches is discussed further in Section 4. There are small regions in the lower tropical to mid-latitude troposphere (pressure $> 500\,\mathrm{hPa}$) that show positive cloud fraction anomalies up to $4\,\%$ and $8\,\%$ for D19 and S89 respectively. The reduction of lower-lying cirrus and an apparent shift to more frequent higher altitude cirrus explains the large positive LW CRE anomalies in Table 4. This shift outweighs the stronger (i.e. more negative) SW CRE anomalies that likely originate from the small positive cloud fraction anomalies for lower-lying clouds. Meanwhile, the overseeding response is amplified by the unrealistic increases of cloud fraction in the stratosphere.

Next, we examine the microphysical response to seeding in Figure 5, which shows the total ICNC anomalies for Seed1 (a-b) and Seed100 (c-d) for both D19 and S89. Determining an exact ICNC response for Seed1 is rather difficult due to ICNC anomaly heterogeneity. For D19 Seed1 (Figure 5a), in some areas we find that seeding produces more ice particles in widespread areas throughout the troposphere, with few areas of negative anomalies that also extend into the lower stratosphere. The S89 signal is similar, being mixed throughout the tropopause and extending into the lower stratosphere, but is less pro-

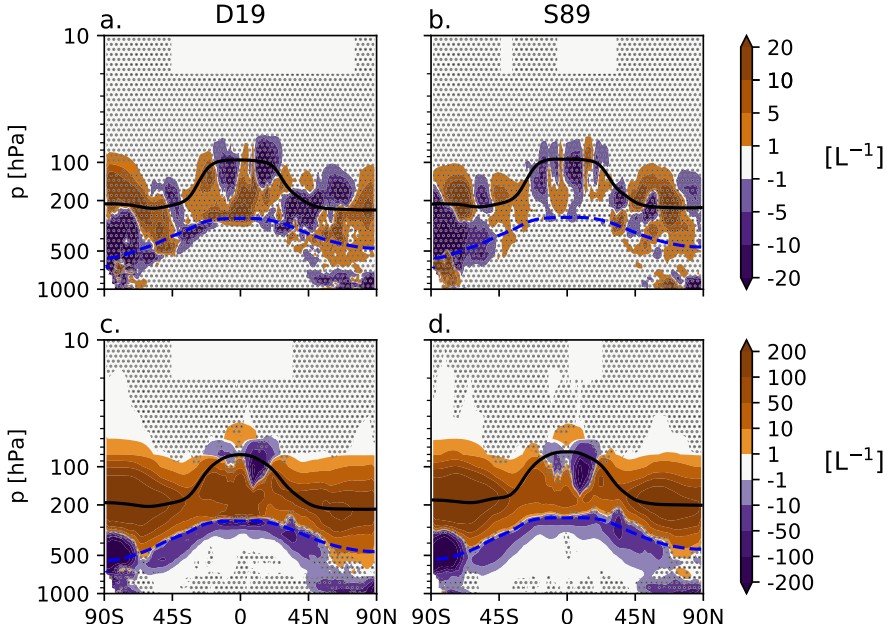

**Figure 5.** Five-year zonal mean ICNC $S_{i,seed} = 1.05$ anomalies in $L^{-1}$ for both D19 and S89 ice-cloud fraction approaches. Seed1 anomalies are presented in a and b, and Seed100 anomalies are presented in c and d. The black line is the five-year mean zonal mean WMO-defined tropopause height on pressure levels, and the blue dashed line is the $238\,\mathrm{K}$ isotherm. The stippling in the difference plots shows insignificant data points on the $95\,\%$ confidence level according to the independent t-test controlled by the "false discovery rate" method.

nounced than D19. For both cases, the positive ICNC anomalies at lower altitudes, in some regards, are in line with one of the desired outcomes of CCT to produce ice at lower altitudes, i.e. warmer temperatures, which emits more LW than higher-altitude ice, thus inducing a cooling effect. However, for both our Seed1 cases, our FDR analysis (Wilks, 2016) reveals that the Seed1 ICNC anomalies contain high uncertainty. As a result, the net CRE (Figure 3 and Table 4) also shows high magnitude of uncertainty relative to the mean response.

The ICNC anomalies are much clearer and certain for the extreme case, Seed100, than for the Seed1 anomalies (Figure 5c-d). Positive ICNC anomalies exceeding $200\,L^{-1}$ are shown at all latitudes throughout the troposphere, and into the lower stratosphere at higher latitudes. The anomaly heterogeneity around the tropics is likely due to the proficiency of seeding particles to nucleate ice and hamper homogeneous nucleation in convective outflow regions around the tropopause. The ICNC anomalies at lower altitudes and towards higher latitudes are much clearer. Here the ICNC anomalies are in line with the cloud fraction anomalies in Figure 4. There is a loss of the lowermost ice crystals that also extends into the mixed-phase regime (below the blue dashed line in Figure 5), while the ICNC in the cirrus regime increases. This is likely due to the proficiency of seeding particles to nucleate ice, leading to more numerous and smaller ice crystals that do not sediment into the mixed phase regime as readily compared to the unseeded case. In fact, we find that ice crystals decrease in size on average by more than

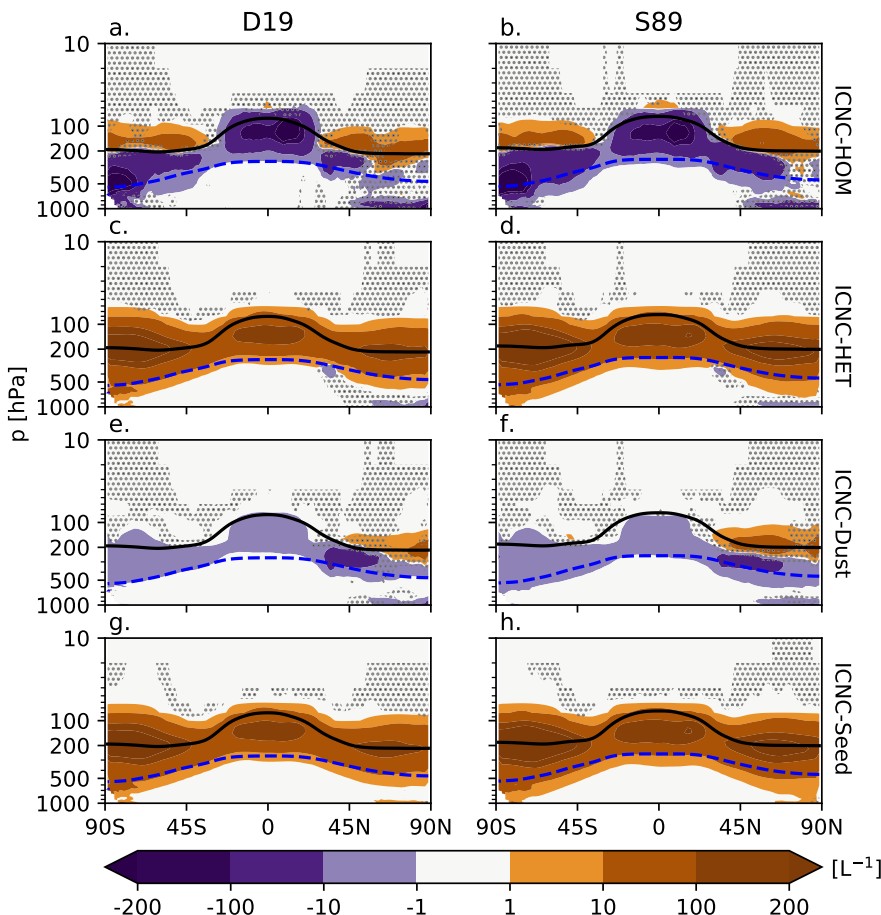

**Figure 6.** Five-year zonal mean in-situ cirrus ice number tracer anomalies in $L^{-1}$ between the simulation with 100 seeding INP $L^{-1}$ for $S_{i,seed} = 1.05$ and the respective unseeded reference case for both D19 and S89 ice-cloud fraction approaches. The anomalies include the in-situ homogeneously-nucleated ice number (a-b), the in-situ heterogeneous number (c-d), the heterogeneously-nucleated ice number formed on mineral dust particles (e-f), and the heterogeneously-nucleated ice number formed on seeding particles (g-h). The black line is the five-year mean zonal mean WMO-defined tropopause height on pressure levels, and the blue dashed line is the 238 K temperature contour. The stippling in the difference plots shows insignificant data points on the 95 % confidence level according to the independent t-test controlled by the "false discovery rate" method.

4.0 μm in the cirrus regime for Seed100 (not shown). In addition, with numerous seeding particles available up to 100 hPa, ICNC increases in the lower stratosphere above higher latitudes. This leads to large cloud fraction increases (Figure 4i-j) in these regions, where in the unseeded case there were fewer clouds (Figure 4a-b). Therefore, for the Seed100 case it is the combination of multiple effects that contributes to the strengthening of the LW CRE by roughly $7.4\,\mathrm{Wm}^{-2}$ and $14.8\,\mathrm{Wm}^{-2}$ in D19 and S89 (Table 4), respectively, and the strong positive Seed100 net TOA anomalies for both cases in Figure 3.

420

It is clear that seeding particles lead to an overseeding effect at higher concentrations, with wide impacts on the total ICNC. For a direct view on the impact of seeding particles on ice nucleation competition, Figure 6 shows the cirrus ice number tracer (Section 2.1) anomalies for Seed100 for D19 and S89. The tracers include in-situ cirrus ice numbers from homogeneous and heterogeneous nucleation, with additional tracers for heterogeneously formed ice on mineral dust particles and seeding particles. Firstly, the anomalies presented in Figure 6 are mainly constrained to the cirrus regime, the area above the blue-dashed line, and the lower stratosphere, with some extension of anomalies into the lower-lying mixed-phase regime following ice crystal sedimentation. In terms of ice nucleation competition, Seed100 shows the desired effect by decreasing homogeneously-nucleated ice by more than $200\,\mathrm{L^{-1}}$ in the middle to upper troposphere in both D19 and S89. The opposite effect occurs in the stratosphere where homogeneously-nucleated ice increases. The shift of homogeneous nucleation to lower pressure levels (Figure 6a-b), is likely due to increased LW cloud-top cooling from thicker cirrus cloud following seeding (Possner et al., 2017). This also impacts heterogeneous nucleation on mineral dust particles in the lower stratosphere. As this latter process is not sufficient at consuming water vapor, homogeneous nucleation proceeds to form additional ice crystals. This cloud top cooling effect likely also explains the heterogeneity of the total ICNC anomaly around the tropical tropopause (Figure 5). As there is a clear separation between the troposphere and the stratosphere, these phenomena point to a complex impact on the stratospheric circulation, which we discuss in Section 3.5.

The reduction of homogeneous nucleation in the troposphere is outweighed by the wider-spread increases in heterogeneous nucleation globally throughout the middle to upper troposphere and into the lower stratosphere for both ice cloud fraction approaches, leading to the positive net TOA and CRE anomalies (Figure 3). For Seed100 the heterogeneous signal is clearly dominated by seeding particles that act to dampen natural processes, including heterogeneous nucleation on dust as well as homogeneous nucleation. While this effect occurs in both D19 and S89, the spatial extent of the ICNC responses is more widespread in the latter in line with the smaller $S_i$ bounds for calculating ice cloud fractions.

We also find ice crystals formed on seeding particles from the cirrus regime ending up in the mixed-phase regime (below the dashed line in Figure 6), pointing to potential impacts on lower-lying cloud layers from seeding. In fact, vertical profiles of IWC and LWC for D19 in Figure 7 confirm this behavior. The positive Seed100 IWC anomaly within the cirrus regime right of the vertical black line in Figure 7 is in line with the total ICNC (Figure 5) and cirrus ice tracer (Figure 6) anomalies. We also find that ice increases to a smaller extent in the upper portion of the mixed phase regime, also in line with the tracer anomalies above. The main impact of seeding appears as a reduction of IWC in wider areas of the mixed-phase and liquid regimes, the latter of which includes sedimenting ice that has not had sufficient time to melt. This is likely due to amplified ice residence times in the cirrus regime fuelled by smaller ice crystals that weaken the sedimentation flux. With less ice falling into the mixed-phase regime at lower altitudes, LWC anomalies responded positively (Figure 7b) due to less efficient riming and/or cloud droplet depletion via the Wegener–Bergeron–Findeisen (WBF) process. This is in line with the positive, albeit small, anomalies of lower-lying cloud fractions in the tropics and mid-latitudes in Figure 4. With a higher frequency of these lower clouds, the SW CRE strengthens by about $-3.3\,\mathrm{Wm^{-2}}$ for D19 Seed100 (Table 4). However, this is outweighed by the larger LW CRE positive anomaly of $7.4\,\mathrm{Wm^{-2}}$ due to optically thicker in-situ cirrus (Krämer et al., 2020). To a smaller extent, a similar pattern is reflected in the Seed10 IWC vertical anomaly, in line with the positive LW CRE (Table 4); however, the LWC

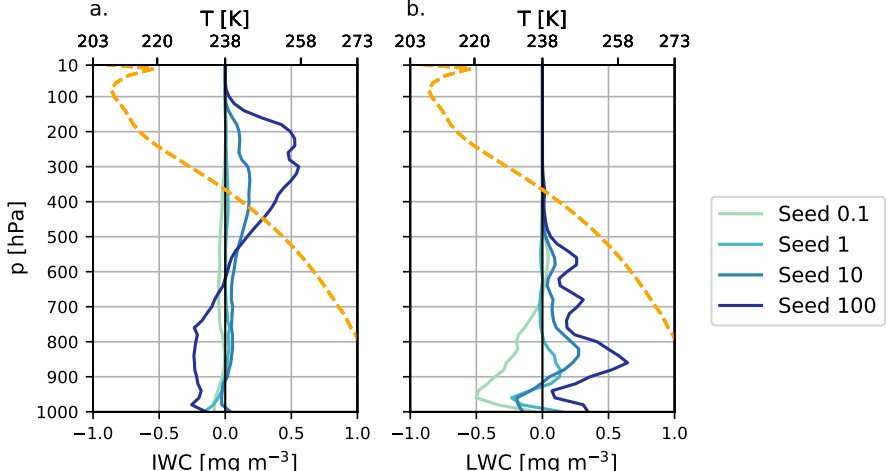

**Figure 7.** Five-year annual global mean (a) IWC and (b) liquid water content (LWC) vertical anomaly profiles in $\mathrm{mg\,m^{-3}}$ for D19 for all seeding particle concentrations for $\mathrm{S_{i,seed}} = 1.05$. The orange dotted line represents the five-year global mean temperature vertical profile centred around the homogeneous freezing temperature limit (238 K).

vertical anomaly is much less clear and therefore the SW CRE anomaly is much smaller with higher uncertainty relative to the mean value. It is unclear why LWC decreases by up to $0.5\,\mathrm{mg\,m^{-3}}$ in liquid clouds for D19 Seed0.1. This may point to an enhanced precipitation efficiency from the few seeding particles in this case that form ice that sediments into warmer regimes
and, thus, consumes available water and depletes lower-lying clouds. The SW CRE anomaly in Table 4 suggests this may be the case, however, as the uncertainty is high and there is no significant signal in the cloud fraction anomalies in Figure 4, it is unclear whether this feedback mechanism is present.

The patterns found in the TOA radiative anomalies, and the ICNC and IWC anomalies can be explained by the competition for water vapor during the formation of in-situ cirrus ice. The cirrus scheme is called during every time-step in the model, and
the nucleation of new ice crystals occurs only if cirrus conditions (T < 238 K) are met. Seeding particles efficiently form new ice crystals with a relatively low $\mathrm{S_{i,seed}} = 1.05$. In addition, our simplified method of including seeding particles as INPs in our cirrus scheme, using a globally uniform approach (i.e. every grid box includes the same concentration of seeding particles), results in accumulation of their impacts. This resulted in ICNC anomalies that were larger than the seeding particle concentration (Figure 5 and Figure 6). As seeding particles consume water vapor with increasing efficiency at higher concentrations, they
leave little supersaturated vapor left for other processes to occur, as indicated by the reduction of homogeneous nucleation and heterogeneous nucleation on mineral dust particles. This phenomenon goes beyond the traditional understanding of overseeding, where only homogeneous nucleation suppression was documented, coupled to a higher number of ice crystals nucleated on seeding particles (Storelvmo et al., 2013). Rather, our results show that overseeding leads to an ice nucleation competition alteration with the suppression of heterogeneous nucleation on mineral dust particles on top of homogeneous nucleation
suppression. In relation to the cloud fraction responses (Figure 4), overseeding in our model appears to lead to the desired

reduction of mid-troposphere clouds. However, at higher altitudes seeding particles overtake natural processes to form higher cloud fractions. As these clouds are in general colder, increases in their coverage lead to a larger TOA warming described above.

Overseeding occurs with both D19 and S89 ice cloud fraction approaches, but is more widespread with the narrower $S_i$
bounds used in the latter. With seeding particles being present in every gridbox of the cirrus scheme and their relatively low $S_{i,seed}$, even small increases in the amount of INPs and hence the amount of ice in an ice-supersaturated environment can lead to dramatic cloud fraction increases with S89. In addition, the low $S_{i,seed} = 1.05$ "out-competes" all other freezing modes to alter nucleation competition away from natural processes and towards seeding particles with both schemes. As this critical saturation ratio threshold is somewhat arbitrary, we investigate CCT sensitivity using seeding particles with a higher critical
$S_{i,seed} = 1.35$ for nucleation.

### 3.3    1.05 $S_i$ versus 1.35 $S_i$ seeding

Additional sensitivity tests were conducted by increasing $S_{i,seed}$ to 1.35 (1.35-seeding) in an effort to limit the overseeding found with $S_{i,seed} = 1.05$ (1.05-seeding). Figure 3b presents the net TOA radiative anomaly for both cloud fraction approaches for 1.35-seeding; results are also presented along with the $95\%$ confidence interval in Table 5. Note the difference in scale to
the 1.05-seeding TOA plot (Figure 3a). 1.35-seeding leads to a drastic reduction of the net TOA anomalies by a whole order of magnitude for both D19 and S89. We find TOA anomaly maxima of $0.33\,\mathrm{Wm^{-2}}$ and $0.37\,\mathrm{Wm^{-2}}$ for both D19 and S89, respectively, for the Seed1 case, with only S89 showing certainty on the $95\%$ confidence level. For both cases, the positive TOA anomalies are driven by positive CRE anomalies, fuelled mainly by weaker SW CRE (i.e. positive anomalies). This indicates a reduction in lower-lying mixed-phase or liquid clouds. In fact, for all S89 cases, the net CRE anomalies either match or exceed
the net TOA anomalies, meaning rapid cloud adjustments are likely contributing to the larger CRE anomalies. For example, for the S89 Seed0.1 case, the net CRE anomaly even contrasts the TOA anomaly. For D19, the TOA anomalies are driven mainly by weaker (i.e. positive) SW CRE anomalies, except for the Seed100 case where we find a stronger positive LW CRE of $0.18\,\mathrm{Wm^{-2}}$ (Table 5).

Consistent across both cloud fraction approaches is the large uncertainty relative to the absolute response, leading to uncer-
tainty in the net TOA radiation and CRE in Figure 3c,d. The only exception is for S89 Seed1, which at the $95\%$ confidence level shows a net warming effect (Table 5). However, with high uncertainty in the net TOA balance and the net CRE for 1.35-seeding with both ice-cloud fraction approaches, plus the use of the unrealistic ice saturation threshold for full gridbox coverage for ice clouds in S89 (Section 3.2), we focus our comparison for the rest of this study between 1.05-seeding and 1.35-seeding with D19 only.

As the Seed1 case showed the largest amount of cooling for 1.05-seeding and the largest warming for 1.35-seeding, we examine the microphysical response by comparing the zonal mean in-situ ice tracer anomalies for Seed1 and Seed1_1.35 in Figure 8. There is no clear response in the homogeneously-nucleated ice number anomalies within the cirrus regime (above the $238\,\mathrm{K}$ isotherm, dashed line in Figure 8) for both 1.05-seeding and 1.35-seeding. Plus, the overall zonal mean anomalies for both cases are uncertain according to the FDR analysis. The signal is clearer in the in-situ heterogeneous tracer anomaly where

**Table 5.** Five-year annual global mean net top-of-atmosphere total radiative balance (TOA) and net CRE, as well as SW CRE and LW CRE in $Wm^{-2}$ for D19 and S89 ice-cloud fraction approaches for seeding at $S_{i,seed} = 1.35$. Each quantity includes the $95\%$ confidence interval equating to two standard deviations of the mean values of the five-year data.

| Seeding Concentration [$L^{-1}$] | | 0.1 | 1 | 10 | 100 |
|---|---|---|---|---|---|
| D19 | net TOA | $0.15 \pm 0.44$ | $0.33 \pm 0.38$ | $0.15 \pm 0.47$ | $0.25 \pm 0.41$ |
| | net CRE | $0.07 \pm 0.42$ | $0.29 \pm 0.37$ | $0.11 \pm 0.48$ | $0.16 \pm 0.36$ |
| | SW CRE | $0.16 \pm 0.37$ | $0.28 \pm 0.38$ | $0.13 \pm 0.47$ | $-0.02 \pm 0.35$ |
| | LW CRE | $-0.10 \pm 0.16$ | $0.01 \pm 0.20$ | $-0.02 \pm 0.17$ | $0.18 \pm 0.19$ |
| S89 | net TOA | $-0.06 \pm 0.28$ | $0.37 \pm 0.30$ | $0.21 \pm 0.41$ | $0.16 \pm 0.43$ |
| | net CRE | $0.08 \pm 0.33$ | $0.37 \pm 0.30$ | $0.27 \pm 0.37$ | $0.44 \pm 0.51$ |
| | SW CRE | $0.08 \pm 0.40$ | $0.34 \pm 0.34$ | $0.20 \pm 0.34$ | $0.08 \pm 0.58$ |
| | LW CRE | $0.00 \pm 0.14$ | $0.03 \pm 0.08$ | $0.07 \pm 0.10$ | $0.36 \pm 0.12$ |

positive values are much more widespread and certain for 1.05-seeding than 1.35-seeding. Heterogeneous nucleation increases by more than $10\,L^{-1}$ in most regions for 1.05-seeding, and to a lesser extent with 1.35-seeding, where there is much widerspread uncertainty. The 1.05-seeding signal is less certain towards higher latitudes in the northern hemisphere (NH) where it shows mixed responses, as well as in the 1.35-seeding case. The differences between 1.05-seeding and 1.35-seeding are clearer in the anomalies for heterogeneous nucleation on mineral dust (Figure 8e,f). With the former, we find a similar situation

as before, where heterogeneous nucleation on dust is overtaken by heterogeneous nucleation on seeding particles. For Seed1 this switch to seeding-particle-dominant heterogeneous nucleation within cirrus clouds appears to replace some homogeneous nucleation throughout the troposphere and leads to a small negative TOA effect (Figure 3 and Table 4). We find the opposite behavior in the TOA response with Seed1_1.35, with the total heterogeneous nucleation anomaly closely resembling that for heterogeneous nucleation on mineral dust particles. Seeding particles in this case, and only in some areas, decrease the

number of ice particles formed by homogeneous nucleation, but are not effective at shutting off background heterogeneous nucleation processes. In fact, the amount of dust-driven nucleation increases throughout the troposphere, except for a small region at roughly $45\,°N$. However, both homogeneous and heterogeneous nucleation ice tracers for 1.35-seeding contain high uncertainty as shown by the stippling in Figure 8. The seeding ice tracer anomaly is more certain and shows increases between $1\,L^{-1}$ and $10\,L^{-1}$ in the tropics, and is much less widespread than the 1.05-seeding scenario. This is due to the fact that a $S_i$ of

1.35 occurs much less often in the atmosphere than a $S_i$ of 1.05. Therefore, seeding particles with a higher $S_{i,seed}$ are much less efficient in this case at consuming water vapor to overtake other nucleation modes like in the 1.05-seeding scenario, leading to the insignificant zonal ice tracer anomalies, despite a clear significant positive anomaly of heterogeneous nucleation on seeding particles (Figure 8h). Our results in this case only partially support the idea of an optimal seeding particle concentration around 1 INP $L^{-1}$ (Gasparini and Lohmann, 2016).

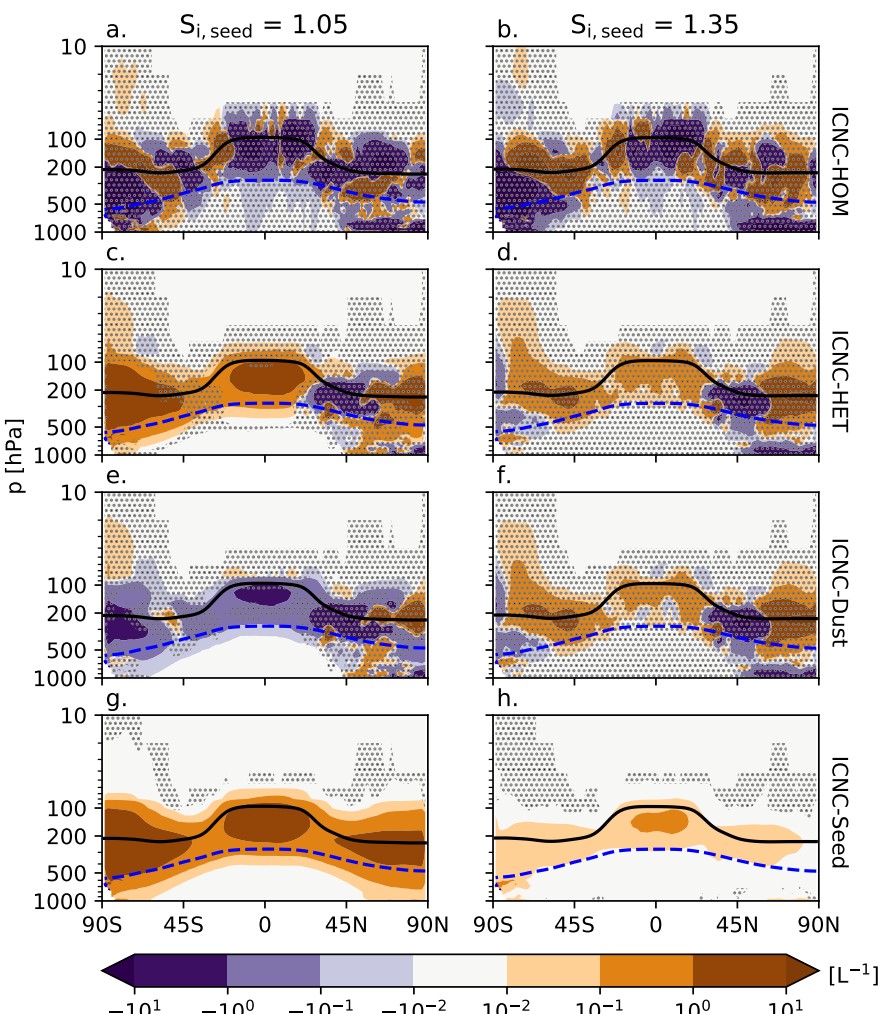

**Figure 8.** Five-year zonal mean ice number tracer anomalies in $L^{-1}$ between the Seed1 simulation and the unseeded reference case D19 for seeding particle critical saturation ratios 1.05 (left) and 1.35 (right). The anomalies include the in-situ homogeneously-nucleated ice number (a-b), the in-situ heterogeneous number (c-d), the heterogeneously-nucleated ice number formed on mineral dust particles (e-f), and the heterogeneously-nucleated ice number formed on seeding particles (g-h). The black line is the five-year mean zonal mean WMO-defined tropopause height on pressure levels, and the blue dashed line is the $238\,\mathrm{K}$ temperature contour. The stippling in the difference plots shows insignificant data points on the $95\,\%$ confidence level according to the independent t-test controlled by the "false discovery rate" method.

The global mean TOA radiative anomalies as well as the zonal mean ICNC tracer anomalies are mostly inconclusive for both 1.05-seeding and 1.35-seeding. Therefore, we examine the zonal mean TOA anomalies for each seeding concentration for both $S_{i,seed}$ thresholds in Figure 9. The most striking finding is that increasing $S_{i,seed}$ to 1.35 reduces the likelihood of overseeding, producing more regions of cooling for all seeding particle concentrations. For Seed100 with $S_{i,seed} = 1.05$ (Figure 9a) the

**Table 6.** Five-year annual mean net top-of-atmosphere total radiative balance in $\mathrm{Wm}^{-2}$ in the Northern and Southern Hemispheres between $60\,°\mathrm{N/S}$ and $90\,°\mathrm{N/S}$ for D19 for seeding with a critical ice saturation ratio of 1.05 and 1.35. Each quantity includes the $95\,\%$ confidence interval equating to two standard deviations of the mean values of the five-year data.

| Hemisphere | $S_{i,seed}$ | Seed0.1 | Seed1 | Seed10 | Seed100 |
|---|---|---|---|---|---|
| Northern | 1.05 | -0.06 $\pm$ 1.11 | 0.10 $\pm$ 1.00 | 0.39 $\pm$ 0.86 | 5.73 $\pm$ 1.08 |
| Southern |  | 0.15 $\pm$ 0.39 | 0.38 $\pm$ 0.62 | 2.49 $\pm$ 0.66 | 10.78 $\pm$ 0.43 |
| Northern | 1.35 | 0.14 $\pm$ 0.87 | 0.58 $\pm$ 1.45 | 0.29 $\pm$ 0.99 | 0.05 $\pm$ 1.25 |
| Southern |  | -0.27 $\pm$ 0.95 | -0.45 $\pm$ 0.71 | 0.03 $\pm$ 1.05 | 0.37 $\pm$ 0.45 |

maximum positive TOA anomaly is around $13.6\,\mathrm{Wm}^{-2}$ in the southern hemisphere (SH), whereas Seed100_1.35 (Figure 9b)
the maximum positive radiative forcing anomaly is about $1.5\,\mathrm{Wm}^{-2}$. There are small regions of negative forcing (i.e. a cooling effect) for all seeding particle concentrations for 1.35-seeding and seeding particle concentrations $\leq 10\,\mathrm{L}^{-1}$ for 1.05-seeding. For the 1.05-seeding case Seed0.1 and Seed1 show some degree of negative forcing between roughly $40\,°\mathrm{S}$ and $15\,°\mathrm{S}$, and between around $30\,°\mathrm{N}$ and $60\,°\mathrm{N}$. The cooling for Seed1 around $50\,°\mathrm{N}$ is the only appreciable signal at roughly $-1.1\,\mathrm{Wm}^{-2}$. Seed10 shows only a small degree of cooling around $30\,°\mathrm{S}$, with a small region with a maximal cooling of $-0.7\,\mathrm{Wm}^{-2}$ at roughly $35\,°\mathrm{N}$. As the Seed1 global mean anomaly showed the largest amount of cooling in the global mean net TOA anomalies for 1.05-seeding (Table 4), we added the $95\,\%$ confidence interval, which shows high uncertainty for 1.05-seeding.

1.35-seeding shows negative forcings in similar latitude regions, but for all seeding particle concentrations (Figure 9b). The $95\,\%$ confidence interval is shown here as well. For the Seed1 anomaly, three regions between $90\,°\mathrm{S}$ and $60\,°\mathrm{S}$, at around $15\,°\mathrm{N}$, and at roughly $45\,°\mathrm{N}$ show the largest amount of cooling. The largest negative anomaly is $-1.2\,\mathrm{Wm}^{-2}$ in the southern polar region. It is significant at the South Pole, perhaps indicating a higher CCT efficacy towards higher latitudes as well in our model (Storelvmo and Herger, 2014; Storelvmo et al., 2014). However, in other regions and like the 1.05-seeding case, the uncertainty around the Seed1 zonal mean anomaly is high. Therefore, it is difficult to determine the exact radiative response around the regions with the largest amount of cooling.

The indication that high latitude seeding may lead to a negative response in the Seed1_1.35 zonal anomaly in Figure 9 is in line with previous findings by Storelvmo and Herger (2014) and Storelvmo et al. (2014). To examine these higher-latitude regions further, Table 6 presents the five-year mean net TOA anomalies between $60\,°\mathrm{N/S}$ and $90\,°\mathrm{N/S}$ as well as the $95\,\%$ confidence interval around the mean. For 1.05-seeding, the only cooling response occurs for Seed0.1 in the NH, but contains an uncertainty one order of magnitude higher than the mean.

There is a clear overseeding response in both hemispheres for Seed100, with mean responses exceeding the net TOA anomaly (Table 4). As shown previously, the positive anomalies are drastically reduced for 1.35-seeding, which shows negative anomalies for Seed0.1 and Seed1 in the SH. The largest cooling response of about $-0.45\,\mathrm{Wm}^{-2}$ occurs for the Seed1 anomaly in the SH, but consistent with the other responses is highly uncertain.

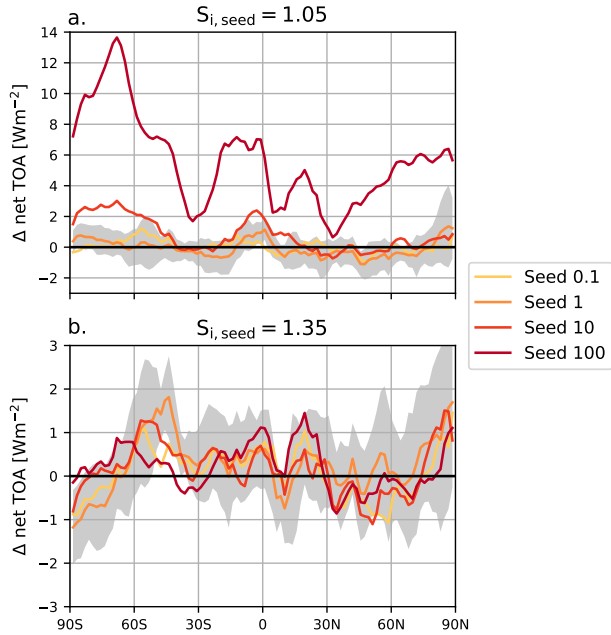

**Figure 9.** Five-year zonal mean net top-of-atmosphere (TOA) radiative balance anomalies in $\mathrm{Wm}^{-2}$ between total SW and LW fluxes for a critical seeding particle saturation ratio of (a) 1.05 and (b) 1.35 for each seeding particle concentration minus the reference unseeded D19 simulation. The grey shaded area is the $95\%$ confidence interval, representing the two-times standard deviation interval, of the Seed1 anomaly based on the variance of the five-year data. Please take note of the different scales in panels (a) and (b).

Figure 10 shows the vertical profiles of IWC and LWC anomalies for each seeding particle concentration, like in Figure 7 but for 1.35-seeding. Unexpectedly, we find that 1.35-seeding does not impact IWC within the cirrus regime, and leads to only very small anomalies in the mixed-phase and liquid regimes. There does appear to be a feedback on lower-lying clouds as the LWC anomalies are larger. LWC increases in the lower part of the mixed-phase regime and the uppermost part of the liquid regime by a small amount only for Seed10_1.35 and Seed100_1.35. At warmer temperatures, the LWC decreases for all seeding particle concentrations. For Seed0.1_1.35 and Seed1_1.35, the LWC decreases in the lowermost mixed-phase regime. It is unclear why there is a shift from negative to positive LWC anomalies in the mixed-phase regime with increasing seeding particle concentrations. The lack of an IWC response, combined with the increase in heterogeneously nucleated ice on seeding and mineral dust particles for Seed1_1.35 (Figure 8), indicates that seeding, to some extent, impacts the ice crystal size, which in turn affects sedimentation from the cirrus regime. Ice crystal size anomalies are also highly uncertain for the 1.35-seeding case (not shown), with a mixed signal in the cirrus regime. At least at lower seeding particle concentrations, it may be that seeding forms larger ice crystals that sediment into the mixed-phase regime and consume liquid water more efficiently via the WBF process or riming. At higher seeding particle concentrations, ice growth may be limited and therefore cirrus ice crystals may be smaller, which weakens the sedimentation flux into the mixed-phase regime. The lack of large ice crystals in the mixed-phase regime reduces cloud droplet consumption via the WBF process or riming, and increases LWC at least for two of our

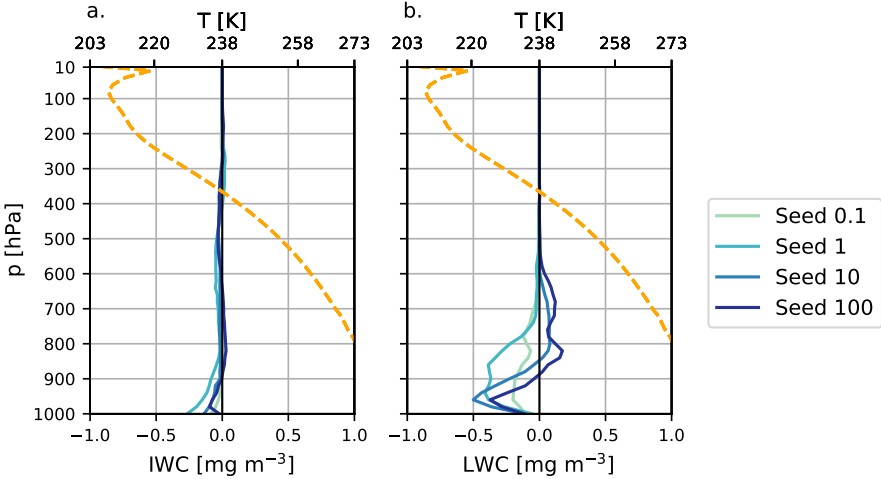

**Figure 10.** Five-year annual global mean (a) IWC and (b) LWC anomaly profiles in $\mathrm{mg/m^3}$ as in Figure 7, but for $S_{i,seed}$ = 1.35.

scenarios. In these cases, the positive LWC anomalies in the upper mixed-phase regime may equate to smaller cloud droplets that lead to a weaker sedimentation flux, which may result in few cloud droplets in the liquid regime. Overall, however, the
cirrus seeding signal on lower-lying cloud feedbacks is ambivalent for 1.35-seeding, as uncertainty surrounding the responses in the cirrus regime remains high.

### 3.4   Seasonal seeding anomalies

Our annual mean results for 1.35-seeding cannot confirm the findings from previous studies that higher latitude regions are the most desirable for CCT implementation (Storelvmo and Herger, 2014; Storelvmo et al., 2014; Gruber et al., 2019). Therefore,
we examine whether there is a seasonal sensitivity on CCT efficacy. Figure 11 shows the NH winter and summer zonal mean TOA radiative anomalies as well as the constituent SW and LW flux anomalies for 1.35-seeding. The net TOA is presented in the first column for NH winter (top) and summer (bottom), with the SW and LW flux anomalies in the second and third columns respectively. Uncertainty is plotted around the Seed1_1.35 mean anomaly. A clear seasonal pattern is difficult to decipher from the TOA anomalies, except that there appear to be more widespread positive TOA anomalies during NH winter.
In the northern polar regions (north of $60\,°\mathrm{N}$), only the Seed10 and Seed100 TOA anomalies show any cooling during NH winter, but at specific latitudes. Due to the negligible SW flux at high latitudes during winter, the net TOA response is entirely driven by LW anomalies. Our model suggests that seeding particles in this case act to enhance the large LW CRE in this region (roughly $11\,\mathrm{Wm^{-2}}$ in the unseeded case) to produce mostly positive TOA anomalies. However, the uncertainty around the Seed1 mean anomaly in this region is high (Figure 11). We find smaller regions of cooling with net negative TOA responses
for Seed1 during NH winter in the northern mid-latitudes (between $30\,°\mathrm{N}$ and $45\,°\mathrm{N}$) and in the southern polar regions (south of $60\,°\mathrm{S}$), (Figure 11a). The cooling in the northern mid-latitudes is driven by a large decrease of around $-2.5\,\mathrm{Wm^{-2}}$ in the

net TOA LW flux, coupled to a weaker TOA SW flux in the same region. This is the same region where we find the negative homogeneous nucleation and total heterogeneous nucleation ice anomalies in Figure 8, indicating that seeding particles in this region may cause a shift in cirrus formation pathway or have an impact on lower-lying mixed phase clouds. The smaller net cooling for the SH in the same time period appears to be driven by a stronger TOA LW that is partially compensated by a stronger TOA SW.

During NH summer the net TOA response is similar to that during NH winter, with the exception of the northern polar regions where the maximum positive anomaly for Seed10_1.35 is nearly $5.0\,\mathrm{Wm}^{-2}$. There are a few regions of cooling in the NH that are driven primarily by SW anomalies. The maximum amount of cooling of $-2.5\,\mathrm{Wm}^{-2}$ for Seed1_1.35 occurs around $70\,°\mathrm{N}$, and is driven by a weaker TOA SW flux, indicating a feedback on lower-lying clouds, that is partly compensated by a weaker (i.e. more positive) TOA LW flux. We also find a similar pattern in the NH tropics around the location of the Intertropical Convergence Zone (ITCZ). Thicker in-situ cirrus clouds to some extent reflect more SW (Krämer et al., 2020), similar to the Twomey effect for lower-lying liquid or mixed-phase clouds (MPCs). However, they also induce a strong compensating LW effect as a result of seeding. In the southern polar region we find a cooling response for Seed1_1.35 of nearly $2.5\,\mathrm{Wm}^{-2}$. Similar as before, the cooling in this region is driven by LW emission due to a lack of SW radiation during SH winter; however, the uncertainty is wide enough in this case that we cannot determine whether the net TOA anomaly is indeed neutral.

### 3.5  Stratospheric Effects

So far our analysis focused on the changes in the troposphere leading to the TOA overseeding presented in Figure 3. However, our findings also point to stratospheric effects as a results of seeding, particularly the positive ICNC anomalies in the lower stratosphere (Figure 5) and the subsequent cloud fraction increase (Figure 4). The former can be partially explained by the seeding strategy we utilize in our cirrus scheme. Seeding particles are available in every gridbox of the cirrus scheme up to the $100\,\mathrm{hPa}$ pressure level. This places some of our seeding particles firmly within the troposphere in the tropics, but in the lower stratosphere in the mid and high latitudes. Therefore, seeding particles are present in environments with little competition between mineral dust (i.e. low INP environments) or liquid sulphate particles, leading to wide extents of the lower stratosphere with large positive ICNC anomalies. Cloud fraction increases accordingly with larger ice crystal number concentrations from seeding. This effect is more widespread with S89 than D19 due to the ice saturation threshold for full gridbox coverage of ice clouds used in the former.

What remains unclear is the positive in-situ homogeneously-nucleated ice number anomaly in the mid-latitudes and towards the poles in the lower stratosphere (Figure 6, top panel), and the higher cloud fractions that extend to pressure levels less than $100\,\mathrm{hPa}$ (i.e. at higher altitudes). As both cloud fraction approaches are relative humidity based, the patterns observed in stratospheric cloud fraction indicate a dynamic response to the INP perturbations by increasing temperature and consequently enhancing upwelling of water vapor into the stratosphere from the tropical troposphere as shown in Figure 12. The anomalies for lower seeding concentrations and for all simulations with $S_{\mathrm{i,seed}} = 1.35$ are insignificant. Here, we only present the anomalies at pressure levels lower than $300\,\mathrm{hPa}$ (higher altitudes) to focus on the effects in the upper troposphere and the stratosphere.

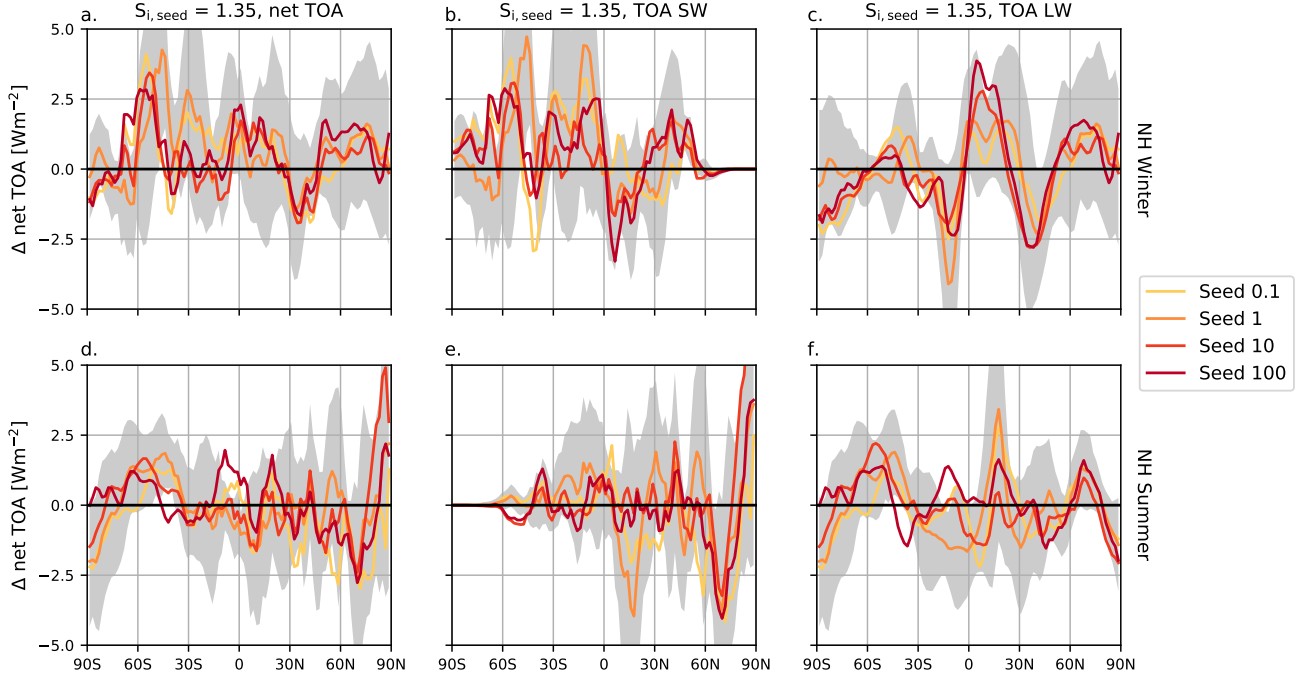

**Figure 11.** 1.35-seeding zonal mean radiative balance anomalies in $\text{Wm}^{-2}$ for all seeding particle concentrations for the net TOA (a,d), the TOA SW (b,e), and TOA LW (c,f). The top row shows the four-year zonal mean for NH winter (December - February) and the bottom row is the five-year zonal mean for NH summer (June - August). The grey shaded area is the $95\,\%$ confidence interval around the mean Seed1 anomaly, representing the two-times standard deviation interval, based on the variance of the annual data.

Overseeding in Seed100 leads to a positive temperature anomaly of more than $4\,\text{K}$ in the tropical troposphere (Figure 12a). As a result of warmer temperatures, the saturation specific humidity increases. Therefore, the specific humidity can increase as well (Figure 12b). This appears to enhance water vapor upwelling into the lower stratosphere from the tropical troposphere, as indicated by the positive specific humidity anomaly above the tropopause (Figure 12b) that also extends into the middle stratosphere. Water vapor in the stratosphere has a cooling effect (Rind and Lonergan, 1995), as indicated by the temperature

response above the tropopause in the tropics and between $45\,°\text{N/S}$ and $90\,°\text{N/S}$. In the same region, updraft velocities increase by more than $0.2\,\text{cm}\,\text{s}^{-1}$. As we observe larger ice-cloud fractions in this region (Figure 4), enhanced LW cloud-top cooling likely fuels the observed positive updraft anomaly. We find LW-cooling in the tropics in upper troposphere and in the extra-tropics in the stratosphere in Figure 13. The latter is likely due to the positive water vapor anomaly in the lower stratosphere (Figure 12b). At lower levels we find LW warming, likely caused by more trapping from more frequent and optically thicker

cirrus clouds. The increase in updraft velocity, in combination with the positive specific humidity anomaly, not only allows the seeding particles to form abundant ice particles, but also allows air parcels to reach the critical saturation for homogeneous nucleation. There are also small areas in the lower stratosphere where the anomaly of ice formed heterogeneously on mineral dust particles is positive (Figure 6). This enhancement of natural ice formation processes at lower levels in the stratosphere

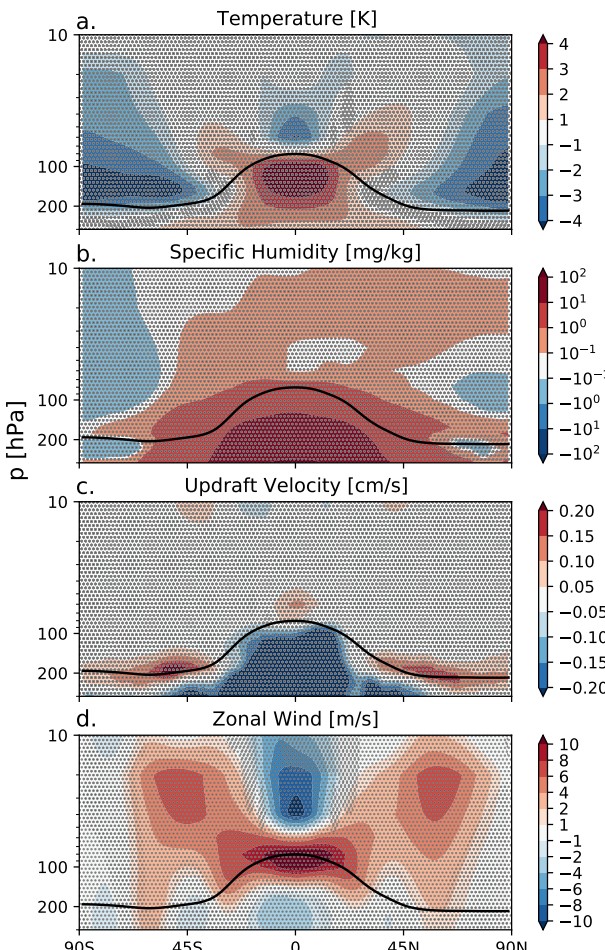

**Figure 12.** Five-year zonal mean anomalies of (a) temperature in K, (b) specific humidity in mg/kg, (c) updraft velocity in cm/s, and (d) zonal wind in m/s for D19 with a seeding particle concentration of 100 INP L$^{-1}$. Anomalies are only shown for the upper troposphere and the stratosphere between $300\,\mathrm{hPa}$ and $10\,\mathrm{hPa}$. The black line is the five-year mean zonal mean WMO-defined tropopause height on pressure levels. The stippling in the difference plots shows insignificant data points on the $95\,\%$ confidence level according to the independent t-test controlled by the "false discovery rate" method.

in response to overseeding in the troposphere (Section 3.2), plus the widespread positive anomaly of ice formed on seeding particles in the same region leads to a higher abundance of clouds that likely contribute to the overall TOA warming effect (Figure 3).

The temperature anomaly presented in Figure 12a is not restricted to the lower stratosphere where we find enhanced ice formation, which indicates that seeding could impact the wider stratosphere as a whole via a dynamic feedback on the Brewer-Dobson Circulation (BDC), (Butchart and Scafie, 2001; Rind et al., 2001; Butchart et al., 2006; Butchart, 2014). The BDC

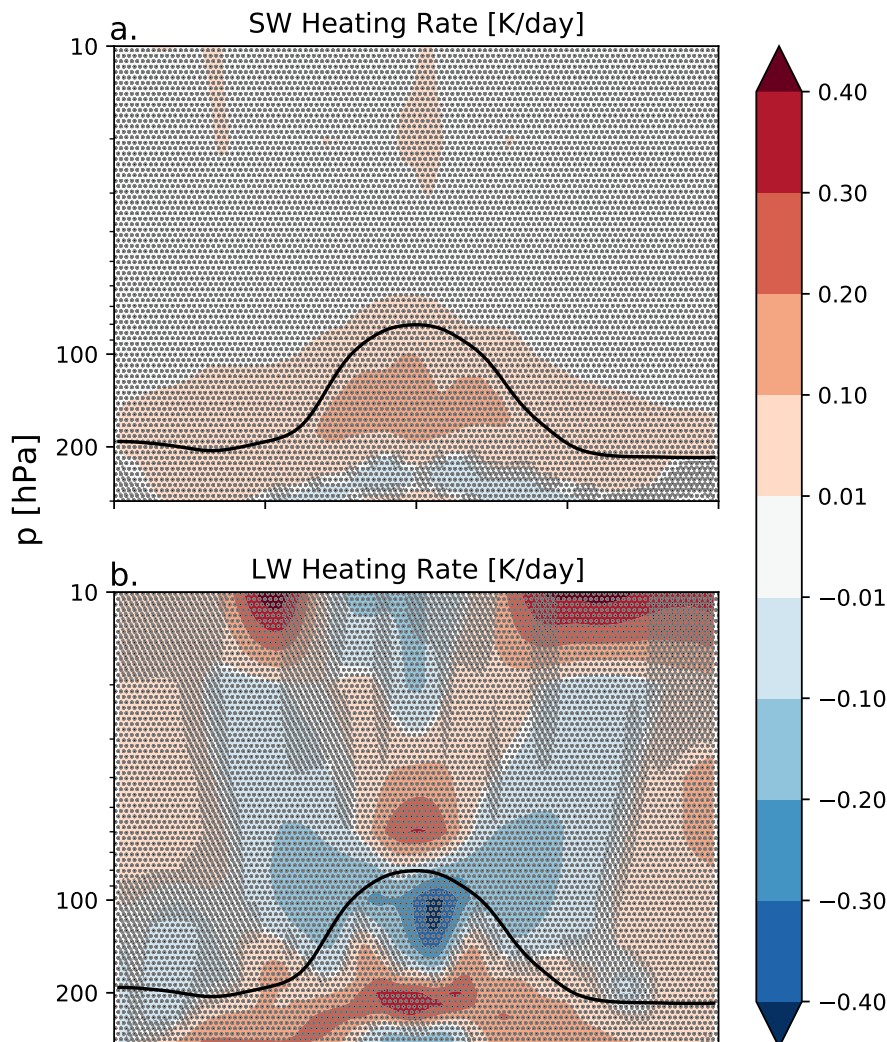

**Figure 13.** Five-year zonal mean (a) SW, and (b) LW heating rate anomalies in $\mathrm{K/day}$ for D19 with a seeding particle concentration of 100 INP $\mathrm{L^{-1}}$. Anomalies are only shown for the upper troposphere and the stratosphere between $300\,\mathrm{hPa}$ and $10\,\mathrm{hPa}$. The black line is the five-year mean zonal mean WMO-defined tropopause height on pressure levels. The stippling in the difference plots shows insignificant data points on the $95\,\%$ confidence level according to the independent t-test controlled by the "false discovery rate" method.

describes the global mass transport from the troposphere into the stratosphere, where air rises in the tropics and descends over higher latitudes. One of the main findings following numerous studies on greenhouse-gas driven climate change is a speeding up of this overturning circulation, with enhanced tropical mass upwelling, leading, in general, to a cooler stratosphere and a warmer troposphere (Butchart, 2014). Calvo et al. (2010) studied the enhancement of gravity wave-fueled tropical upwelling into the stratosphere during warm El Niño-Southern Oscillation (ENSO) events. They found that during such events tropo-

spheric warming paired with tropical stratospheric cooling enhances the meridional temperature gradient. This strengthens the subtropical-jet, as seen by the increase in the zonal mean zonal wind, which is proportional to enhanced gravity wave drag forcing that fuels increased tropical upwelling (Calvo et al., 2010). Our results show a similar response with the positive temperature anomaly in the tropical tropopause (Figure 12a) that subsequently intensifies the sub-tropical jet, which we diagnosed from the zonal mean zonal wind anomaly in Figure 12d. The updraft anomaly in Figure 12c on the one hand shows a negative

updraft anomaly in the troposphere as a result of enhanced atmospheric stability due to a warmer upper troposphere, similar to the stabilization found by Kuebbeler et al. (2012) following stratospheric sulphur injections. On the other hand the positive updraft anomaly indicates a small enhancement of tropical upwelling in the stratosphere that would indicate a strengthening of the BDC. However, with this effect, the downward branch of the BDC leads to stronger warming in the stratosphere at higher latitudes (see Figure 6 from Calvo et al. (2010)) due to adiabatic compression. Our results show a negative temperature anomaly

at high latitudes in contrast to BDC-enhancement findings, rather pointing to a weakening of the downward branch. Instead, our results point to enhanced radiative cooling of the lower stratosphere in response to positive specific humidity anomalies.

## 4 Discussion

The results we presented in this study highlight a few important factors governing the sensitivity of CCT, namely, the approach for calculating ice-cloud fractions, the representation of cirrus ice nucleation competition and stratiform ice microphysics, and

the choice of $S_{i,seed}$ for ice nucleation to occur on seeding particles. Our results also show the potential for unwanted side-effects of CCT on mixed-phase clouds and in the stratosphere.

In a first step, we tested the sensitivity of CCT between the original approach in ECHAM-HAM for calculating cloud fractions by Sundqvist et al. (1989), (S89) and the updated approach by Dietlicher et al. (2018, 2019), (D19). Overall we found that the D19 scheme reduces net TOA warming (i.e. the positive radiative forcing) by a factor of more than two for Seed100

compared to S89 (Figure 3). Similar to the findings by Gasparini and Lohmann (2016), more frequent ice formation on seeding particles in our simulations led to cirrus cloud formation in previously cloud-free regions, using both ice-cloud fraction approaches. The conceptual difference between the two cloud fraction approaches can explain why positive cloud fraction anomalies were not as large with D19 than S89. With the latter, the ice saturation threshold for full gridbox coverage of cirrus clouds meant that more frequent ice formation on seeding particles in ice supersaturated conditions artificially expanded cloud

fractions to unity, increasing the zonal average cloud fraction by more than $12\%$. On the other hand, while the reduction of homogeneous nucleation with D19 (Figure 6) reduced the frequency of fully covered grid boxes, the increase of heterogeneous nucleation on seeding particles increased the fractional cloud cover. Therefore, while both approaches showed a positive radiative effect as a result of seeding, D19 responses are lower because ice formation at a supersaturation suitable for heterogeneous nucleation on seeding particles does not induce as high cloud fractions as in S89. This highlights limitations in both approaches

for calculating ice cloud fractions. Where S89 artificially expanded ice-cloud fractions upon ice formation at supersaturation with respect to ice, ice-cloud fractions using D19 might be artificially low following seeding due to the criterion for full gridbox ice-cloud coverage only reached at homogeneous nucleation conditions. These limitations have wider implications on the

radiative transfer calculations used to compute TOA fluxes. The prognostic cloud scheme by Muench and Lohmann (2020) that explicitly calculates variables for cloud-free and cloudy air, including in-cloud water vapor, could be used to overcome some of the limitations of the RH-based approaches, S89 and D19, and investigate cloud-fraction sensitivity to seeding particles.

Compared to CCT studies using the same model, ECHAM-HAM (Gasparini and Lohmann, 2016; Gasparini et al., 2017, 2020), we found much higher positive net TOA anomalies in response to seeding. This points to differences in the in-situ cirrus scheme (Kärcher et al., 2006, Kuebbeler et al., 2014, Muench and Lohmann 2020) and the treatment of ice microphysics (P3: Morrison and Milbrandt 2015, Dietlicher et al., 2018, 2019 versus 2M: Lohmann et al., 2007). However, the propensity of heterogeneous nucleation on seeding particles to alter cirrus ice formation in our study is consistent with ongoing research into the complexities of cirrus ice nucleation competition (Lohmann et al., 2008; Mitchell and Finnegan, 2009; Jensen et al., 2016a, b; Kärcher et al., 2022). In this study we updated the scaling of available aerosols for each freezing mode in the cirrus scheme by the fraction of ice in each nucleation mode from the previous timestep out of the total amount of stratiform pre-existing ice (Section 2.1). We deem this approach as more accurate than the previous approach to scale the available aerosols by the total amount of pre-existing ice. In a series of tests (not shown) we found that the updated scaling generated more heterogeneously-nucleated ice that only slightly decreased the amount of homogeneously-nucleated ice. The overall impact of the updated scaling did produce more in-situ ice from the cirrus scheme, but did not greatly alter ice nucleation competition. As such, we do not attribute the majority of the differences in our results to previous CCT studies to the scaling changes of available aerosol in each nucleation mode in the cirrus scheme.

It is more likely that our results differ from previous CCT studies due to the updated approach to represent ice as a single prognostic category in the microphysics scheme (P3, Morrison and Milbrandt, 2015; Dietlicher et al., 2018, 2019), as opposed to the size-separation approach of in-cloud ice and snow (2M, Lohmann et al., 2007) in earlier versions of the model. The single category approach with P3 is achieved by a prognostic treatment of sedimentation, whereby this process is calculated as a vertical transport tendency based on the total ice particle size distribution (PSD). Ice removal is represented in a much more realistic way than in the 2M scheme, in which only a part of the ice PSD could undergo sedimentation. With the 2M scheme, as soon as ice grows larger than a certain threshold size it is converted to the snow category and falls out of the atmosphere in a single model timestep. In order to maintain realistic cloud IWC values in the 2M scheme compared to observations, ice removal via snow formation was artificially enhanced by converting more cloud ice to snow. This was achieved by setting the tuning parameter for snow formation via ice crystal aggregation to an artificially high value ($\gamma_s$ = 900, Neubauer et al., 2019; Dietlicher et al., 2019). This is no longer the case with the P3 scheme coupled to D19. A consequence of the slower and more realistic ice removal is that the ice crystal aggregation tuning parameter is no longer relevant (Table 2, Dietlicher et al., 2019). Instead, ice crystal removal via larger crystals is augmented by an ice self-collection tuning parameter that is set to 5.5 (Section 2.1). Overall this means that ice in P3 remains in the atmosphere for a longer period of time. As a result, when seeding particles are introduced as additional INPs with P3, the more numerous and smaller ice crystals (Figure 5 and Figure 6) do not necessarily grow into snow-sized ice particles and quickly sediment. This explains why we obtained much higher TOA radiative responses to seeding in this study compared to Gasparini and Lohmann (2016) and Gasparini et al.

(2017). These previous CCT studies that did not include a prognostic representation of ice sedimentation likely underestimated the overseeding response as ice was removed too readily.

Another striking result from our study was the sensitivity of our model to the choice of $S_{i,seed}$. In a separate test, we increased $S_{i,seed}$ from 1.05 to 1.35 in an attempt to avoid impacts on heterogeneous nucleation on mineral dust particles, and only target homogeneous nucleation of liquid sulphate aerosols. Our results to some extent confirmed this hypothesis. 1.35-seeding led to drastic net TOA reductions on a global-scale (Figure 3) and zonally (Figure 9) compared to 1.05-seeding. The net TOA reductions we found with 1.35-seeding were also confirmed by the zonal ICNC tracer anomalies (Figure 8). For 1.35-seeding, seeding particles were much less effective at overtaking other nucleation modes. Therefore, our results likely point to a trade-off when pursuing further CCT studies: increasing $S_{i,seed}$ is likely an attractive alternative to avoid wide nucleation competition alterations as seen with lower $S_{i,seed}$, however, the scale to which seeding particles could produce the desired cooling effect remains to be examined with more detailed regional analyses.

The potential side effects of CCT were only starting to be investigated within the last few years (Lohmann and Gasparini, 2017). In high resolution simulations Gruber et al. (2019) found that CCT not only resulted in thinner cirrus clouds, but also the larger ice particles formed by heterogeneous nucleation on seeding particles acted to reduce lower-lying MPCs through enhanced riming and ice crystal growth via the WBF process. The combination of these two effects resulted in a net TOA cooling effect. Gasparini et al. (2017) also found an impact on lower-lying clouds in their simulations using an increased sedimentation velocity as a proxy for CCT with seeding particles, following Muri et al. (2014). The "redistribution" of ice to lower-lying MPCs counteracted cooling from reduced cirrus cloud fractions in their sedimentation simulations. In their CCT simulations using seeding INPs, they also found an MPC feedback, resulting from increased convective activity drying the lower troposphere that led smaller MPC fractions. As noted above, our results also showed a sedimentation flux reduction in line with a reduction in convective activity due to LW warming by a maximum of $0.3 - 0.4\,\mathrm{K/day}$ for D19 Seed100, which led to tropospheric stabilization (Figure 12). However, our results do not show significant cloud fraction anomalies in the mixed-phase regime, and rather highlight that the weaker sedimentation flux explains the positive LWC anomaly as shown in Figure 7. MPCs with larger LWC led to stronger SW cooling, but this was outweighed by warming from the increase of cirrus cloud fractions with smaller and more numerous ICs in the 1.05-seeding case.

Seeding particles were simulated to nucleate ice as a threshold freezing process in our model (Section 2.1), meaning all aerosol particles within the mode that were available in any given gridbox would nucleate ice upon the right conditions being met. This led to the large overseeding responses we found with the lower $S_{i,seed}$ = 1.05 that were drastically reduced by increasing $S_{i,seed}$ to 1.35. Based on our findings, it is clear that the choice of $S_{i,seed}$ is an important factor in determining CCT efficacy. Therefore, more detailed investigations of specific seeding particle materials and their ice-nucleating ability, perhaps in line with the continuous freezing process in this study (Section 2.1), are needed in order to move CCT studies in line with potential real-world applications.

We also showed that seeding with small particles appears undesirable as they lead to smaller ice particles following nucleation, reduced sedimentation fluxes, and longer-lived cirrus clouds. Gasparini et al. (2017) found seeding with larger particles to lead to larger cooling that can somewhat offset $CO_2$-induced warming.

The timing of seeding particle injection is also key so as to only seed regions prior to natural cirrus formation. This poses one of the largest uncertainties for CCT, as forecasting cirrus formation is difficult with current techniques. In addition, predicting where cirrus ice forms predominantly via homogeneous nucleation will be a significant challenge. Studies like those by Storelvmo and Herger (2014), Storelvmo et al. (2014), and Gruber et al. (2019) suggest that high-latitude, wintertime seeding is optimal primarily due to the lack of cirrus SW CRE during this period (i.e. cirrus only act in the LW spectrum via warming). In addition, higher latitude regions on average contain lower background aerosol concentrations, making them more ideal for homogeneous nucleation within cirrus. In line with Penner et al. (2015), our results do not confirm high-latitude wintertime seeding as an effective strategy, as we found that seeding amplifies the already large cirrus LW CRE in such regions for most cases to produce net TOA warming. Overall, our results indicate that more thorough investigations of targeted seeding within high latitude regions are needed for future work. This could be partially addressed with more high resolution studies using cloud-resolving models, like Gruber et al. (2019). On the other hand, further CCT studies using GCMs can address this issue by using a more complex, non-uniform approach to include seeding particles as INPs for cirrus ice nucleation competition. This is the subject of further investigation in our group.

The results presented in this study underscore the need to investigate the methods in which seeding particles are included as INPs within models. We propose three topics in which future work should focus: (1) a dedicated seeding particle parameterization that accounts for the mechanism of ice formation on seeding particles and feasible $S_{i,seed}$ values, instead of using a somewhat arbitrary value as was used in CCT studies to date, (2) an optimal seeding particle size, and (3) the spatial and temporal distribution of seeding particles in models.

Additionally, there are still large differences in the outcome of CCT studies between the two leading climate models that at the time of writing were used to study CCT globally, ECHAM6-HAM2 (ECHAM) and CESM1-CAM5 (CESM) (Storelvmo et al., 2013; Storelvmo and Herger, 2014; Storelvmo et al., 2014; Penner et al., 2015; Gasparini and Lohmann, 2016; Gasparini et al., 2017, 2020). Such differences can be partially attributed to a lack of reliable remote sensing measurements and in-situ observations of cirrus in order to constrain models, though this gap is starting to be closed with more recent studies (Krämer et al., 2016; Sourdeval et al., 2018; Gryspeerdt et al., 2018; Krämer et al., 2020).

Gasparini et al. (2020) were the first to present a comparative analysis of CCT between the two models. They noted a much higher cooling potential of CCT in CESM than in ECHAM ($-1.8\,\mathrm{W\,m^{-2}}$ versus $-0.8\,\mathrm{W\,m^{-2}}$). This is in part due to the different cirrus ice microphysics scheme used in either model. CCT studies using CAM5 to date follow the scheme by Barahona and Nenes (2008, 2009) that explicitly links the number of ice crystals formed from nucleation events to the dynamical environment as well as to the properties of the available INPs (i.e. number, size, freezing threshold). This scheme replaced an earlier one by Liu and Penner (2005) that was based on classical nucleation theory for ice formed by deposition or immersion freezing on mineral dust and soot particles, respectively (Liu et al., 2007; Barahona and Nenes, 2009; Liu et al., 2012). CCT studies using ECHAM6, including this study, also use a cirrus ice nucleation scheme that resolves ice number dependence on aerosol properties that is based on the time integration of $S_i$ by Kärcher et al. (2006), following updates made by Kuebbeler et al. (2014) and Muench and Lohmann (2020), (Section 2).

A notable difference between the two models is the inclusion of pre-existing ice particles in ECHAM, which are not included in the default version of CESM (Gasparini et al., 2020) nor in any CCT study using this latter model. The one exception is the study by Penner et al. (2015), who included pre-existing ice particles in some of their simulations, following Shi et al. (2015). They found no significant cooling by CCT in any of their cases where pre-existing ice particles were included in CESM, despite better agreement with observations of ICNC in the temperature range relevant for CCT than the cases without pre-existing ice particles (Penner et al., 2015). The inclusion of pre-existing ice acts to decrease the frequency of homogeneous nucleation in all cirrus clouds as more water vapor is consumed on these particles and prevents the development of high ice supersaturation. Therefore, the potential homogeneous-to-heterogeneous nucleation shift as a result of CCT is also reduced when pre-existing ice particles are considered. This is the case in ECHAM and explains why the "optimal" seeding particle concentration differs between the two models ($1\,\text{L}^{-1}$ for ECHAM versus $18\,\text{L}^{-1}$ for CESM), (Gasparini et al., 2020). Almost any amount of cooling that was found in ECHAM as a result of CCT is smaller in magnitude than in CESM (Storelvmo et al., 2013; Gasparini and Lohmann, 2016; Gasparini et al., 2020), (with the notable exception of Penner et al. (2015), see above), or, as is this case with our results, is not evident (Figure 3). Moreover, for similar seeding particle concentrations ($> 10\,\text{L}^{-1}$) ECHAM produces more numerous ice crystals, which contribute to new cirrus cloud formation or cirrus lifetime prolongation (i.e. an overseeding response) that lead to positive TOA anomalies (Figures 5 and 6, and Gasparini and Lohmann (2016)). In CESM, CCT in general leads to a reduction in cirrus frequency (Storelvmo et al., 2013; Storelvmo and Herger, 2014; Storelvmo et al., 2014; Gasparini et al., 2020) that is not present in ECHAM (Gasparini and Lohmann, 2016; Gasparini et al., 2020). While our results show a reduction in the frequency of the lowest cirrus clouds, we also find new cloud formation in previous clear-sky regions (Figure 4). Cloud fraction anomalies in our study are amplified by the slower ice removal when using the P3 scheme (as discussed above). This highlights differences between the cloud fraction approaches used in CESM (Slingo, 1987; Gettelman et al., 2010) and ECHAM (Sundqvist et al., 1989; Dietlicher et al., 2018, 2019), (Section 2).

Finally, inconsistent approaches also exist between studies using the same model. For example, in our study, we excluded the orographic gravity-wave vertical velocity parameterization by Joos et al. (2008, 2010), unlike Gasparini and Lohmann (2016). Verification of this approach is presented in Appendix A. In summary, we found that the orographic gravity wave paramaterization in its current form is incompatible with ECHAM6.3 when using the P3 scheme, and leads to worse agreement of median ICNC values between the model and the in-situ observations by Krämer et al. (2020). As gravity waves were found to be an influential component for cirrus ice nucleation competition (Jensen et al., 2016a), we argue that this incompatibility when using the parameterization by Joos et al. (2008, 2010) with the P3 cloud microphysics scheme should be investigated in greater detail in future work.

## 5 Conclusions

We tested the sensitivity of CCT efficacy to the approach used for calculating ice cloud fractions and $S_{\text{i,seed}}$ using the new physically-based P3 ice microphysics scheme in the ECHAM-HAM GCM (Dietlicher et al., 2018, 2019). We conclude with the following main findings:

1. Increasing the RH threshold for the calculation of cirrus cloud fractions reduces the positive forcing from overseeding by avoiding artificial cirrus cloud expansion upon ice nucleation.

2. The prognostic treatment of sedimentation in the P3 microphysics scheme, leading to slower and more physically-based ice removal, is likely the reason why we find such large seeding responses compared to the study by Gasparini and Lohmann (2016), using the default ECHAM 2M scheme. Our model produces smaller and more numerous ice particles that amplify the already longer ice residence times within clouds to induce a strong positive TOA forcing.

3. Increasing $S_{i,seed}$ to 1.35 reduces the large overseeding found with the lower $S_{i,seed}$ of 1.05, but also reduces the competition ability of seeding particles, which amplifies uncertainty in the mean response.

4. Globally CCT is unlikely to produce the desired cooling effects due to dynamic adjustments and background aerosol concentration heterogeneity. Instead, small regions centered around specific latitudes show only a small potential of targeted seeding.

5. Our results do not confirm that wintertime high-latitude seeding can optimize CCT efficacy, contrasting the results obtained by Storelvmo and Herger (2014) and Storelvmo et al. (2014). Thus, targeted seeding for specific regions or time periods should be further investigated in higher resolution modeling studies like the one by Gruber et al. (2019).

In line with the proposed real-world delivery mechanism of seeding particles using commercial aircraft (Mitchell and Finnegan, 2009), there is a need to test the impact of aviation soot emissions on cirrus formation by including soot particles as potential INPs within the cirrus regime (e.g., Lohmann et al., 2020). Following on from that analysis, designing future CCT studies to include aviation will more closely align modelling studies to potential implementation.

Finally, based on our discussion, we extend the assertion by Gasparini et al. (2020) that a consistent CCT approach among climate modeling groups is needed, especially if the desire amongst the scientific community is to critically assess this proposed method as a feasible climate intervention strategy.

**Appendix A: Orographic cirrus verification**

In this section we verify our approach to not include orographic effects on vertical velocity in our model, using the P3 ice microphysics scheme. We ran an additional reference simulation with the D19 setup with the orographic velocity enhancement parameterization by Joos et al. (2008, 2010) activated (P3 Oro). Here we provide a comparative analysis between the Full_D19 simulation of the main text (i.e.: P3 Ref that does not include orographic effects on vertical velocity) with P3 Oro.

In the main text we validated our model using the in-situ measurements by Krämer et al. (2020), (K20). Here we extend this validation in Figure A1 that shows the model validation comparison between P3 Ref, P3 Oro, and K20. The most notable feature we find with P3 Oro is large increase in ICNC between roughly 200 K and 220 K in Figure A1a. The largest difference is at 202 K, where median ICNC increased by over two orders of magnitude compared to P3 Ref. There is a similar magnitude of discrepancy between the K20 data and P3 Oro. With the orographic velocity component, the model predicts high frequencies

(near $100\%$) of ICNC around $2000\,\mathrm{L}^{-1}$. Such values in the K20 data (Figure A1c) and P3 Ref (Figure A1b) have a frequency of less than $1\%$. We note that P3 Ref and P3 Oro show much less variability than the K20 data as they are averaged over five years, whereas the aircraft data are instantaneous. However, we also note that P3 Ref shows excellent agreement in median ICNC values with the K20 data that is not evident for cirrus clouds at lower temperatures with P3 Oro.

Figure A2 presents the spatial distributions of ICNC per temperature bin from $203\,\mathrm{K}$ to $233\,\mathrm{K}$ for the ten-year mean DAR-
DAR observations (Sourdeval et al., 2018),(a-c), and for the five-year mean model predictions for P3 Ref (d-f) and P3 Oro (g-i). Our model shows much wider ICNC variation than the DARDAR data for all temperature bins presented here. Krämer et al. (2020) provide several reasons that explain the differences between the ICNC of these two observation platforms. Most notably is that DARDAR cannot detect the low ICNC associated with aged thin cirrus clouds at cold temperatures that were observed in the in-situ measurements. This is primarily due to the assumptions made in the retrieval algorithm that is based
on the parameterization by Delanoë et al. (2005) on particle size distribution (PSD) parameter constraints. As Sourdeval et al. (2018) note, this parameterization does not necessarily capture the multi-modality of the ice PSD observed in the in-situ measurements they compared in their study. This culminates in a potential over-prediction of small ice crystals associated with high ICNC values at low temperatures that Krämer et al. (2020) explain is due to the transient nature of homogeneous nucleation and the complexities in observing this process in in-situ field campaigns. This is compounded by the fact that lidar and radar
measurements are not always available simultaneously (Sourdeval et al., 2018).

Mountainous regions such as the Himalayas, the Andes, and the Rockies are already evident with local ICNC maxima in our P3 Ref simulation for all three temperature bins (Figure A2d-f). By adding the orographic velocity component, ICNC spatial heterogeneity is reduced, leading to much higher ICNC over wider areas. We argue this over-predicts high ICNC values and leads to additional warming.

The competition between homogeneous and heterogeneous nucleation in in-situ cirrus is highly uncertain (Cziczo et al., 2013; Krämer et al., 2016; Sourdeval et al., 2018). As the number of ice crystals following a homogeneous nucleation event is highly dependent on the vertical velocity that determines the degree of ice supersaturation (Jensen et al., 2016b), it follows that accounting for vertical velocity variability by including orographic enhancement is requisite. Gasparini and Lohmann (2016), who also used the ECHAM-HAM GCM, showed that even with the orographic parameterization by Joos et al. (2008, 2010)
that the dominant source of cirrus ice crystals at $200\,\mathrm{hPa}$ was through heterogeneous nucleation (see their Figure 3). In our model the opposite is the case (Figure A3). Homogeneously-nucleated ice outweighs heterogeneously-nucleated ice in P3 Ref, and is only enhanced further when including the orographic velocity component such that spatial heterogeneity is also reduced. This is due to the difference between the default ECHAM-HAM microphysics scheme by Lohmann et al. (2007) (2M) and the new P3 scheme (Dietlicher et al., 2018, 2019). With the prognostic sedimentation employed in the latter, leading to slower
ice removal, smaller ice crystals remain in the atmosphere for longer periods than in 2M. Therefore, we argue, that while the enhancement of homogeneous nucleation was required in the model with the 2M scheme, it is no longer required when using the P3 scheme as homogeneous nucleation is not underpredicted relative to in-situ cirrus ice nucleation competition.

Vertical motions in ECHAM6.3 are computed from the sum of a grid mean vertical velocity and a turbulent component based on the TKE parameterization by Brinkop and Roeckner (1995), (Stevens et al., 2013; Neubauer et al., 2019). The scheme allows

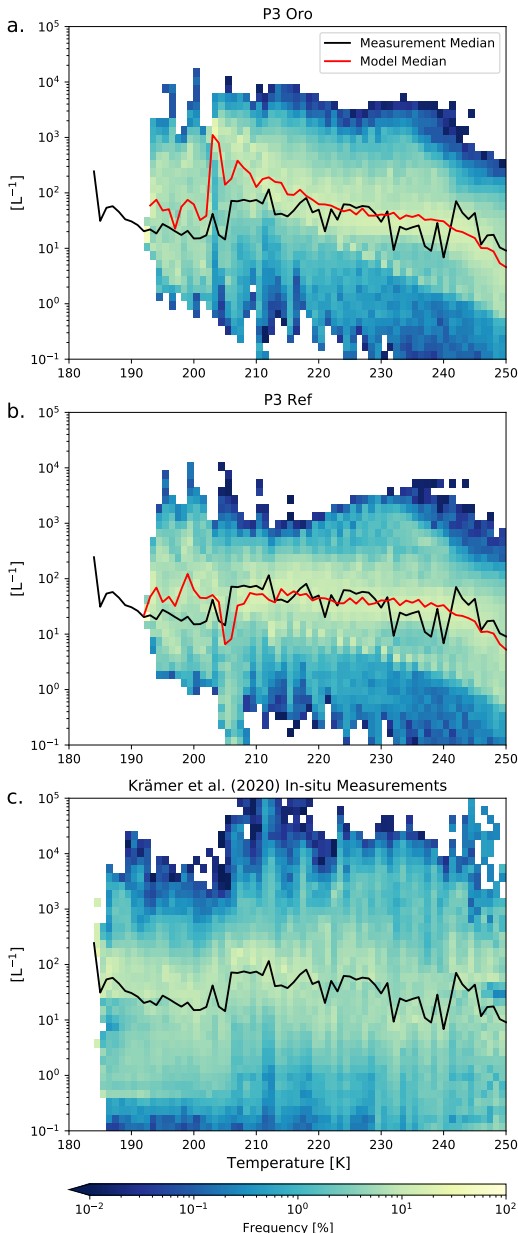

**Figure A1.** ICNC (in $L^{-1}$) frequency diagrams for ice crystals with a diameter of at least $3\,\mu m$ as a function of temperature between $180\,K$ and $250\,K$ binned like in Krämer et al. (2020) for every $1\,K$. The five-year global mean data from the model with the orographic vertical velocity based on Joos et al. (2008, 2010) activated is plotted in (a), the five-year global mean data for the "Full_D19" as in the main manuscript is plotted in (b), and the compilation of in-situ flight data from Krämer et al. (2020) is plotted in (c). The red line in the upper plot represents the binned median ICNC value of the model data, and the black line in both plots is the same value for the observational data.

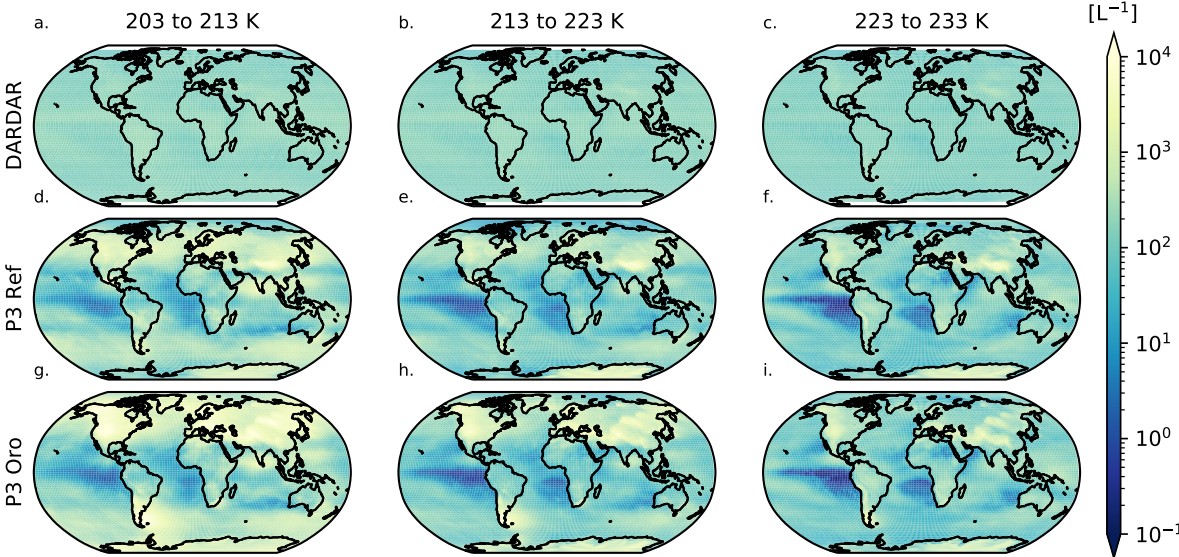

**Figure A2.** Spatial distributions of ICNC in $L^{-1}$ per DARDAR-defined 10 K temperature bin for (a-c) 2006-2016 mean DARDAR ICNC > 5 µm (Sourdeval et al., 2018), and five-year annual mean model ICNC for (d-f) P3 Ref (Full D19 setup per the main text) with no orographic effects, and (g-i) P3 Oro with active orographic effects.

for the momentum transport either horizontally or vertically via turbulent diffusion. Above cloud layers, turbulence is formed as a result of longwave cloud-top cooling. When the orographic gravity wave parameterization is activated as in P3 Oro, the turbulent component of the vertical velocity is computed such that TKE and orographic gravity-waves do not overlap spatially, i.e. turbulent effects are not double-counted within model gridboxes. Figure A4 presents the total vertical velocity for P3 Ref (a) and P3 Oro (b) on the 200 hPa level that is used as input to the cirrus ice nucleation scheme (Section 2 of the main text).

The orographic gravity wave component has a clear impact on the total vertical velocity as expected over mountain ranges such as the Rockies, the European Alps, and the Himalayas. It is unclear why the orographic component is less prominent over the northern Andes in our model, but rather leads to a shift towards southern high latitudes. We also note positive vertical velocity impacts over high-terrain regions such as Greenland and the Antarctic Peninsula when activating the orographic scheme. Positive vertical velocity changes of more than 8 cm/s as seen in Figure A4 greatly impact the formation environment of ice

crystals within cirrus clouds. Kärcher and Lohmann (2002) developed a theoretical framework for simulating homogeneous freezing within young cirrus, which serves as the basis of the cirrus ice nucleation scheme used in our model (Kärcher et al., 2006; Kuebbeler et al., 2014; Muench and Lohmann, 2020). They showed that the number of ice particles resulting from a homogeneous nucleation event is rather insensitive to the particle size distribution, but instead is highly dependent on the strength of the updraft, with higher sensitivity for increasingly lower temperatures. Jensen et al. (2016b) also found a direct

relationship between the number of ice crystals formed by homogeneous nucleation and updraft strength. The behavior we find in our model when activating the orographic gravity wave component is consistent with these theoretical frameworks. The

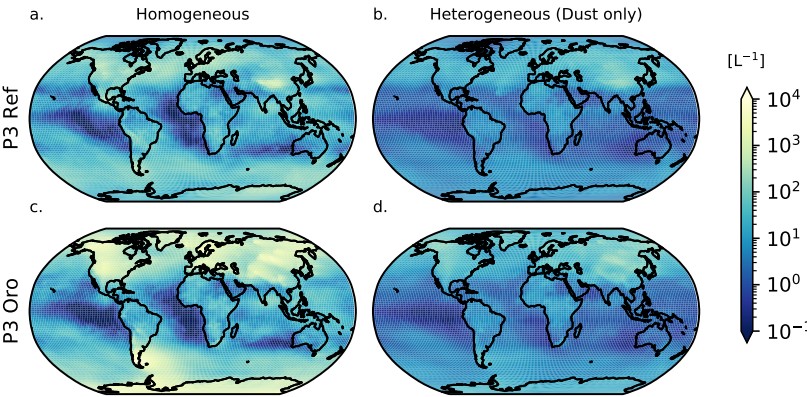

**Figure A3.** Five-year annual mean spatial distributions of in-situ ice number tracers in $L^{-1}$ at $200\,\text{hPa}$ for (a-b) P3 Ref (Full D19 setup per the main text) without orographic effects, and (c-d) P3 Oro with orographic effects active. The first column shows the distribution of the in-situ homogeneously nucleated ice number and the second column shows the total in-situ heterogeneously nucleated ice number, which includes dust only as these are non-seeding simulations.

large median ICNC increase we find with P3 Oro at $202\,\text{K}$ compared to P3 Ref and the in-situ observations by Krämer et al. (2020) in Figure A1 is the direct result of more frequent homogeneous nucleation in our cirrus scheme (Figure A3) in response to stronger vertical velocities. While our model follows directly from theory, this enhancement of the number of ice particles

forming in cirrus clouds with the orographic component activated, worsens model agreement with observations.

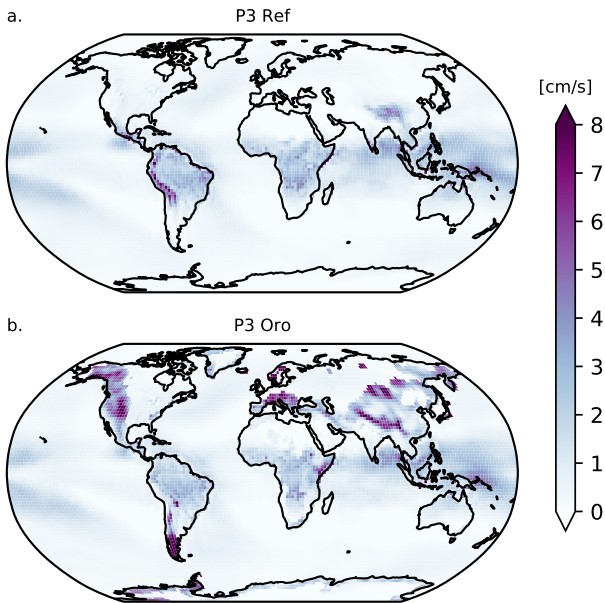

**Figure A4.** Five-year annual mean spatial distributions of the total vertical velocity as calculated in the P3 ice microphysics scheme and sent to the cirrus ice nucleation scheme on the $200\,\mathrm{hPa}$ level for (a.) P3 Ref without the orographic velocity component activated and (b.) P3 Oro with the orographic component of the vertical velocity activated.

*Code and data availability.* The data for this study are available online at: https://doi.org/10.5281/zenodo.6813968 (Tully et al., 2022a). The scripts used for post-processing the raw output data and producing the figures for this manuscript are available online at: https://doi.org/10.5281/zenodo.6983173 (Tully et al., 2022b). In-situ measurement data were provided directly by Martina Krämer (m.kraemer@fz-juelich.de) as well as the DARDAR satellite data by Odran Sourdeval (odran.sourdeval@univ-lille.fr)

*Author contributions.* Colin Tully, David Neubauer and Ulrike Lohmann designed the experiments. Colin Tully ran the model simulations, analysed the data, formulated and ran post-processing and plotting scripts, and wrote the manuscript with comments from all co-authors. Nadja Omanovic contributed to the model development and the data analysis. Ulrike Lohmann and David Neubauer helped with the interpretation of the results.

*Competing interests.* The authors declare that they have no conflict of interest.

*Acknowledgements.* This Project is funded by the European Union under the Grant Agreement No. 875036 (ACACIA). This work was supported by a grant from the Swiss National Supercomputing Centre (CSCS) under project ID s903 and s1144. The authors would like to thank Martina Krämer for graciously providing the in-situ measurement data for model validation and Odran Sourdeval for providing DARDAR satellite data to our group, Jörg Wieder for invaluable help with Python to prepare the figures for this manuscript, Sylvaine Ferrachat for technical assistance running the model and analysing the data, Bernat Jiménez Esteve for providing helpful advice for interpreting our results
for discussion of stratospheric impacts, and Steffen Münch for helping to understand the cirrus scheme and for fruitful discussions on seeding particle implementation potential within the model. Finally, we would like to thank the reviewers and the editor of this manuscript for taking time to provide useful feedback for improving this study.

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
