# Peer review of "Cirrus cloud thinning using a more physically-based ice microphysics scheme in the ECHAM-HAM GCM"

_Atmospheric Chemistry and Physics, 2021_

## Referee Comment (RC2)

**Review for ACP**

Title: Cirrus cloud thinning using a more physically-based ice microphysics scheme in the ECHAM-HAM GCM Author(s): Colin Tully et al. MS No.: acp-2021-685 MS type: Research article

**General Comments:**

This study is well organized and the paper is well written, with a comprehensive introduction. However, the results cannot be taken seriously in my opinion due to the treatment of vertical motions in the ECHAM-HAM GCM simulations. At the bottom of p. 6 and top of p. 7, the paper states that "The adiabatic cooling rate is determined by the vertical velocity, which is represented by a grid-mean value plus a turbulent component based on the turbulent kinetic energy (TKE), (Kuebbeler et al., 2014). Orographic effects on vertical velocity as well as smallscale gravity waves (Kärcher et al., 2006; Joos et al., 2010; Jensen et al., 2016a) in the upper troposphere are not included in this study." Why was the contribution of orographic effects on vertical velocity (w) ignored? As shown in Joos et al. (2008, JGR), outside the tropics over land, this orographic component is the dominant component (i.e., it is much greater than the combined large scale motion component and the TKE component that are the only components considered in this study). Ignoring this orographic component will greatly diminish the global contribution of homogeneous ice nucleation (henceforth hom) to cirrus cloud microphysics, rendering the results of this study a mere modeling exercise that ignores the main driving force responsible for the potential efficacy of CCT. Claiming relevance to CCT therefore appears misguided.

The authors must be familiar with the NASEM report "Reflecting Sunlight" that recommends federal funding for research on SAI, MCB and CCT? In this report on p. 49 it states: "Relative to SAI and MCB, CCT has received relatively less attention, and there is relatively higher uncertainty, due to uncertainty in the current fraction of cirrus formed through homogeneous versus heterogeneous nucleation (Cziczo et al., 2013; Gryspeerdt et al., 2018; Krämer et al., 2016; Mitchell et al., 2016; Mitchell et al., 2018; Sourdeval et al., 2018) ...". Four of the six references cited here provide strong evidence from satellite remote sensing that hom strongly affects the microphysical properties of cirrus clouds over mountainous terrain (and considerably downwind as well). Gasparini and Lohmann (2016, JGR) included the mountain-induced gravity wave parameterization described in Joos et al. (2008, JGR) and Kuebbeler et al. (2014, ACP) for estimating w. In view of this, why was this w contribution ignored in the current study? Evidence relating this orographic component of w to cirrus cloud microphysics and hom is described below.

---

## Author Comment (AC1)

**Cirrus cloud thinning using a more physically-based ice microphysics scheme in the ECHAM-HAM GCM (acp-2021-685)**

*Colin Tully, David Neubauer, Nadja Omanovic, and Ulrike Lohmann*

**Referee #2 Author Response**

Thank you for taking the time to review our manuscript and for providing useful comments on improving this study. We have quoted each of your comments below with our response.

1. **Comment:** Line 197-199: If you are using the Karcher et al 2006 method to represent ice nucleation, which includes water vapor consumption, why is there a need to add a downdraft to update the water vapor consumption? More explanation is needed here.
   a. **Response:** The cirrus model works such that changes to the ice saturation ratio ($S_i$) only occur by the updraft. Therefore, we need some way of altering this variable to account for the effect of water vapor consumption during ice formation events or onto pre-existing ice crystals in a single cirrus model timestep. We calculate an updated updraft velocity every cirrus model timestep, with the deposited water vapor accounted for by the fictitious downdraft. Although the amount of water vapor consumption in one cirrus timestep may not completely deplete ice supersaturation (and therefore shut off further ice formation/growth), the consumption will alter the way the updraft evolves and therefore how the $S_i$ evolves in subsequent cirrus model timesteps. We altered the text to make this clearer for readers.
   b. **Changes in the text at lines 191-194, and 196-198:**

*"… The scheme uses a sub-stepping approach to simulate the temporal evolution of ice saturation during the formation-stage of a cirrus cloud. This is achieved by calculating the balance between the adiabatic cooling of rising air, with the associated saturation increase, and the diffusional growth of ice particles that consume the available water vapor. …*

*As the magnitude of the ice saturation ratio is determined only by the vertical velocity, a fictitious downdraft is introduced at the end of each timestep of the cirrus scheme to quantify the effect of water vapor consumption during new ice formation events or onto pre-existing ice particles (Kuebbeler et al., 2014). The updated vertical velocity therefore determines the evolution of the ice saturation ratio in sub-sequent sub-timesteps. …"*

2. **Comment:** Line 219-220: What is the time step in the cirrus scheme that is referred to here? Is it the 7.5 min time step or the sub-stepping time step? The latter would be more accurate.
   a. **Response:** The cirrus model uses variable sub-stepping that is based on the 7.5 minutes of the model timestep but is calculated according to how $S_i$ will evolve with the input updraft velocity such that changes in $S_i$ equate to 1% for each cirrus model timestep. The cirrus timestep is

updated to 1 second after a threshold freezing process, like homogeneous nucleation, to better "capture the details after the nucleation event" (Münch, 2020) and then readjusted back to a longer timestep after the next cirrus timestep.

We added the detail of the dynamic sub-stepping to the text for clarity.

b. **Change in the text at line 220:**

*"The sub-stepping approach in the cirrus scheme is computed dynamically based on a 1.0 % rate of change of the ice saturation ratio between each sub-timestep."*

3. **Comment:** Lines 221-240: It would be useful to add a table summarizing the different ice nucleating properties, the sizes included, their ice saturation for nucleation and whether the AF treatment is used.
   a. **Response:** This is a great idea. Please find an example of the proposed table below that we will include in the revised manuscript, with the last column indicating whether freezing occurs using active fraction (continuous) or through a threshold process, which is explained in more detail in the text.

| Particle Type | Radius | Critical $S_i$ | Freezing Mechanism | Freezing Method |
|---|---|---|---|---|
| Insoluble dust | 0.05 – 0.5 $\mu m$ | Temperature-dependent, but > 1.1 | Deposition nucleation | Continuous |
| | > 0.5 $\mu m$ | Temperature-dependent, but > 1.2 | | |
| Soluble dust | > 0.05 $\mu m$ | 1.3 | Immersion freezing | Threshold |
| Aqueous Sulphate | All size modes from < 0.005 $\mu m$ to > 0.5 $\mu m$ | ~1.4 | Homogeneous nucleation | Threshold |

4. **Comment:** Lines 253-255: Can you explain a bit more here? What is RHi becomes 100% under a heterogeneous ice simulation?
   a. **Response:** This refers to the default saturation adjustment approach, where any ice supersaturation used to form new ice particles is adjusted down to ice saturation (RH$_i$ = 100%) for the cloud fraction parameterisation and a cirrus cloud is assumed to fully cover a gridbox. With D19, this is no longer the case, as it allows for partial cirrus cloud fractions above ice saturation. We changed the example in the text to explain the difference between the two schemes more clearly. We also added text that provides more description in line with Figure 1 to make it clearer for readers.
   b. **Changes in the text at lines 244-246, 251-255, and 256-258:**

*"… This formulation works well for warm clouds, but as Kuebbeler et al. (2014) and Dietlicher et al. (2018, 2019) note, it breaks down for mixed-phase clouds (T < 273 K) that may or may not include ice, presenting a difficult choice between RH with respect to liquid ($RH_l$) or ice ($RH_i$) to determine cloud fraction. …*

*… Dietlicher et al. (2019) updated the cloud fraction formulation for pure ice clouds to differ from liquid clouds by updating the RH conditions in which an ice cloud can partially cover a gridbox. In this new scheme (hereafter, D19) that we use in this study, ice saturation ($S_i$= 1.0) is set as the lower boundary condition for partial ice cloud fractions. The upper boundary condition for full gridbox coverage for ice clouds is set following the theory for homogeneous nucleation of solution droplets by Koop et al. (2000). …*

*… As a contextual example, if ice were to form at 233 K in an environment with $S_i$ = 1.2, then D19 would calculate an ice cloud fraction <1.0, whereas S89 would adjust the ice supersaturation down to ice saturation and would produce a cloud fraction of 1.0."*

5. **Comment:** Lines 265-267: This sentence needs more explanation. As it is now, I cannot understand what is being said.
    a. **Response:** This refers to the scaling introduced to the available aerosol concentrations. The sentence was changed to make it clear that we apply scaling to the available aerosol concentration for each freezing mode to account for the aerosol particles that already nucleated ice crystals in previous time steps. This is necessary as no in-cloud aerosol tracers are available. The scaling was updated to account for only the fraction of each mode out of the total pre-existing ice. Previously the scaling was applied such that the total pre-existing ice concentration was removed from all modes, which resulted in an overestimation of the in-cloud aerosol concentration and an underestimation of the interstitial aerosol concentration.
    b. **Change in the text at lines 265-269:**

*"… The implementation of these tracers highlighted an error when accounting for the number of aerosols that previously nucleated ice. The aerosol concentration of each freezing mode of the cirrus scheme was scaled by the total amount of pre-existing ice. This approach overestimated the concentrations of in-cloud aerosols and underestimated the interstitial aerosol concentration. We updated the scaling of each mode aerosol concentration to account for the fraction of each mode out of the total pre-existing ice concentration. …"*

6. **Comment:** Line 279: Here you say you have a fractional ice cover scheme, but Lines 253-255 states that there is no fractional cover. When and where do you have fractional ice cover?
    a. **Response:** Agreed. This is an inconsistency in the text, and it leaves out some important detail. The new D19 cloud fraction scheme allows for fractional cirrus coverage under ice formation conditions, as supersaturation is required. The default ECHAM S89 scheme would not allow this, where ice forming above ice saturation would be part of a cloud that would fully cover the gridbox. The manuscript was changed to clarify the description of the fractional ice-cloud cover

scheme related to your Comment 5 above. We also edited this line to remove ambiguity.

b. **Change in the text at line 279:**

*"We performed cirrus seeding simulations using P3 with the cirrus scheme coupled to the new ice-cloud fraction approach (D19) described above."*

7. **Comment:** Lines 316-318: It appears to me that the model is too high from 190-205K by about the same factor as too high from 230-240. Please correct.

   a. **Response:** We would argue that the disagreement between the model and the observations is not as consistent between 190-205K than it is between 230-240K. However, there is a noticeable difference and we amended the text to reflect that. In line with your next comment, we edited the text as well to note that the agreement above 240 K is better than the two temperature ranges quoted here but is slightly underpredicted.

   b. **Change in the text at lines 316-318:**

*"… Model-median ICNC values agree rather well with the observational median at temperatures between roughly 205K and 230K. …"*

8. **Comment:** Lines 319-321: Can you explain this statement better? Why do you think the finding is due to the dust immersion freezing rate? What aspect could cause this?

   a. **Response:** In Figure 2a we see that between 230 and 240 K the model overpredicts ICNC, whereas above 240 K the model slightly underpredicts ICNC. We declare the cirrus regime at 238 K. Therefore, the disagreement in these two temperature ranges could be linked to a mixed-phase process. The Villanueva et al. (2021) study we cite looked into one such process in ECHAM, mixed-phase dust immersion freezing. In that study they compared the ECHAM-default rate-based parameterization for dust immersion freezing to a new active fraction (AF) approach. They note that using the new AF approach in combination with a higher dust-INP efficiency leads to better agreement with satellite observations, as the default rate-based approach underpredicts the amount of ice formation by dust immersion freezing in the mixed-phase regime. This leads to weak ice formation and a higher availability of cloud droplets from the mixed phase regime to be advected into the cirrus regime where they can freeze homogeneously, leading to a high ICNC just below the homogeneous temperature limit (238 K). We believe that the ICNC patterns we find in the model compared to the Krämer et al. (2020) observations reflect this issue. Model ICNC is slightly underpredicted above 240 K due to a too-slow mixed-phase dust immersion freezing rate that allows more cloud droplets to be advected into the cirrus regime and form excess ice at temperatures between 230 and 240 K.

   b. **Change in the text at lines 319-327:**

*"… The small disagreements in these two temperature ranges may be linked to the default parameterization for heterogeneous nucleation on mineral dust particles in mixed-phase clouds in ECHAM. The results by Villanueva et al. (2021) offer an explanation in this regard. In their study, they conducted several sensitivity tests with ECHAM-HAM using the default rate-based immersion freezing scheme by Lohmann and Diehl. (2006) and a newer AF approach based on dust particle surface area and active site density. They found better agreement with satellite-based observations using the AF approach in combination with higher dust particle freezing efficiency as compared to the default rate-based approach, and noted an under-prediction of mixed-phase ice with the latter that led to a higher abundance of cloud droplets being transported into the cirrus regime where they could undergo homogeneous nucleation. …"*

9. **Comment:** Lines 394-395: How can the change in ICNC (200 / L) be larger than the seeding number of 100?
    a. **Response:** The zonal anomalies we are presenting are the ICNC tracers we implemented into the model. The anomaly value can exceed the concentration of seeding particles for two reasons. Firstly, we use a simplified uniform seeding method in our model that does not include seeding-INP budgeting. This means that at every cirrus model timestep the same number of INPs is available and will activate if the $S_i$ value is sufficient. This means we can achieve much higher ICNC values out of the cirrus scheme than the number of available seeding particles. Secondly, the ICNC variables are passed from the cirrus model to the microphysics scheme where they can be advected and/or undergo growth/shrink processes. With the anomaly value being so high, this also indicated that seeding at this concentration leads to more and smaller ice crystals that do not sediment out of the cirrus regime, but rather remain and increase the total ICNC. The combination of these two factors feeds into the overseeding response we find. We added a description related to the first point to the Experimental Setup section in the text to make this clearer for readers.
    b. **Change in the text at Line 286-288:**

*"… For both model configurations (see Table 2) we implemented seeding particles as an additional heterogeneous freezing mode in the cirrus ice-nucleation scheme continuously at every timestep, following on from previous approaches (i.e. without accounting for those that already formed ice). Only gridboxes that are supersaturated with respect to ice (i.e. $S_i > 1.0$) are seeded. …"*

**References**

1. Münch, S., Development of a two-moment cloud scheme with prognostic cloud fraction and investigation of its influence on climate sensitivity in the global climate model ECHAM., *Doctoral Thesis*, https://doi.org/10.3929/ethz-b-000454801, 2020.
2. Villanueva, D., Neubauer, D., Gasparini, B., Ickes, L., and Tegen, I.: Constraining the Impact of Dust-Driven Droplet Freezing on Climate Using Cloud-Top-Phase Observations, Geophysical Research Letters, 48, https://doi.org/https://doi.org/10.1029/2021GL092687, 2021.

---

## Author Comment (AC2)

**Cirrus cloud thinning using a more physically-based ice microphysics scheme in the ECHAM-HAM GCM (acp-2021-685)**

*Colin Tully, David Neubauer, Nadja Omanovic, and Ulrike Lohmann*

**David Mitchell Review Author Response**

Dear David,

Firstly, we would like to thank you for taking the time to review our study and provide detailed feedback on a specific area for improvement. Regarding the orographic component of the vertical velocity by Joos et al. (2008, 2010), it was excluded from the first submission, as in initial tests we believed we were double counting the TKE and orographic components of the vertical velocity in grid cells where orography was active. This resulted in high ICNC values that did not provide us with confidence in our results. It turned out this was not the case and was merely down to a numerical issue, related to parallelization, when using the parameterization in ECHAM6.3 with the new P3 ice microphysics scheme (Morrison and Milbrandt, 2015; Dietlicher et al. 2018, 2019). After reworking the code to make it compatible with P3 we could easily include this vertical velocity component in our simulations. However, after re-running the Full_D19 reference simulation to verify this new approach, we found that including the orographic component is not needed when using the P3 microphysics scheme. In this response we provide our findings that support this claim, and lay out a solution that is implemented in the revised manuscript.

In the manuscript we validate our model with the in-situ measurements by Krämer et al. (2020). Figure 1, below, shows the model validation comparison between our original model that is presented in the manuscript (P3 Ref) and the revised model including the orographic velocity component (P3 Oro) for the reference Full_D19 simulation. It is clear that this extra component has an impact on the modelled ICNC values at lower temperatures (T < 215 K). The model no longer captures the higher frequency of low ICNC values at these temperatures. Instead, the model median is about two orders of magnitude higher than the observed median value. The high frequency ICNC of 1000 $L^{-1}$ around 205 K is not present in the in-situ measurements. Furthermore, at higher temperatures, the model also does not capture frequent low ICNC values.

[Figure]

*Figure 1: ICNC frequency diagrams for ice crystals with a diameter of at least 3 μm as a function of temperature between 180 K and 250 K binned like in Krämer et al. (2020) for every 1 K for P3 without the orographic velocity component (P3 Ref) and with the orographic velocity component (P3 Oro). The five-year global mean data from the model is plotted in the top row and the compilation of in-situ flight data from Krämer et al. (2020) is plotted in the bottom row. The red line in the upper plot represents the binned median ICNC value of the model data, and the black line in both plots is the same value for the observational data.*

We examined this further by comparing our model to the DARDAR satellite remote-sensing ICNC dataset by Sourdeval et al. (2018) in Figure 2. Firstly, our model shows much wider ICNC variation than the DARDAR data for all temperature bins presented here. Muench and Lohmann (2020) note similar biases in their reference simulation associated with convective detrainment as well as the reduction of ice crystal sedimentation enhancement factors with their new cloud scheme. With the prognostic treatment of sedimentation with P3 (Dietlicher et al., 2018,2019), ice crystal removal is also slower and may contribute to high ICNC biases compared to DARDAR. Figure 3 shows the scatter of the modeled ICNC with and without the orographic component activated relative to the DARDAR ICNC dataset. We see that including the orographic component improves the correlation between the model and the satellite only between 223 and 233 K. For the colder temperature bins, activating the orographic velocity parameterization increases the root mean square error and worsens the correlation. However, the DARDAR data is not without its own biases that may not capture wider variability in the observed ICNC. Figure 4 from Krämer et

al. (2020) below shows the DARDAR retrieval frequency as well as ICNC percentiles for both the DARDAR dataset and the in-situ measurements. Firstly, it is noted that the majority of DARDAR measurements (~50%) are at temperatures > 225 K likely due to the overlapping occurrence of in-situ origin and liquid-origin cirrus clouds (Krämer et al., 2020; Wernli et al., 2016; Gasparini et al., 2018). However, what is important here is the variability of the ICNC values in the figure between the in-situ measurements (blue) and the DARDAR observations (red). Krämer et al. (2020) provide several reasons that explain the differences between the ICNC of these two observation platforms. Most notably is that DARDAR cannot detect the low ICNC associated with aged thin cirrus clouds at cold temperatures that were observed in the in-situ measurements. This is primarily due to insufficient sensitivity of the satellite instruments to these low ICNC values. A further bias originates from the assumptions made in the retrieval algorithm that is based on the parameterization by Delanoë et al. (2005) on particle size distribution (PSD) parameter constraints. As Sourdeval et al. (2018) note, this parameterization does not necessarily capture the multi-modality of the ice PSD observed in the in-situ measurements they compared in their study. This culminates in a potential overprediction of small ice crystals associated with high ICNC values at low temperatures that Krämer et al. (2020) explain is due to the transient nature of homogeneous nucleation and the complexities in observing this process in in-situ field campaigns. Finally, while our model shows more variability than the DARDAR data, likely as we can capture more regions of low ICNC at lower temperatures, it is within the range of the instantaneous in-situ measurement variability. In addition, by adding the orographic parameterization, high ICNC biases are enhanced due to the higher frequency of homogeneous nucleation, see below.

[Figure]

*Figure 2: Spatial distribution of ICNC per DARDAR temperature bin. DARDAR data from Sourdeval et al. (2018) is plotted in a-c, 2010 annual mean model data without the orographic velocity component (P3 Ref) is plotted in d-f, and with the orographic velocity component in g-i.*

[Figure]

*Figure 3: Adapted from Lohmann et al. (2020). Scatterplots of ICNC for the 2006-2016 annual mean DARDAR satellite remote-sensing dataset and the 2010 annual mean ECHAM-HAM ICNC for three DARDAR-defined temperature bins. The left column show the spread for the model without the orographic velocity component activated, and the right column the same but with the orographic component active. RMS is the root mean square error, R is the correlation coefficient, and sigma is the standard deviation of the difference between the modeled and satellite ICNC.*

The key comparison is between P3 Ref (Figure 2d-f) and P3 Oro (Figure 2g-i). At the coldest temperatures between 203 and 213 K we find the highest mean ICNC values in our model for both P3 Ref and P3 Oro. In the former one can clearly see that mountainous regions around the Himalayas, the Andes, and the Rockies show local ICNC maxima. This is enhanced by adding the orographic velocity component in P3 Oro such that it weakens regional ICNC heterogeneity. In our P3 Ref simulation local ICNC maxima over mountainous regions become more apparent at higher temperatures, for example the elevated ICNC over Northern Chile and Southern Peru between 223 and 233 K (Figure 2f).

[Figure]

*Figure 4: From Krämer et al. (2020). ICNC-temperature climatology. The colors in the background indicate the DARDAR retrieval frequency. The lines on the plot refer to the 10th and 90th (dotted), then 25th and 75th (dashed), and the 50th (solid) percentiles of DARDAR dataset (red) and the in-situ observations (blue).*

Your main concern regarding our study as we understand it was that homogeneous nucleation within cirrus is underpredicted when excluding the orographic parameterization due to its strong dependence on vertical velocity. Figure 5 presents the ice number tracers at 200 hPa that were added to the model for this study. Like Figures 6 and 9 in the manuscript, these tracers represent the ice formed in-situ in the cirrus scheme that are then passed back to the microphysics scheme. Homogeneous nucleation forms the majority of in-situ cirrus ice in our model regardless of whether we include the orographic velocity component (Figure 5a and 5c). In fact, with this component activated homogeneous nucleation becomes more dominant, and like the DARDAR comparison above, some spatial heterogeneity is no longer evident. Heterogeneous nucleation also increases as critical ice saturation ratio values are reached more easily with the orographic component activated. Furthermore, Figure 6 is taken from Gasparini and Lohmann (2016) and shows the sources of cirrus ice using the default ECHAM microphysics scheme (2M) by Lohmann et al. (2007). Where heterogeneous nucleation was the dominant source of ice at 200 hPa for Gasparini and Lohmann (2016), that is not the case for our model with the P3 ice microphysics. Therefore, we argue that we do not underpredict homogeneous nucleation in in-situ cirrus in our model.

[Figure]

*Figure 5: 2010 annual mean spatial distribution of in-situ ice number tracers on 200 hPa for the model without the orographic velocity component (a-b) and with the orographic velocity component (c-d).*

[Figure]

*Figure 6: From Gasparini and Lohmann (2016), five year annual mean ice sources at 200 hPa for homogeneous nucleated ice, heterogeneously nucleated ice, and detrained ice crystals..*

Based on the findings presented here, we conclude that the inclusion of the orographic velocity component is not needed with the P3 ice microphysics scheme. This highlights the fundamental difference between this newer microphysics scheme and the default ECHAM scheme (2M) by Lohmann et al. (2007) that was used in previous studies from our group (Gasparini and Lohmann, 2016; Gasparini et al., 2017). P3 utilizes prognostic sedimentation of ice hydrometeors by simulating the ice population using a single category, whereas the 2M scheme separates ice into two classes, in-cloud and precipitating. In order to maintain cloud-ice values and cloud radiative properties within the range of observations with the 2M scheme, ice

removal was sped up by enhancing ice crystal aggregation to form snow (Neubauer et al., 2019). This is no longer necessary with P3 as the size-class separation is no longer included in the model, and the updated cloud fraction scheme allows for fractional cirrus cloud cover above ice saturation. The result is much slower ice removal via sedimentation. We can clearly see the effects of this behavior in the plots presented here. Large ICNC values at the coldest temperatures (Figure 2d), originating predominantly from homogeneous nucleation (Figure 5a), are already achieved in our model. With the prognostic sedimentation in our model, these small ice crystals remain in the atmosphere for an extended period. The effect of the orographic velocity component is only to enhance homogeneous nucleation and form more ice that remains in the atmosphere for an even longer time period. We argue that while the orographic component was vital for ensuring homogeneous nucleation was not underpredicted when using the 2M scheme, it is no longer needed with the P3 scheme.

In conclusion, we excluded the orographic velocity component from our study based on our findings. However, as this is the first time the P3 ice microphysics scheme was validated with and without this component, we added an Appendix to the revised manuscript with the figures and explanations presented here.

Sincerely,
Colin Tully (on behalf of all co-authors)

**References**

[revised manuscript text omitted]

---

## Author Comment (AC3)

**Cirrus cloud thinning using a more physically-based ice microphysics scheme in the ECHAM-HAM GCM (acp-2021-685)**

*Colin Tully, David Neubauer, Nadja Omanovic, and Ulrike Lohmann*

**Referee #4 Author Response**

Thank you for taking the time to review our manuscript and providing useful comments on improving this study. We have quoted each of your comments below with our response.

1. **Comment:** It might be helpful to take a step back and better validate both the shortwave (SW) and longwave (LW) cloud radiative effect (CRE) in the model. After model tuning, it was mentioned on lines 275-277 that the net CRE was too negative and that the 5-year global LW CRE is weaker than the observed range. What is this "structural issue within the model" on line 276 referring to and why does it cause a presumably more negative SW CRE? What is the cause for the CRE biases? Is it due to differences in cloud fraction, cloud height or cloud optical thickness?

    a. While SW as well as LW CRE global mean values are within observational ranges the net CRE is not. This was recognized in different configurations of ECHAM-HAM (e.g. Dietlicher et al., 2019; Neubauer et al., 2019). Furthermore, this holds for many different parameter configurations (not shown) and therefore points towards a possible structural error (e.g. Johnson et al., 2020). Neubauer et al. (2019) report an underestimation of stratocumulus clouds but the exact nature of the possible structural problem is not known. Therefore, we amended the text between lines 275-276 to read:

    *"We also note a too negative net CRE after tuning. Dietlicher et al. (2019) state this points to a possible structural problem within the model, which relates to the coarse vertical resolution that results in the under-prediction of low-level clouds (Pelucchi et al., 2021)."*

    b. In response to your question on what could cause this structural issue, Dietlicher et al. (2019) report an improved vertical structure and high-level cloud fraction in ECHAM-HAM with the P3 scheme but an underestimation of mid- and low-level cloud fractions. They further report an underestimation of cloud ice compared to satellite observations, but part of this underestimation is due to the satellite observations, including convective precipitation, which was not included in the ECHAM-HAM-P3 total ice water content. This underestimation is related to a known problem in ECHAM, involving the coarse vertical resolution employed in the model. A recent study by Pelucchi et al. (2021) reported that low-level stratocumulus clouds extent is underpredicted due to poor representation of the vertical relative humidity profile, and low-level cloud occurrence frequency.

c. A few of your comments focused on SW and LW CRE. Therefore, we decided to collate our explanations related to the changes in the manuscript here for ease. This relates to Comments #1, 4d, 5a, and 7a. Tables 2 and 3 (now Tables 4 and 5 in the revised manuscript) were reconfigured to reflect the SW and LW CREs as well as the 95% confidence interval, see the example layout below. We decided to keep Figure 3 as is to avoid a plot that is too cumbersome. Lines 449-461 were reworked to reflect the changes to the Tables.

**Reconfigured Tables 3 and 4 (values provided in revised manuscript)**

| Seeding Concentration [L$^{-1}$] | | 0.1 | 1 | 10 | 100 |
|---|---|---|---|---|---|
| D19 | Net TOA | | | | |
| | Net CRE | | | | |
| | SWCRE | | | | |
| | LWCRE | | | | |
| S89 | Net TOA | | | | |
| | Net CRE | | | | |
| | SWCRE | | | | |
| | LWCRE | | | | |

2. **Comment:** Although P3 is a more physically-based ice microphysics scheme, does it result in ice removal processes that are more *realistic*? Please include a discussion in the context of snowfall.

   a. We partially agree with this comment. The vertical velocity of hydrometeors is no longer tuned using the P3 scheme as it is in the default microphysics scheme (Lohmann et al., 2007), and is instead based on particle mass-to-size relationships. Therefore, in the context of sedimentation velocity, we can state that P3 uses a more realistic approach. However, we cannot state whether the P3 scheme is more realistic in the context of other microphysical processes (e.g. accretion, aggregation, etc.). Therefore, we can merely state that the P3 scheme uses a more physically based representation of ice microphysics, which leads to much larger radiative responses due to slower ice removal processes as compared to the default EHCAM microphysics scheme (Lohmann et al., 2007).

   b. **Changes in the text at lines 15-16 in the abstract, and Point #2 under in conclusions:**

*"This effect is amplified by longer ice residence times in clouds due to the slower removal of ice via sedimentation in the P3 scheme."*

*"The prognostic treatment of sedimentation in the P3 microphysics scheme, leading to slower and more physically-based ice removal, is likely the reason why we find such large seeding responses compared to the study by Gasparini and Lohmann (2016), using the default ECHAM 2M scheme. Our model produces smaller and more numerous ice particles*

*that amplify the already longer ice residence times within clouds to induce a strong positive TOA forcing."*

3. **Comment:** The tuning in the model appears to be quite arbitrary. To reduce the overseeding effect in the model, the authors increased Si,seed to 1.35. Why was this particular value chosen, e.g. why not 1.4 or 1.45?

   **Response to critical Si value:** We chose to increase the critical seeding ice saturation ratio (Si) from 1.05 to 1.35 for two reasons. First, at this value we avoid impacting heterogeneous nucleation on mineral dust as much as possible, which can occur via immersion freezing at a minimum Si of 1.3; dust deposition freezing can initiate at lower Si values. Second, we did not want to make the seeding Si value higher so that seeding particles remain competitive with homogeneous nucleation in our cirrus model, which can occur at a minimum Si value of roughly 1.4. We believe this is justified as this is the first time in a CCT study using a GCM that the sensitivity to the critical Si value was tested. As our results show that this in fact appears to be an important factor determining CCT efficacy, we argue it is justified as a new finding relative to previous CCT studies that could be used to inform further work into this geoengineering proposal.

4. **Comment:** I disagree with the statement that the model "agrees remarkably well with the Kramer et al. (2020) measurements for in-situ formed cirrus" (lines 340-341). The discussion comparing the modelled and measured ICNC appears to be only based on the median values. It appears that there is a large discrepancy in the 215 K to 250 K range for relatively low ICNC (bottom right of plot) which is unexplained. Also, did the Karcher et al. in situ measurements account for the ice crystal shattering effects on probes? Lines 452-454 also seem inaccurate because a small cooling effect is not seen for all seeding concentrations other than S89 Seed100 in Table 3--- it is also small and positive for 5 other values too.

   a. **Response:** This appears to be three separate comments. Therefore, we have divided them into the following sub-points:

   b. **Response to first statement on missing explanation for low ICNC values:** We agree that this explanation should be included in the manuscript. The model agrees well for median values but misses lower ICNC values because we plot annual mean data, whereas the in-situ measurements are instantaneous. **Changes in the text at lines 340-341:**

*"The model does not capture the wide variability of ICNC values as seen in the in-situ measurements, as we compare five-year annual mean model data to instantaneous values recorded during various aircraft campaigns. However, for the purposes of our CCT analysis we find that the model median ICNC as a function of temperature agrees well with the Krämer et al. (2020) measurements for in-situ formed cirrus."*

c. **Response to ice crystal shattering:** Yes, Krämer et al. (2020) considered ice crystal shattering in their results and aimed to minimize its effect on older datasets where possible. See their Appendix A2.4.

d. **Response to second statement on cooling effect:** We disagree with this comment as what you are referring to is the net CRE anomalies in Table 3. What we cover in Lines 452-454 is the net TOA anomalies, of which all the mean values show a slight cooling effect except S89 Seed100. This is also consistent with Table 3. We agree that there should be some discussion of the net CRE anomalies in line with Figure 3 and Table 3 in this paragraph to make it clearer. For ease, please see the response under Comment #1.

5. **Comment:** Given the competing effects of CCT on both the SW and LW CRE, I would recommend including the breakdown of these effects (as opposed to only the net CRE) in Table 2, Table 3 and Figure 3.

   a. **Response:** We agree with this assessment. The breakdown of the SW and LW CREs is useful in order to understand the impact seeding has on cloud properties. In order to avoid a cumbersome figure, we refrained from adding it to Figure 3, but instead expanded Tables 2 and 3 (now Tables 3 and 4 in the revised manuscript) to show this effect. For ease, please see the response under Comment #1.

6. **Comment:** Figure 5: Please carefully explain the unexpected result of the heterogeneous change in ICNC.

   a. **Response:** We agree that this is not covered in enough detail. This was amended in the text to include a better description of this heterogeneous signal, which also links to the Stratospheric Impacts section further down in the results.

   b. **Changes in the text at lines 371-382 and 418-422:**

*"The ICNC anomalies are much clearer and certain for the extreme case, Seed100, than for the Seed1 anomalies (Figure 5c-d). Positive ICNC anomalies exceeding 200 L⁻¹ are shown at all latitudes throughout the troposphere, and into the lower stratosphere at higher latitudes. The anomaly heterogeneity around the tropics is likely due to the proficiency of seeding particles to nucleate ice and hamper homogeneous nucleation in convective outflow regions around the tropopause. …"*

*"… The shift of homogeneous nucleation to lower pressure levels (Figure 6a-b), is likely due to increased LW cloud-top cooling from thicker cirrus cloud following seeding (Possner et al., 2017). This also impacts heterogeneous nucleation on mineral dust particles in the lower stratosphere. As this latter process is not sufficient at consuming water vapor, homogeneous nucleation proceeds to form additional ice crystals. This cloud top cooling effect likely also explains the heterogeneity of the total ICNC anomaly around the tropical tropopause (Figure 5). As there is a clear separation between the troposphere and the stratosphere, these phenomena point to a complex impact on the stratospheric circulation, which we discuss in Section 3.4."*

7. **Comment:** Does the intended side effect of CCT on mixed-phase clouds dominate the intended main effect on CCT? The impact on mixed-phase clouds in Figs 7 and 11 seem quite large. Please discuss. I would also recommend adding this result to the Abstract as well.
   a. **Response:** This was discussed in lines 417-427. In the revised manuscript, this is discussed now discussed between lines 429-442, and 560-576. For Si = 1.05 with a seeding particle concentration of 100 $L^{-1}$ we find an impact on lower-lying MPCs through less efficient MP processes that enhances the SWCRE. However, this is outweighed now by the overseeding effect on LWCRE from more numerous and smaller ice crystals in cirrus clouds. For ease, please see the response under Comment #1. In terms of the abstract, we included this with the line: "*due mostly to rapid cloud adjustments*". However, as this is ambiguous, we amended the text in the abstract at **Line 24-27:**

*"Our results also show feedbacks on lower-lying mixed-phase and liquid clouds through the reduction of ice crystal sedimentation that reduces cloud droplet depletion and results in stronger cloud albedo effects. However, this is outweighed by stronger longwave trapping from cirrus clouds with more numerous and small ice crystals. "*

8. **Comment:** What is the reason for the isolated southern hemisphere cooling effect in the summer due to seeding with Si,seed = 1.35 in Fig. 10?
   a. **Response:** We are unsure whether this refers to the summer quoted on the figure (second row) or southern hemisphere summer (top row). For the former, the isolated areas of SH cooling were due to weaker LWCRE as there is no SWCRE during this season. This points to wintertime seeding having the desired effect in these small regions. If you are referring to the latter (SH summer), which is what we have assumed for the revised text, then small regions of cooling are related to the feedback we find related to MPCs. We agree this is not appropriately covered in the manuscript and have revised the text between lines 546 and 528 to cover more of what we find in Figure 10. Here we quote the text referring to our new results.
   b. **Additional text after lines 528:**

*We also find smaller regions of cooling with net negative TOA responses for Seed1 during NH winter in the SH (summer) around 45 °S, and between the Equator and 30 °S (Figure 10a). The net TOA response is driven mainly by negative SW anomalies, indicating either a shift in cirrus formation pathway or an impact on lower-lying mixed phase clouds.*

*During NH summer the net TOA response is smaller overall than during NH winter. For the Seed1_1.35 zonal mean anomaly we find only small regions of cooling in the NH and in the SH polar regions. However, the uncertainty is wide enough in this case that we cannot determine exact radiative impact in these regions. The small amount cooling shown towards high latitudes in the SH is driven by LW reductions due to a lack of SW radiation in this region during the period, but like the net TOA anomaly is highly uncertain. The few regions of cooling we find in the NH are driven by SW anomalies, highlighting a potential feedback*

*on cirrus cloud formation or on mixed-phase clouds, but are compensated by positive LW anomalies. This is especially noticeable in the northern hemisphere tropics around the location of the Intertropical Convergence Zone (ITCZ). Thicker in-situ cirrus clouds to some extent reflect more SW (Krämer et al., 2020), similar to the Twomey effect for lower-lying liquid or MPCs. However, they also induce a strong compensating LW effect as a result of seeding.*

**Minor:**

1. **Comment:** Please include letter labels for every panel of all multi-panel plots.
   a. **Response:** This is a good point. After double-checking, you are referring to Figures 4, 6, and 9. We amended our plotting scripts for these figures to include lettering for multiple plots and have adjusted the text where necessary to reference a specific plot.
2. **Comment:** Line 302: "cannot not" double negative. I think you mean "cannot"?
   a. **Response:** Thank you for pointing that out. We do mean "cannot". The manuscript was edited to delete the double negative.

**References**

1. Dietlicher, R., Neubauer, D., and Lohmann, U.: Elucidating ice formation pathways in the aerosol–climate model ECHAM6-HAM2, Atmo- spheric Chemistry and Physics, 19, 9061–9080, https://doi.org/10.5194/acp-19-9061-2019, 2019.
2. Johnson, J. S., Regayre, L. A., Yoshioka, M., Pringle, K. J., Turnock, S. T., Browse, J., Sexton, D. M. H., Rostron, J. W., Schutgens, N. A. J., Partridge, D. G., Liu, D., Allan, J. D., Coe, H., Ding, A., Cohen, D. D., Atanacio, A., Vakkari, V., Asmi, E., and Carslaw, K. S.: Robust observational constraint of uncertain aerosol processes and emissions in a climate model and the effect on aerosol radiative forcing, *Atmospheric Chemistry and Physics*, 20, 9491-9524, https://doi.org/10.5194/acp-20-9491-2020, 2020.
3. Krämer, M., Rolf, C., Luebke, A., Afchine, A., Spelten, N., Costa, A., Meyer, J., Zöger, M., Smith, J., Herman, R. L., Buchholz, B., Ebert, V., Baumgardner, D., Borrmann, S., Klingebiel, M., and Avallone, L.: A microphysics guide to cirrus clouds – Part 1: Cirrus types, *Atmospheric Chemistry and Physics*, 16, 3463–3483, https://doi.org/10.5194/acp-16-3463-2016, 2016.
4. Lohmann, U., Stier, P., Hoose, C., Ferrachat, S., Kloster, S., Roeckner, E., and Zhang, J.: Cloud microphysics and aerosol indirect effects in the global climate model ECHAM5-HAM, *Atmospheric Chemistry and Physics*, 7, 3425–3446, https://doi.org/10.5194/acp-7-3425-2007, 2007.
5. Mauritsen, T., Stevens, B., Roeckner, E., Crueger, T., Esch, M., Giorgetta, M., Haak, H., Jungclaus, J., Klocke, D., Matei, D., Mikola- jewicz, U., Notz, D., Pincus, R., Schmidt, H., and Tomassini, L.: Tuning the climate of a global model, *Journal of Advances in Modeling Earth Systems*, 4, https://doi.org/https://doi.org/10.1029/2012MS000154, 2012.
6. Neubauer, D., Ferrachat, S., Siegenthaler-Le Drian, C., Stier, P., Partridge, D. G., Tegen, I., Bey, I., Stanelle, T., Kokkola, H., and Lohmann, U.: The global

aerosol–climate model ECHAM6.3–HAM2.3 – Part 2: Cloud evaluation, aerosol radiative forcing, and climate sensitiv- ity, *Geoscientific Model Development*, 12, 3609–3639, https://doi.org/10.5194/gmd-12-3609-2019, 2019.

7. Pelucchi, P., Neubauer, D., and Lohmann, U.: Vertical grid refinement for stratocumulus clouds in the radiation scheme of the global climate model ECHAM6.3-HAM2.3-P3, Geoscientific Model Development, 14, 5413-5434, https://doi.org/10.5194/gmd-14-5413-2021, 2021.

---

## Referee Report (RR1)

ACP review of Tully et al. (2022), 2nd review

MS No.: acp-2021-685
Title: Cirrus cloud thinning using a more physically-based ice microphysics scheme in the ECHAM-HAM GCM
Author(s): Colin Tully, David Neubauer, Nadja Omanovic, and Ulrike Lohmann
MS type: Research article
Iteration: Revised submission

General Comments:

As I understand the reasoning in the rebuttal, since the orographic component of vertical motions resulted in modeled ice crystal number concentrations (ICNC) that substantially exceeded ICNC from in situ measurements and satellite retrievals, the orographic component was not used in order to achieve better agreement with these observations. When doing model development work, it is natural to strive to narrow the gap between observations and predictions for a given prognostic quantity, such as ICNC. Ideally, this gap narrowing is achieved by improving the model physics, but the practice adopted here for narrowing this gap is to remove a process of known importance to the ICNC; the process producing orographic gravity waves. The justification given is that homogeneous ice nucleation (henceforth hom) in this version of the ECHAM-HAM model is already an important process affecting ICNC within in situ cirrus when using the P3 cloud microphysics scheme. That's great, but that does not provide a rationale for removing the orographic gravity wave component of predicted vertical motions.

That orographic gravity waves are important to ICNC are not unique to Mitchell et al. (2016, 2018); this subject is also discussed in Gryspeerdt et al. (2018, ACP; in reference to their Fig. 1) regarding the satellite retrieval discussed in this rebuttal. That is, the DARDAR results reported in Fig. 1 of Gryspeerdt et al. (2018, ACP) clearly show a strong enhancement of ICNC over mountainous terrain outside the tropics, as do the results presented in Fig. 17 of Mitchell et al. (2018, ACP) and in Mitchell et al. (2016, ACPD) and Mitchell et al. (2020, ACPD). This can also be inferred from Figs. 4 and 7 in Barahona et al. (2017, Nature) as discussed in my first review. Moreover, cirrus cloud fraction is associated with orographic gravity waves as is evident in Fig. 4 of the satellite remote sensing study by Matus and L'Ecuyer (2017, JGR). As mentioned in all but the last of these studies, this is likely due to stronger vertical motions over mountainous terrain where the RHi threshold for hom can be achieved. The fact that ICNC retrieval methods indicate a strong ICNC dependence on orographic waves should motivate the modeling community to include the orographic component of vertical motions, regardless of how it affects ICNC in model simulations. If there is a gap between predictions and observations, try to improve some of the physics, but don't discard it. CCT can only be properly evaluated when all relevant processes are included, and orographic effects appear to be primary in importance.

Another point is to consider the subtle aspects of what determines ICNC in the in situ environment vs. the modeled environment. This was done in Appendix B of Mitchell et al. (2020, ACPD), where ICNC was calculated from the cirrus climatology of Kramer et al. (2020) using the reported in situ ice water contents (IWC) and mean volume ice radius $r_{ice}$, assuming an exponential ice particle size distribution (PSD) as often assumed in a climate model. With this PSD shape constraint, this calculated ICNC was often a factor of 2 or more greater relative to the in situ ICNC reported in Kramer et al. (2020) between 185 K and 220 K, although the calculated and in situ ICNCs were in general agreement between 220 K and 244 K. This may help explain the results under P3 Oro in Fig. 1 of this rebuttal.

Recalling that radiation transfer through clouds is determined by effective diameter $D_e$ and IWC, and not ICNC, more emphasis on $D_e$ (relative to ICNC) would seem appropriate. Note that two PSDs can have the same $D_e$ and IWC while their ICNCs differ considerably.

Specific Comments:

Under conclusions, it is stated: "Such wide differences can be partially attributed to a lack of reliable in-situ observations of cirrus in order to constrain models, though this gap is starting to be closed with more recent studies (Krämer et al., 2016; Krämer et al., 2020)." Since cloud optical properties depend only on $D_e$ and IWC in climate models, it is not clear how the in situ measurements in Krämer et al. (2020) will constrain these optical properties since only the mean volume ice particle radius, $R_v$, is reported in Krämer et al. (2020). We have modeled the relationship between $R_v$ and $D_e$ for gamma PSDs having different values of the PSD dispersion parameter $v$, and for a given $R_v$, a wide variety of $D_e$ are possible, depending on the value of $v$. This is something this ETH group can easily verify. Note that there is no information concerning $v$ in Krämer et al. (2020).

Given the arguments under General Comments, regrettably, I do not understand how the conclusions in this manuscript can be justified.

David Mitchell

---

## Author Response (AR2)

**Cirrus cloud thinning using a more physically-based ice microphysics scheme in the ECHAM-HAM GCM (acp-2021-685)**

*Colin Tully, David Neubauer, Nadja Omanovic, and Ulrike Lohmann*
25th May 2022

**Author Responses**

Dear Prof. Liu

      On behalf of all coauthors, I would like to thank you for serving as the editor of our submission and considering it for publication in ACP. I would also like to thank the reviewers for their useful comments on improvements to our study. Please find our author responses to the second round of reviews from Referees #4 and #3.

Sincerely,
Colin Tully

**Response to Referee #4**

1. **Comment:** Do the authors know why there is a relatively large discrepancy between the model and observations between the 215 to 250 K range for low ICNC in Figure 2?
   a. **Response:** Yes, we believe this is due to the fact that we compare five-year annual mean model data to instantaneous observational data. We added text in the manuscript to describe this specifically
   b. **Changes in the text at lines 343-346:**

*"… The model also does not capture 350 the wide variability of ICNC values as seen in the in-situ measurements, as like the higher frequency of low ICNC values between roughly 205 K and 250 K. This is due to the fact that we compare five-year annual mean model data to instantaneous values recorded during various aircraft campaigns. …"*

2. **Comment:** please include justification of chose for Si=1.05 and 1.35 in the manuscript itself
   a. **Response:** We included explicit justification for our choice of Si,seed values in the revised manuscript
   b. **Changes in the text at lines: 290-297:**

*"The Si,seed of 1.05 follows Storelvmo and Herger (2014) and Gasparini and Lohmann (2016), and is based on suggestions of a hypothetical, highly-efficient seeding particle material. However, it is unclear whether this Si,seed can be applied to a realistic seeding particle material. Mitchell and Finnegan (2009) suggested bismuth tri-iodide, but the specific ice nucleating properties of this material are unknown. Therefore, to test the sensitivity of ice nucleation competition to Si,seed, we conducted additional seeding simulations with all seeding particle concentrations described above, with a Si,seed of 1.35 (Table 3). We chose this relatively high Si,seed value to ensure that seeding can occur in ice supersaturated*

*environments below the lower homogeneous nucleation Si,crit threshold roughly ≥ 1.40 and, in order to be less competitive with background heterogeneous nucleation processes, above the maximum Si,crit for dust of 1.3."*

3. **Comment:** one new comment: Lines 198-201 and lines 207-211 of the version of the manuscript with tracked changes: If water vapour consumption by ice crystal growth is accounted for as a sink of supersaturation, then why does an artificial downdraft need to be introduced in the model at the end of each time step?

   a. **Response:** Supersaturation in our cirrus model can only be updated through changes to the updraft velocity. Therefore, in order to quantify the effect of water vapor consumption, a fictitious downdraft is introduced to counteract the updraft. We expanded our explanation in the text to make this discussion clearer:

   b. **Changes in the text at lines 187-215:**

*"A separate scheme by Kärcher et al. (2006) that was adapted for ECHAM-HAM by Kuebbeler et al. (2014) handles in-situ ice nucleation within cirrus clouds. It simulates the competition for water vapor between heterogeneous and homogeneous nucleation, and between depositional growth onto pre-existing ice particles that are transported into the cirrus regime from 190 deep convective detrainment or from stratiform mixed-phase clouds. The scheme uses a sub-stepping approach to simulate the temporal evolution of ice saturation ratio (Si) in an air parcel rising adiabatically during the formation-stage of a cirrus cloud. Ice formation occurs only when Si reaches the critical values for heterogeneous or homogeneous nucleation (see below). The evolution of Si is determined by the balance between the adiabatic cooling rate of rising air and the diffusional growth of ice particles that consume the available water vapor. As the cooling rate, and therefore the magnitude of Si, is directly related to the strength of vertical velocity, a fictitious downdraft that counteracts the vertical velocity is introduced at the start of each timestep of the cirrus sub-model to quantify the effect of water vapor consumption onto pre-existing ice particles, which includes new ice formation in the previous cirrus sub-model timestep (Kuebbeler et al., 2014). This "effective vertical velocity" (updraft + fictitious downdraft), therefore, determines the magnitude of Si, and is calculated at the end of a single sub-timestep of the cirrus scheme. It is used in the subsequent sub-timestep to update Si.*

*Vertical velocity is represented by a grid-mean value plus a turbulent component based on the turbulent kinetic energy (TKE), (Brinkop and Roeckner, 1995; Kuebbeler et al., 2014). Orographic effects on vertical velocity as well as small-scale gravity waves (Kärcher et al., 2006; Joos et al., 2008, 2010; Jensen et al., 2016a) in the upper troposphere are not included in this study. We provide a short analysis that verifies our model without orographic effects in Appendix A. In summary, by using the new P3 ice microphysics and the updated cirrus ice nucleation schemes, including orographic effects acts to drastically increase cirrus ICNC while reducing spatial heterogeneity. Muench and Lohmann (2020) updated the water vapor consumption by ice, following the diffusional growth equation (Lohmann et al., 2016). The temporal change of the saturation ratio follows such that if the updraft is stronger than the water vapor consumption by pre-exisiting ice and heterogeneous INPs, then it may reach a suitable magnitude for homogeneous nucleation to occur. The opposite is true in weaker updraft regimes or in high INP concentration environments (Kärcher et al., 2006). The sub-stepping approach in the cirrus scheme is computed dynamically based on a 1 % rate of change of the ice saturation ratio between each sub-timestep."*

I am sending you this letter in response to one reviewer who does not concur with the exclusion of orographic-induced gravity waves in our model. In summary, we do not agree with the statements in their response to our rebuttal that was submitted with the revised manuscript in March 2022. We refer to the original rebuttal as the basis for understanding our stance on this issue and provide additional data in this letter to support our argument.

The motivation behind our study is to re-evaluate the relatively new climate intervention proposal, cirrus cloud thinning (CCT), after improvements were made to the representation of ice microphysics in our model. Firstly, we use the Predicted Particle Properties (P3) ice microphysics scheme by Morrison and Milbrandt (2015) that was ported to ECHAM6 by Dietlicher et al. (2018, 2019). The scheme represents the ice population under a single prognostic category, instead of differentiating between in-cloud and precipitating ice like in the default ECHAM6 two-moment (2M) microphysics scheme by Lohmann et al. (2007). In the 2M scheme, once cloud ice grew to a certain size, it was transferred to the snow category and would reach the surface within one model timestep. To maintain radiative balance within the modeled climate, this conversion from cloud-ice to snow was greatly enhanced (Neubauer et al., 2019), thus artificially accelerating ice removal. This is no longer the case using P3, and, as we note in the manuscript, this impacts the lifetime of ice within clouds, which has subsequent effects on cloud fractions and radiative properties. Secondly, our cloud microphysics scheme is coupled to a separate cirrus ice formation scheme based on Kärcher et al. (2006) and Kuebbeler et al. (2014), with updates to the code made by Muench and Lohmann (2020), including using the water vapor deposition equation by Lohmann et al. (2016). The scheme simulates the competition for available water vapor between pre-existing ice, heterogeneous nucleation onto mineral dust particles, and homogeneous nucleation of liquid sulfate aerosols. We made an additional improvement, which is described in the manuscript, that addressed the overestimation of the number of aerosols that previously nucleated ice. Finally, in our study we assess the sensitivity of CCT to the choice of relative humidity-based cloud fraction parameterizations, using the default Sundqvist et al. (1989), (S89) scheme and the updated scheme by Dietlicher et al. (2019), (D19). S89 represents ice cloud fraction analogously to liquid cloud such that at ice saturation, a cirrus cloud fully covers a gridbox. As ice supersaturation is required for ice formation and newly formed cirrus will not necessarily cover the entire coarse resolution of a model gridbox (roughly 160 km x 160 km), we use D19 that allows for partial cirrus cloud gridbox coverage above ice saturation. Therefore, we argue that, cumulatively, these updates made to our model lead to a more physically-based representation of ice microphysics that lends itself to a reassessment of CCT. In the following we provide additional data to discuss the exclusion of the orographic gravity wave component of the vertical velocity based on our specific model setup described above. We cover a comparison to the in-situ data by Krämer et al. (2020) and a discussion on the turbulent vertical velocity in our model.

As you may have read in our original rebuttal, we did not activate the orographic gravity wave parameterization by Joos et al. (2008, 2010) in the first instance as in initial tests we believed we were double counting the turbulent kinetic energy (TKE) and orographic components of the vertical velocity in grid cells where orography was active. This resulted in high ice crystal number concentration (ICNC) values that did not provide us with confidence in our results. It turned out this was not the case and was merely due to a numerical issue, related to parallelization, when using the parameterization in ECHAM6.3 with the P3 ice microphysics scheme. After reworking the code to make it compatible with P3, we could easily include this vertical velocity component in our simulations. However, after re-running our reference simulation (Full_D19 of the manuscript) to verify this new approach, we found that including the orographic component worsens the model agreement with in-situ observations as well as with satellite retrievals of cirrus clouds at increasingly colder temperatures (see our original rebuttal).

We validated our model using the in-situ measurements by Krämer et al. (2020), (K20). Figure 1 shows the model validation comparison between our original model that is presented in the manuscript (P3 Ref) and the revised model including the orographic velocity component (P3 Oro) for the reference, Full_D19, simulation. This figure is now included in Appendix A of the revised manuscript. The most notable feature we find with P3 Oro is the large increase in ICNC between roughly 200 K and 220 K in Figure 1a. The largest difference is at 202 K, where ICNC increased by over two orders of magnitude compared to P3 Ref. There is a similar magnitude of discrepancy between the K20 data and P3 Oro. With the orographic velocity component, the model predicts high frequencies (near 100 %) of ICNC around 2000 $L^{-1}$. Such values in the K20 data (Figure 1c) and P3 Ref (Figure 1b) have a frequency of less than 1%. We note that P3 Ref and P3 Oro show much less variability than the K20 data as they are averaged over five years, whereas the aircraft data are instantaneous. However, we also note that P3 Ref shows excellent agreement in median ICNC values with the K20 data that is not evident for colder cirrus clouds with P3 Oro.

[Figure]

Figure 1: ICNC frequency diagrams for ice crystals with a diameter of at least 3 μm as a function of temperature between 180 K and 250 K binned like in Krämer et al. (2020) for every 1 K for (a.) P3 with the orographic velocity component (P3 Oro), (b.) without the orographic velocity component (P3 Ref), and (c.) the in-situ flight data from Krämer et al. (2020). The red line in the upper two plots represents the binned median ICNC value of the model data, and the black line in all plots is the same value for the observational data.

[Figure]

*Figure 2: Five-year annual mean spatial distribution of the total as calculated in the P3 ice microphysics scheme and sent to the cirrus ice nucleation scheme on the 200 hPa level for (a.) P3 Ref without the orographic velocity component activated and (b.) P3 Oro with the orographic component of the vertical velocity activated.*

Vertical motions in ECHAM6.3 are computed from the sum of a grid mean vertical velocity and a turbulent component based on the turbulent kinetic energy (TKE) parameterization by Brinkop and Roeckner (1995), (Stevens et al., 2013; Neubauer et al., 2019). The scheme allows for the momentum transport either horizontally or vertically via turbulent diffusion. Above cloud layers, turbulence is formed as a result of longwave cloud-top cooling. When the orographic gravity wave parameterization is activated as in P3 Oro, the turbulent component of the vertical velocity is computed such that TKE and orographic gravity-waves do not overlap spatially, i.e. turbulent effects are not double-counted within model gridboxes. Figure 2 presents the total vertical velocity for P3 Ref (a) and P3 Oro (b) on the 200 hPa level that is used as input to the cirrus ice nucleation scheme (Section 2 of the main text). The orographic gravity wave component has a clear impact on the total vertical velocity as expected over mountain ranges such as the Rockies, the European Alps, and the Himalayas. It is unclear why the orographic component is less prominent over the northern Andes in our model, but rather leads to a shift towards southern high latitudes. We also note positive vertical velocity impacts over high-terrain regions such as Greenland and the Antarctic Peninsula when activating the orographic scheme. Positive vertical velocity changes of more than 8 cm/s as seen in Figure 2 greatly impact the formation environment of ice crystals within cirrus clouds. Kärcher and Lohmann (2002) developed a theoretical framework for simulating homogeneous freezing within young cirrus, which serves as the basis of the cirrus ice nucleation scheme used in our model (Kärcher et al., 2006; Kuebbeler

et al., 2014; Muench and Lohmann, 2020). They showed that the number of ice particles resulting from a homogeneous nucleation event is rather insensitive to the particle size distribution, but instead is highly dependent on the strength of the updraft, with higher sensitivity for increasingly lower temperatures. Jensen et al. (2016) also found a direct relationship between the number of ice crystals formed by homogeneous nucleation and updraft strength. The behavior we find in our model when activating the orographic gravity wave component is consistent with these theoretical frameworks. The large median ICNC increase we find with P3 Oro at 202 K compared to P3 Ref and the in-situ observations by Krämer et al. (2020) in Figure 1 is the direct result of more frequent homogeneous nucleation in our cirrus scheme in response to stronger vertical velocities. While our model follows directly from theory, this enhancement of the number of ice particles forming in cirrus clouds with the orographic component activated, worsens model agreement with observations.

We argue that the orographic gravity wave parameterization by Joos et al. (2008, 2010) in its current form is incompatible with ECHAM6.3 when using the P3 ice microphysics scheme. While we accept that including physical processes controlling ice hydrometeor populations is vital to understanding their impacts on cloud properties, the inclusion of additional parameterizations should only be included if they improve our representation of climate system. Based on our findings in the manuscript, our original rebuttal, and what we presented in this letter, the orographic gravity wave velocity parameterization requires additional validation when coupled to P3, which exceeds the scope of this study. We extended the appendix in the revised manuscript to include the figures and the text provided in this response. You will see those changes as well as our adaptations to the Appendix in the tracked changes PDF.

**References**

[revised manuscript text omitted]

---

## Author Response (AR3)

**Cirrus cloud thinning using a more physically-based ice microphysics scheme in the ECHAM-HAM GCM (acp-2021-685)**

*Colin Tully, David Neubauer, Nadja Omanovic, and Ulrike Lohmann*

8th July 2022

**Author Responses**

Dear Prof. Liu

      On behalf of all coauthors, thank you once again for taking the time to handle our manuscript for publication in ACP. I have quoted your comments below with our responses. You will see our changes in the tracked changes PDF.

Sincerely,
Colin Tully

**Main Text & Abstract:**
1. **Comment:** Line 13: "artificial ice-cloud expansion upon ice nucleation". it is unclear, what you mean "ice-cloud expansion upon ice nucleation". ice cloud fraction increase?
   a. **Response:** Agreed, this is unclear for an abstract, we amended the text in the manuscripts to make this clearer
   b. **Changes in the text at line 13:**

*"The most notable response from our extreme case is the reduction of the maximum global-mean net top-of-atmosphere (TOA) radiative anomalies from overseeding by about 50%, from 9.9 Wm⁻² with the original cloud fraction approach, down to 4.9 Wm⁻² using the new cloud fraction RH thresholds that allow partial gridbox coverage of cirrus clouds above ice saturation, unlike the original approach."*

2. **Comment:** Line 20. "In response, we examined the TOA responses..." There are two "respone" in this sentence. please improve.
   a. **Response:** Thank you for finding this. We changed the second "responses" to "anomalies".
   b. **Changes in the text at Line 20:**

*"In response, we examined the TOA anomalies regionally and found that specific regions only show a small potential for targeted CCT, which is partially enhanced by using the larger $S_{i,seed}$."*

3. **Comment:** Lines 124-127, 145-146, and 772-774. The large differences in the outcome of CCT studies between ECHAM and CESM-CAM5 is mainly due to the treatment of ice nucleation in cirrus clouds in the two models. While the manuscript introduces the ice nucleation scheme and preexisting ice treatment in ECHAM, the authors may need to refer the related cirrus schemes in CESM-CAM5 to help understanding. For example, the ice

nucleation in CAM5 is based on Liu and Penner (2005) which treats both homogeneous and heterogeneous ice nucleation, and also includes the preexisting ice treatment (Shi et al., ACP, 2015). CAM5 has a high frequency of homogeneous ice nucleation. The authors may add some discussion of cirrus cloud states simulated in CAM5 to better explain the differences from ECHAM.

a. **Response:** Based on our search and a review of the literature, all CCT studies to date with CAM5 (Storelvmo et al. 2013; Storelvmo and Herger 2014; Storelvmo et al., 2014; Penner et al., 2015; Gasparini et al., 2020) used the cirrus ice microphysics scheme by Barahona and Nenes (2008, 2009). Only Storelvmo et al. (2013) "partly replaced" the older scheme by, as they quote, Liu et al. (2007) with the scheme by Barahona and Nenes (2008, 2009). Therefore, we based our discussion in the revised manuscript on the Barahona and Nenes (2008, 2009) scheme, with reference to Liu and Penner (2005). In addition, Penner et al. (2015) were the only ones using CAM5 with pre-existing ice in some of their simulations, following Shi et al. (2015) as you cite above. This made a difference in the outcome of CCT compared to CAM5 simulations that did not include pre-existing ice. As Gasparini et al. (2020) state CAM5, being part of the wider CESM1 that was used for studying CCT to date, does not include pre-existing ice, whereas CESM2 does. We agree that some more detail can be provided about the differences in cirrus microphysics in CAM5 and ECHAM6, as this is a large point of uncertainty in determining CCT efficacy. The largest difference between the two models is the inclusion of pre-existing ice as we describe above. Moreover, in studies using ECHAM6, the impact of cirrus seeding is to produce some new cirrus clouds, whereas this does not appear to be the case with CAM5. We also argue that a full review of the specific parameterizations governing homogeneous and heterogeneous nucleation within cirrus clouds in the two models is beyond the scope of this specific study, but some short reference to this was added. Therefore, for brevity, we added this discussion only to the discussion section and moved some related text from the conclusions to avoid repetition.

b. **Changes made in the text from Line 770 in the Discussion and Line 837 in the Conclusions:**

[revised manuscript text omitted]

4. **Comment:** Line 205. "updated cirrus ice nucleation schemes". do you mean "updated cirrus ice fraction scheme"?
   a. **Response:** That is a typo. Thank you for finding that. We meant only the scheme that simulates in-situ ice nucleation within cirrus, as quoted on Lines 187-188, with updates following Muench and Lohmann (2020). We amended the text to make this clearer.
   b. **Change in the text at line 205:**

*"In summary, by using the new P3 ice microphysics with the in-situ cirrus ice nucleation scheme (Muench and Lohmann, 2020), including orographic effects acts to drastically increase cirrus ICNC while reducing spatial heterogeneity, in worse agreement with observations.*

**Appendix:**
1. **Comment:** Line 812: Figure A2. why the model ICNC is so low (<10 L-1) in SH? this is different from Figure A3 (for homogeneously nucleated ice).
   a. **Response:** Thank you for finding this inconsistency in our appendix. The ICNC diagnostics we were using to compare to the DARDAR data were not prescribed correctly and, therefore, lacked some ice number data. We fixed this issue and re-ran our simulations and find much better agreement not only with the DARDAR data in Figure A2, but also with our cirrus ice number tracers in Figure A3.

2. **Comment:** Figure A1. Where and when the ice number in P3 Oro increases substantially near 202 K in Figure A1. Is this strong increase a result of increased vertical velocity?
   a. **Response:** Yes, this is due to the strong increase in vertical velocity that is used as input to our cirrus scheme, as shown in Figure A4 of the Appendix. Our cirrus model is based on the fundamental theoretical work by Kärcher and Lohmann (2002) on homogeneous nucleation. Their study found a strong link between the strength of the vertical velocity and the number of ice crystals formed following a homogeneous nucleation event. We find the same behavior in our model when activating the orographic vertical velocity parameterization by Joos et al. (2008, 2010), with a noticeable increase in ICNC at cold temperatures (Figure A2) that consists mostly of ice formed by homogeneous nucleation (Figure A3).

**Minor comments:**

1. Line 828. change "due" to "due to".
2. Line 843. two "terrain" here.
3. **Combined response:** thank you for finding these typos. These are amended in the text.

**References**

[revised manuscript text omitted]

---

## Author Response (AR4)

**Cirrus cloud thinning using a more physically-based ice microphysics scheme in the ECHAM-HAM GCM (acp-2021-685)**

*Colin Tully, David Neubauer, Nadja Omanovic, and Ulrike Lohmann*

11th August 2022

**Author Response**

Dear Prof. Liu

Thank you for agreeing to extend the review process for our manuscript. In line with the review process, we prepared this short letter to explain the changes in the manuscript since our previous upload.

Shortly after the previous upload, we found in our reference simulation (Full_D19) that the total in-situ cirrus heterogeneous ice number tracer, that we use in Figures 6 & 9 in the manuscript, was inconsistent with the ice number tracer for in-situ cirrus heterogeneous nucleation on mineral dust particles. After investigating this further, we found this originates from the sub-timestepping process in the P3 ice microphysics scheme. As these quantities are diagnostic tracers, they ultimately do not change the outcome of our study, and we wanted to ensure this before finalizing our manuscript. Our solution in this case is to, instead, use the sum of the heterogeneously nucleated ice number sources in cirrus as the total cirrus heterogeneous ice number:

$$Total\ HET = HET\ on\ mineral\ dust + HET\ on\ seeding\ particles,$$

where the seeding particle signal is equal to zero in our reference cases Full_D19 and Full_S89.

This change is reflected in our revised manuscript in Figures 6 & 9 as well as on Zenodo with a new version of the data analysis and plotting scripts repository. However, as this does not change our results or the conclusions of our paper, the tracked changes do not show any revisions in the text.

Sincerely,
Colin Tully